# Maintenance and transformation of representational formats during working memory prioritization

Daniel Pacheco-Estefan [1,10] ✉, Marie-Christin Fellner[1,10], Lukas Kunz [2], Hui Zhang[1], Peter Reinacher [3,4], Charlotte Roy[5], Armin Brandt[5], Andreas Schulze-Bonhage [5], Linglin Yang[6], Shuang Wang[7], Jing Liu[8], Gui Xue [9] & Nikolai Axmacher [1,9]

Visual working memory depends on both material-specific brain areas in the ventral visual stream (VVS) that support the maintenance of stimulus representations and on regions in the prefrontal cortex (PFC) that control these representations. How executive control prioritizes working memory contents and whether this affects their representational formats remains an open question, however. Here, we analyzed intracranial EEG (iEEG) recordings in epilepsy patients with electrodes in VVS and PFC who performed a multi-item working memory task involving a retro-cue. We employed Representational Similarity Analysis (RSA) with various Deep Neural Network (DNN) architectures to investigate the representational format of prioritized VWM content. While recurrent DNN representations matched PFC representations in the beta band (15–29 Hz) following the retro-cue, they corresponded to VVS representations in a lower frequency range (3–14 Hz) towards the end of the maintenance period. Our findings highlight the distinct coding schemes and representational formats of prioritized content in VVS and PFC.

Visual working memory (VWM) refers to the ability to store visual information for a short period of time and to flexibly manipulate this information according to task demands. One essential aspect of VWM memory is prioritization, i.e., the ability to selectively allocate attention to particular features or items depending on behavioral or cognitive requirements. Influential theories have proposed that WM prioritization entails the transformation of maintained representations from a purely mnemonic to a task-optimized state[1,2]. On a neurophysiological level, these accounts predict that working memory prioritization involves a task-dependent transformation of representational patterns in executive control areas which can be disentangled from a mnemonic coding scheme that maintains perceptual stimulus features in sensory brain regions[1,3–5]. Here we set out to test this prediction. We analyzed the representational format of VWM stimuli using electrophysiological recordings in human epilepsy patients implanted with electrodes in ventral visual stream (VVS) and/or prefrontal cortex (PFC).

[1]Department of Neuropsychology, Institute of Cognitive Neuroscience, Faculty of Psychology, Ruhr University Bochum, 44801 Bochum, Germany. [2]Department of Epileptology, University Hospital Bonn, Bonn, Germany. [3]Department of Stereotactic and Functional Neurosurgery, Medical Center – Faculty of Medicine, University of Freiburg, Freiburg, Germany. [4]Fraunhofer Institute for Laser Technology, Aachen, Germany. [5]Epilepsy Center, Medical Center – Faculty of Medicine, University of Freiburg, Freiburg, Germany. [6]Department of Psychiatry, Second Affiliated Hospital, School of medicine, Zhejiang University, Hangzhou, China. [7]Department of Neurology, Epilepsy center, Second Affiliated Hospital, School of medicine, Zhejiang University, Hangzhou, China. [8]Department of Applied Social Sciences, The Hong Kong Polytechnic University, Hong Kong, Hong Kong SAR. [9]State Key Laboratory of Cognitive Neuroscience and Learning and IDG/McGovern Institute for Brain Research, Beijing Normal University, Beijing 100875, PR China. [10]These authors contributed equally: Daniel Pacheco-Estefan, Marie-Christin Fellner. ✉e-mail: Daniel.PachecoEstefan@ruhr-uni-bochum.de

There is now abundant evidence on the neural correlates of VWM control processes in humans and animals[6–10]. Early studies focused on prioritization of to-be-encoded items, using paradigms in which participants were asked to selectively attend to particular items before these items were shown (e.g., refs. 6,11–14). More recent investigations often employed retrospective cueing paradigms, in which prioritization is applied to information after its encoding into VWM[10,15–18]. These studies revealed that prefrontal and parietal regions which underlie the allocation of attention during perception are also engaged in the prioritization of items in VWM[2,7–9,19–22]. Notably, a recent meta-analysis observed selective responses to retro-cues but not to cues that allocate attention prior to item encoding in VWM in various prefrontal areas[1,23]. This literature suggests that prioritization affects VWM representations in the PFC, yet this prediction has not been tested experimentally in humans.

The critical role of the PFC in WM prioritization is commonly believed to depend on dynamic recurrent computations. A classical model of WM suggests that persistent activity depends on reverberatory excitation within a local recurrent neural network[24,25]. Computational studies have shown that recurrence is crucial for the selection and integration of task-relevant features in the PFC[26], the integration of working memory and planning[27], the flexibility of WM and the avoidance of interference in the presence of competing representations[28], and—most importantly for our study—WM prioritization[29,30]. Recurrent computations might be particularly relevant for selective attention to specific features or items in WM because they enable the stabilization of reverberating activity in attractor states that modulate the excitability of assemblies which represent prioritized contents[24,25,31]. In addition to their theorized role in PFC prioritization, recurrent computations have been proposed to be critical for information processing in the VVS during visual perception[32,33], and for offline 'generation' of stimuli during visual imagery[34]. However, a specific role of recurrency in the VVS for VWM maintenance has not been previously investigated.

In addition (and possibly related) to the relevance of recurrent computations, theories have emphasized the important role of brain oscillations for VWM, in particular for the prioritization process. Oscillatory activity in the gamma frequency range (50–120 Hz) is thought to convey bottom-up information during VWM encoding, while oscillations at beta frequency (20–35 Hz) are supposed to provide top-down control over VWM contents[4,35–38]. The significance of these oscillatory patterns has been validated experimentally in a series of studies in macaques[4,20,35]. Furthermore, a recent study in humans confirmed the crucial role of gamma-band activity (30–75 Hz) for conveying bottom-up information from lower-level visual areas to regions processing higher-level information[36]. In addition to their role in top-down control over WM content, several studies have now associated beta oscillations with the reactivation of stimulus-specific activity during the VWM prioritization process. Content-specific beta activity has been shown to carry information about internalized task rules[39], stimulus categories[40–42], scalar magnitudes[43,44] and perceptual decisions[45]; for review, see ref. 46. These studies highlight the role of beta oscillations in encoding task-relevant stimulus properties.

In humans, intracranial EEG (iEEG) recordings in epilepsy patients have been used to investigate the neurophysiological patterns underlying content-specific memory representations. This research has employed multivariate analysis techniques, such as pattern classification and representational similarity analysis (RSA), to identify representations of specific stimuli[47,48]. Studies have demonstrated that frequency-specific representations in the gamma, beta and theta (3–8 Hz) frequency bands contain item- and category-specific information, playing a crucial role in episodic memory retrieval[49,50]. In addition to identifying the relevant oscillatory frequencies that carry representational content during visual perception and episodic memory, recent iEEG studies have investigated the 'formats' of VWM representations. This research employed deep neural networks (DNNs) to investigate how different aspects of natural images are represented in the brain during mnemonic processing. These studies assume that mnemonic representations require specialized circuits for processing distinct aspects (or formats) of natural images, from low-level sensory features to higher-level contents and conceptual/semantic information[51–56]. Indeed, several studies assessed the different representational formats during VWM encoding and maintenance and demonstrated substantial transformations of VWM representations into a format that aligns with late layers of a convolutional DNN[57,58]. While these results and methodological advancements have provided valuable insights into the format of VWM representations, no study so far has investigated the representational transformations that accompany VWM prioritization in humans. Thus, whether the prioritization of VWM representations involves a change in the representational format of the stored content and distinct coding schemes of attended (i.e., task-relevant) items is currently unknown.

Here, we leveraged the heuristic potential of DNNs as models of visual representation, the flexibility of RSA, and the high spatio-temporal resolution of iEEG to investigate this topic. We analyzed electrophysiological activity from VVS and PFC while patients performed a multi-item VWM paradigm involving a retro-cue. Participants encoded a sequence of three images and were then prompted by a retro-cue to maintain either one of these items or all items (Fig. 1A; Methods). The objects belonged to six categories, each containing ten exemplars (60 images in total). With the exception of the behavioral data, we only analyzed activity during the single item condition in this study, given our focus on the prioritization process. This experimental design allowed us to evaluate how information about specific contents is represented in the brain during initial encoding and how it is transformed due to the retro-cue, both in terms of representational formats and regarding the frequencies of brain oscillations in VVS (438 electrodes) and PFC (146 electrodes; Fig. 1B). We hypothesized that frequency-specific representations would reflect bottom-up storage and top-down information transfer, respectively, with a particular role of gamma and beta oscillations[4,35–38]. Specifically, we predicted that oscillatory PFC activity in the beta frequency range may reflect representational transformations due to top-down control following the retro-cue[4]. In addition, we expected recurrent convolutional architectures to better explain representations than feedforward DNNs during VWM maintenance[25,28,32,33,59,60].

## Results

### Behavioral results

Successful prioritization in the single-item condition should result in better performance than in the multi-item trials. Indeed, we found that participants performed significantly better in single-item trials (proportion correct trials: $0.8 \pm 0.12$) than in the multi-item condition ($0.75 \pm 0.13$; $t(31) = 3.21$, $p = 0.0031$; Fig. 1C). This suggests that participants followed instructions and benefited from prioritizing task-relevant representations in the single-item trials.

### Maintenance and transformation of category-specific representations

We investigated the electrophysiological patterns supporting the representation of category-specific information in VVS and PFC. As a first approach, we assessed the presence of categorical representations, employing RSA and a simple model of category information (Fig. 2A). We constructed an item-by-item representational similarity matrix (RSM) reflecting the hypothesis that items of the same category would elicit more similar patterns of brain activity compared to items of different categories (Fig. 2A). We correlated this model RSM with temporally resolved neural RSMs (windows of 500 ms, overlapping by 400 ms). Representational patterns included power values across electrodes ($16.85 \pm 8.92$ electrodes in VVS, $9.73 \pm 11.1$ in PFC; Mean ± STD) and time points (5 time points of 100 ms in each time window;

Fig. 1D, Methods) and were analyzed separately in each of 52 different frequencies between 3 and 150 Hz. To determine the similarity between feature vectors, we used Spearman's Rho[50,57].

Our simple category model revealed a marked presence of categorical information during encoding in the VVS. This was observed in a significant frequency cluster ranging from 3–120 Hz that started immediately after stimulus presentation and lasted for the whole encoding period (0.8 s; $p = 0.001$). In the PFC, we observed two clusters of significant fit in the beta (17–28 Hz; 200–800 ms; $p = 0.001$) and the theta frequency range (3–7 Hz; 200–600 ms; $p = 0.044$; Fig. 2B, left). During maintenance, we did not observe any significant fits between model and neural RSMs in either VVS or PFC (VVS: all $p > 0.51$; PFC: all $p > 0.105$; Fig. 2B, right).

The absence of fit of the category model during maintenance might be attributed to a weakening of the representations during the maintenance period—e.g., due to a decrease in signal to noise ratio—or to a rapid transformation of activity patterns during encoding[58]. To evaluate whether transformed activity patterns from encoding reoccur

during the maintenance period, we performed a category-specific pattern similarity analysis (Methods). This analysis involved contrasting correlations of items belonging to the same category with correlations of items from different categories (Fig. 2C, top), both during encoding (encoding-encoding similarity; EES) and between encoding and maintenance (encoding-maintenance similarity, EMS; Fig. 2C, bottom). Notably, while the category model can track the presence of categorical representations at the level of the representational geometry of our stimuli set, the EES and EMS analyses test for re-occurrence of category-related neural activity patterns from different encoding periods. This analysis was conducted in five conventional frequency bands (theta, 3–8 Hz; alpha, 9–12 Hz; beta, 13–29 Hz; low-gamma, 30–75 Hz; high-gamma, 75–150 Hz), with electrodes, time points (including both matching and non-matching time points; see Methods), and frequencies in each band as features.

We first analyzed the timing and temporal stability of representations during encoding, using EES. Consistent with the results observed in the category model analysis, the EES analysis revealed

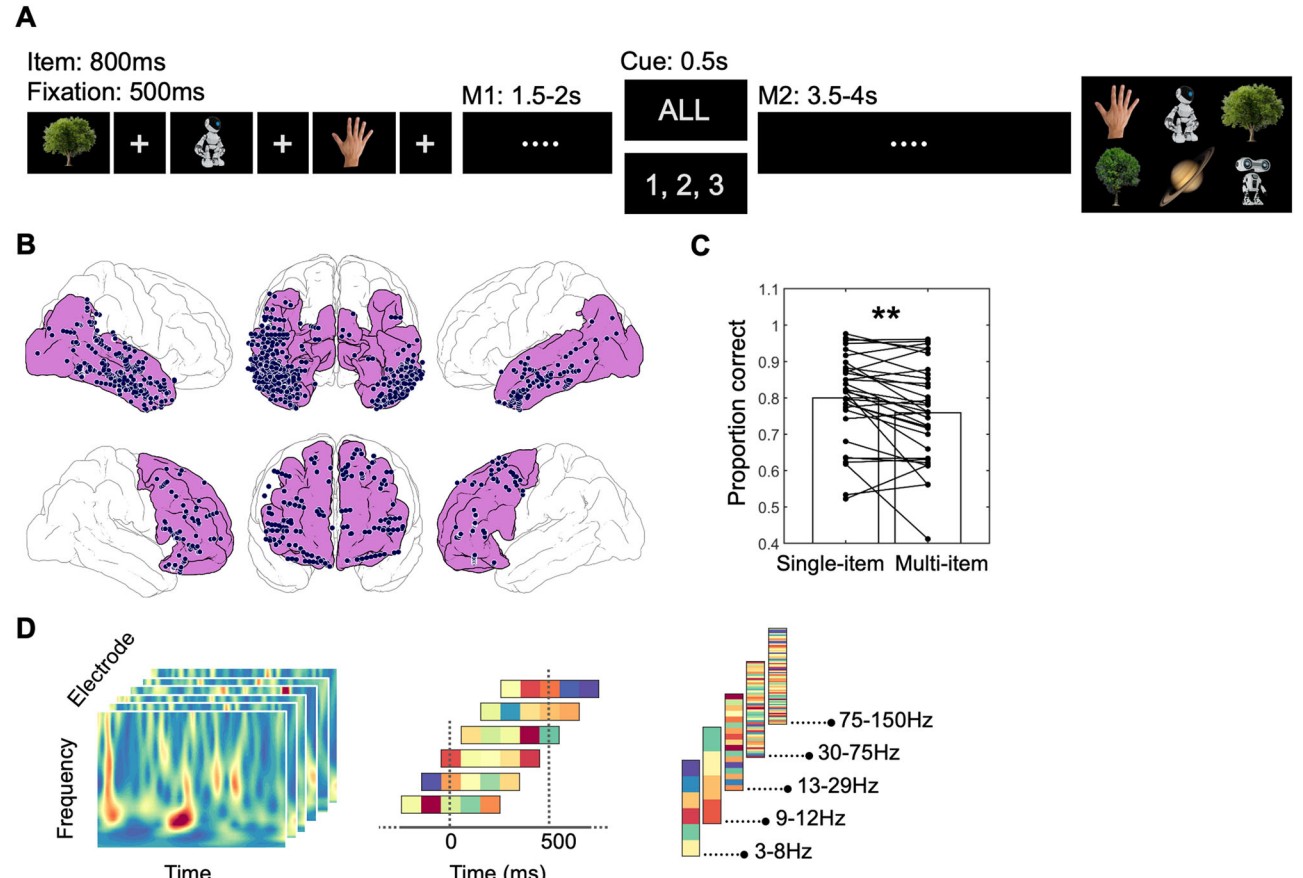

**Fig. 1 | Experimental procedure, electrode coverage, and behavioral results. A** Participants encoded a sequence of 3 images of natural objects and were asked to remember this content during two subsequent maintenance periods that were separated by a retro-cue (M1 – retro-cue – M2). The cue prompted participants to either selectively maintain items at particular list positions (single-item trials, "1, 2, 3" in the figure) or to maintain all items in their order of presentation (multi-item trials, "All" in the figure). During the probe, six items were presented, which included all encoded items and three exemplars from previously presented or novel categories (Methods). The figure displays representative images very similar to those shown during the experiment, in compliance with a CC BY license (https://creativecommons.org/licenses/by/4.0/). **B** Electrode implantation included 438 electrodes in the ventral visual stream (VVS, $N = 28$ participants; top) and 146 electrodes in the prefrontal cortex (PFC, $N = 16$ participants; bottom). Cortical areas included in each region are highlighted in pink. **C** Behavioral performance was

significantly higher for single as compared to multi-item trials in our patient group (paired $t$-test, two-sided, $n = 32$). **D** Left: Representational patterns in the RSA analyses included neural activity across electrodes, time points (5 time points in each 500 ms window), and frequencies. Spearman correlations were computed in windows of 500 ms, incrementing in 100 ms steps (middle). Analyses were performed in individual frequencies in the 3-150 Hz range in the model-based RSA analyses, and within different frequency bands (theta, alpha, beta, low-gamma, high-gamma) in the contrast-based analyses. Source data are provided as a Source Data file. Image sources (**A**): Tree 1: https://www.istockphoto.com/en/portfolio/YutthasartYanakornsiri; Tree 2: https://www.istockphoto.com/en/portfolio/Coldimages; Robot 1 and 2: https://www.istockphoto.com/en/portfolio/Ociacia; Hand: https://www.istockphoto.com/en/portfolio/Hanis; Planet: https://www.istockphoto.com/en/portfolio/GeorgeManga. **$p < 0.01$.

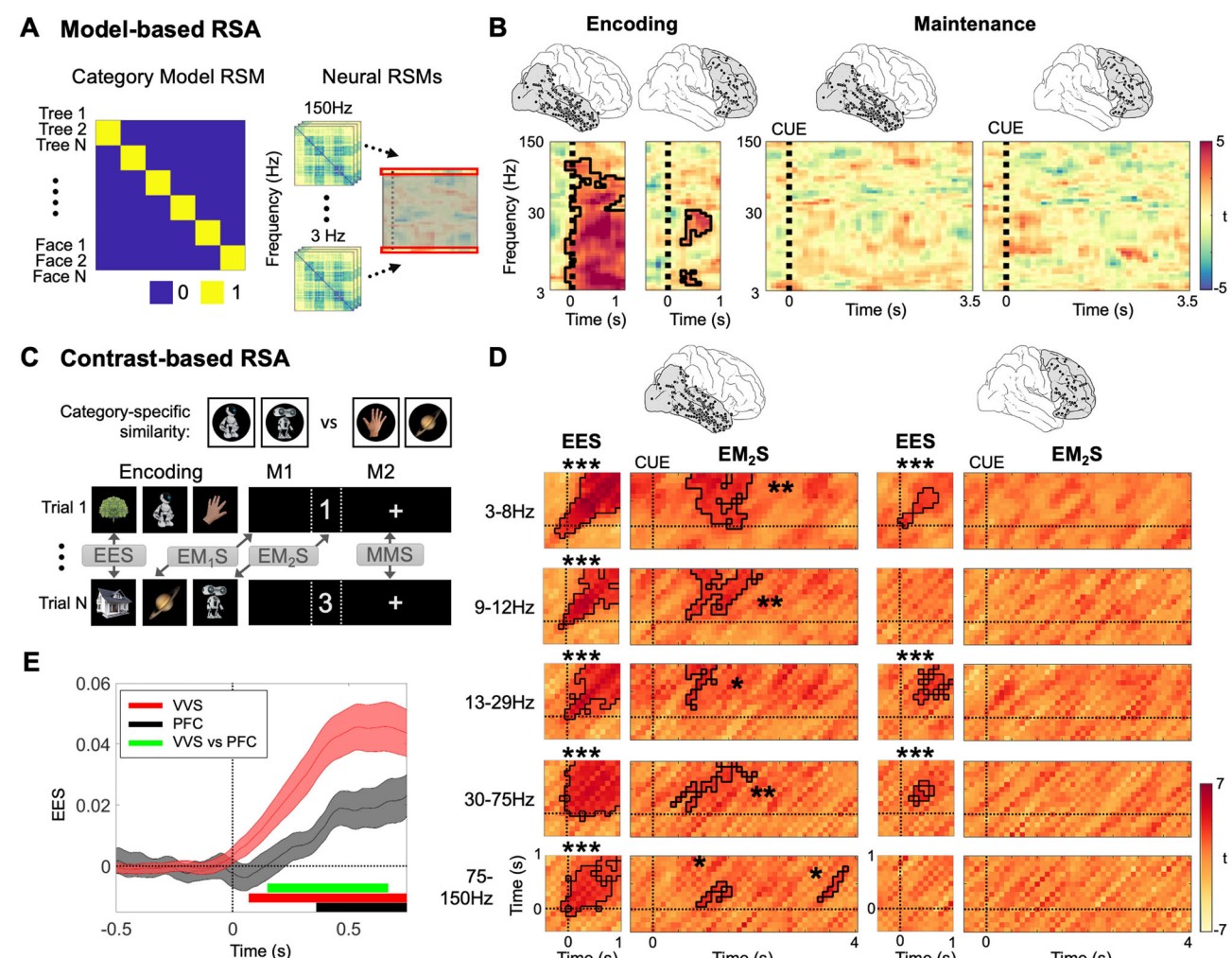

**Fig. 2 | Encoding and maintenance of category-specific representations.**
**A** Model-based RSA. Representational similarity matrix (RSM) reflecting the
hypothesis that category information structures the representational geometry of
stimuli (left) were correlated with a time-series of neural RSMs (right) at each
individual frequency. **B** Fit of the category model in the VVS and the PFC during the
encoding (left) and maintenance (right) period. Zero indicates the onset of image
presentation during encoding, and the onset of the presentation of the cue during
maintenance. **C** Contrast-based RSA. Top: Category-specific similarity was com-
puted by contrasting correlations between (different) items of the same category vs
items of different categories. Bottom: Similarity was calculated between the
encoding periods of different trials (encoding-encoding similarity, EES), between
the encoding and the maintenance periods of different trials (encoding-main-
tenance similarity, EMS), and between maintenance periods of different trials
(maintenance-maintenance similarity, MMS). Note that two types of EMS analysis
were conducted for the first and second maintenance periods (EM₁S, see Supple-
mentary Fig 2, and EM₂S). **D** EES and EM₂S analyses: Category contrasts for five
frequency bands during encoding and maintenance in VVS (left) and PFC (right). In
the EES plots, zero indicates the onset of image presentation on both time axes. In

the EM₂S plots, zero indicates image onset during encoding and retro-cue onset
during maintenance, respectively. Significant differences between same and dif-
ferent categories were assessed using two-sided paired $t$-tests at each time bin.
Significant time periods surviving correction for multiple comparisons using
cluster-based permutation statistics are outlined in black. **E** Category specificity
analysis at higher temporal resolution showing different latencies of effects in VVS
(red) and PFC (black). Each line shows the time course of within minus between
category correlations in each region (Mean ± S.E.). Horizontal bars indicate time-
periods when EES values are significantly different from zero in each region, and
significantly different between the two regions (green). Please note that t-maps in
**B** and **D** have the same color scale, indicated in the color bar at the right of each
panel. Source data are provided as a Source Data file. Images in **C** are published in
compliance with a CC BY license (https://creativecommons.org/licenses/by/4.0/).
Sources: Robot 1 and 2: https://www.istockphoto.com/en/portfolio/Ociacia; Hand:
https://www.istockphoto.com/en/portfolio/Hanis; Planet: https://www.
istockphoto.com/en/portfolio/GeorgeManga. Tree 1: https://www.istockphoto.
com/en/portfolio/YutthasartYanakornsiri; House: https://www.istockphoto.com/
en/portfolio/SittidhetJoollasawok. ***$p < 0.001$; **$p < 0.01$; *$p < 0.05$.

prominent category-specific information during the encoding phase in
both VVS and PFC. In the VVS, this was observed in all frequency bands
(all $p_{corr} < 0.005$; Fig. 2D, first column). In the PFC, category-specific
information was found in the theta, beta and low-gamma frequency
bands (all $p_{corr} < 0.01$; Fig. 2D, third column). To examine the relative
timing of the effects in greater detail, we increased the temporal
resolution by shortening the sliding windows to steps of 10 ms and
including all frequencies in the 3–150 Hz range as RSA features. This
analysis demonstrated that categorical information reached the PFC
360 ms after stimulus presentation, i.e., 290 ms later than the VVS

(Fig. 2E). Indeed, a direct comparison of latencies showed significantly
higher EES in VVS than PFC starting 150 ms after stimulus onset ($p_{corr} =
0.007$). Furthermore, in the majority of frequency bands category-
specific representations were most pronounced at matching time
points across trials (diagonal values in the EES analyses; Fig. 2D) and
did not generalize to other time periods, in line with theories on
dynamic coding[5,57].

Next, we analyzed encoding-maintenance similarity during the
second maintenance period (EM₂S) to investigate whether category-
specific representations established during encoding reoccurred after

the presentation of the retro-cue. In the $EM_2S$ analysis, we observed significant reoccurrence of category-specific information in all frequency bands in the VVS (all $p_{corr} < 0.025$). These effects were transient and most pronounced within the first 2 seconds after the retro-cue (Fig. 2D, second column). In contrast, we did not observe reoccurrence of category-specific information in the PFC in any frequency band (all $p_{corr} > 0.19$; Fig. 2D, fourth column), suggesting a transformation of representational formats in this region.

In several additional control analyses, we comprehensively characterized the representational formats in VVS and PFC (Supplementary Fig 1), evaluated the presence of representations in the Maintenance 1 period ($EM_1S$ analysis; Supplementary Fig 2 and Supplementary Note 1), investigated the representation of individual exemplars during encoding and maintenance (Supplementary Fig 3 and Supplementary Note 2), and evaluated differences in performance and neural representations for items encoded in different positions during encoding (EES, $EM_2S$ and MMS analyses; Supplementary Fig 4 and Supplementary Note 3).

Together, these results show the formation of category-specific representations in both VVS and (later) in PFC during encoding, but reoccurrence of encoding patterns during maintenance in the VVS only.

## Representational formats of category-specific representations

Our findings presented thus far indicate maintenance of category-specific representations from encoding in the VVS that was not observed in the PFC. The absence of an effect in the PFC may be attributed to a transformation of VWM representations driven by the prioritization process. Indeed, recent behavioral[61], neuroimaging[53] and iEEG studies[57,58] established a crucial role of transformed representational formats, particularly abstract representational formats devoid of specific sensory information, in VWM maintenance. Based on these insights, we hypothesized that the PFC might represent stimuli in a representational format devoid of low-level sensory information that maps to deep DNN layers during the prioritization period.

To evaluate this hypothesis, we employed different deep neural network (DNN) architectures. First, we used the feedforward DNN 'AlexNet'[62] that has been extensively employed to characterize neural representations of natural images during perceptual and mnemonic processes[36,57,58,63–70]. Additionally, we applied two recurrent DNNs, the BL-NET and the corNET-RT. The BL-NET consists of seven convolutional layers which include lateral recurrent connections and has previously been applied to predict human behavior, specifically reaction times, in a perceptual task[71]. The corNET-RT has a relatively shallow architecture compared to similarly performing networks for image classification and has been designed to model information processing dynamics in the primate VVS[72]. Similar to the BL-NET, corNET-RT exhibits recurrent dynamics that propagate within (but not between) layers. All 3 DNNs represent stimuli in various representational formats, ranging from low-level visual features in superficial layers to higher-level properties in deep layers. While AlexNet processes stimulus features in a single feedforward pass, the lateral recurrent connections of BL-NET and corNET-RT generate temporally evolving time-series of stimulus representations in each layer, thus capturing core properties of recurrent dynamics during WM processing in the PFC. The number of recurrent passes is fixed to 8 time-points in BL-NET, while the corNET-RT model exhibits layer-specific recurrent passes that range from 2 to 5 time points (see Methods). Following previous studies, and in order to ensure that the networks achieved stable representations of our images in each layer, we focused on the RSMs at the last time-point of each layer[73].

We first characterized stimulus representations in different layers of AlexNet. We constructed RSMs from DNN representations by computing the similarities between all unit activations in each layer for all pairs of images (see refs. 57,58; Fig. 3A, top). For visualization, we projected the data into two-dimensional space using Multidimensional Scaling (Fig. 3A, bottom). To evaluate representational changes throughout the DNN, we correlated the RSMs between different layers. RSMs were most similar among the convolutional layers 2–5 and among the fully connected layers 6-7, while the input layer 1 and the output layer 8 exhibited the most distinct representational patterns (Fig. 3B). We computed the Category Cluster Index (CCI; see refs. 74,75), defined as the difference in average distances of stimulus pairs from the same category vs. stimulus pairs from different categories (Fig. 3C). CCI takes a value of 1 if clusters are exclusively built by stimuli from the same category and approaches 0 if the representational geometry shows no categorical organization. Using permutation statistics (i.e., label shuffling), we observed that CCI values were significantly higher than chance in all layers of the network (all $p_{corr} = 0.008$, Bonferroni corrected for the 8 layers). Notably, we observed a four-fold increase in CCI scores from the first (CCI = 0.11) to the last layer (CCI = 0.46) of the AlexNet (Fig. 3C). This effect was explained by both an increase of within-category correlations (average slope of linear fit across layers = 0.046; $p = 0$; Supplementary Fig 5, top left), and a decrease of between-category correlations across layers (average slope across layers = −0.008; $p = 0$; Supplementary Fig 5, top right).

We next set out to evaluate the similarity between stimulus representations in AlexNet and neural representations in VVS and PFC. In order to characterize the frequency profile of reactivations, we performed a frequency-resolved analysis of fits between neural and DNN representations: We constructed RSM time-series for every frequency independently and grouped them into a time-frequency map of model fits (Methods). In the VVS, we found that representational geometries during encoding were captured by network representations in all layers in the 3–75 Hz range (all $p_{corr} < 0.008$; Fig. 3D, top row). In layers 4 and 6–8, this effect extended into the high-gamma frequency range. Similar to the results observed in the category model analysis (Fig. 2B), we did not observe any matching between neural and AlexNet representations during the maintenance period (all $p_{corr} > 0.056$; Fig. 3E). In the PFC, we did not observe any significant fit during either encoding (all $p > 0.064$; Fig. 3D, bottom row) or maintenance (all $p > 0.168$; Fig. 3F).

Taken together, these results show that representations in the AlexNet are aligned with encoding representations in VVS but not PFC. Importantly, during the maintenance period neither VVS or PFC representations showed a significant fit with representations in the AlexNet network, suggesting that the format of prioritized VWM representations cannot be explained by feedforward DNNs.

We thus employed the recurrent neural networks BL-NET and corNET-RT to characterize representational formats in VVS and PFC. We first assessed the temporal evolution of network representations in the different layers of BL-NET and correlated the layer-wise RSMs between successive time points (Methods). In all layers, representations changed most prominently between intervals 1–2 and least between intervals 7–8 (Fig. 4A). In layers 2 to 7, representations remained largely constant following time step 3, while the first layer showed more substantial dynamics until the last time interval (Fig. 4A). Directly comparing the representations between the initial (1st) and the final (8th) time points separately for each layer revealed larger changes in the first two layers and substantially smaller changes in layers 3–7 (Fig. 4B, C). Similar to the AlexNet, CCI values were significantly higher than chance in all layers (all $p = 0.007$, Bonferroni corrected for 7 layers), and we observed a fourfold increase of CCI values from the first (CCI = 0.07) to the last layer (CCI = 0.40) of the network (Fig. 4D). Contrary to the AlexNet, the increases in CCI in the BL-NET network were only due to a decrease in between-category correlations (average slope of linear fit across layers = −0.06; $p = 0$; Supplementary Fig 5, middle right), while the within-category correlations did not change across layers (average slope of linear fit across

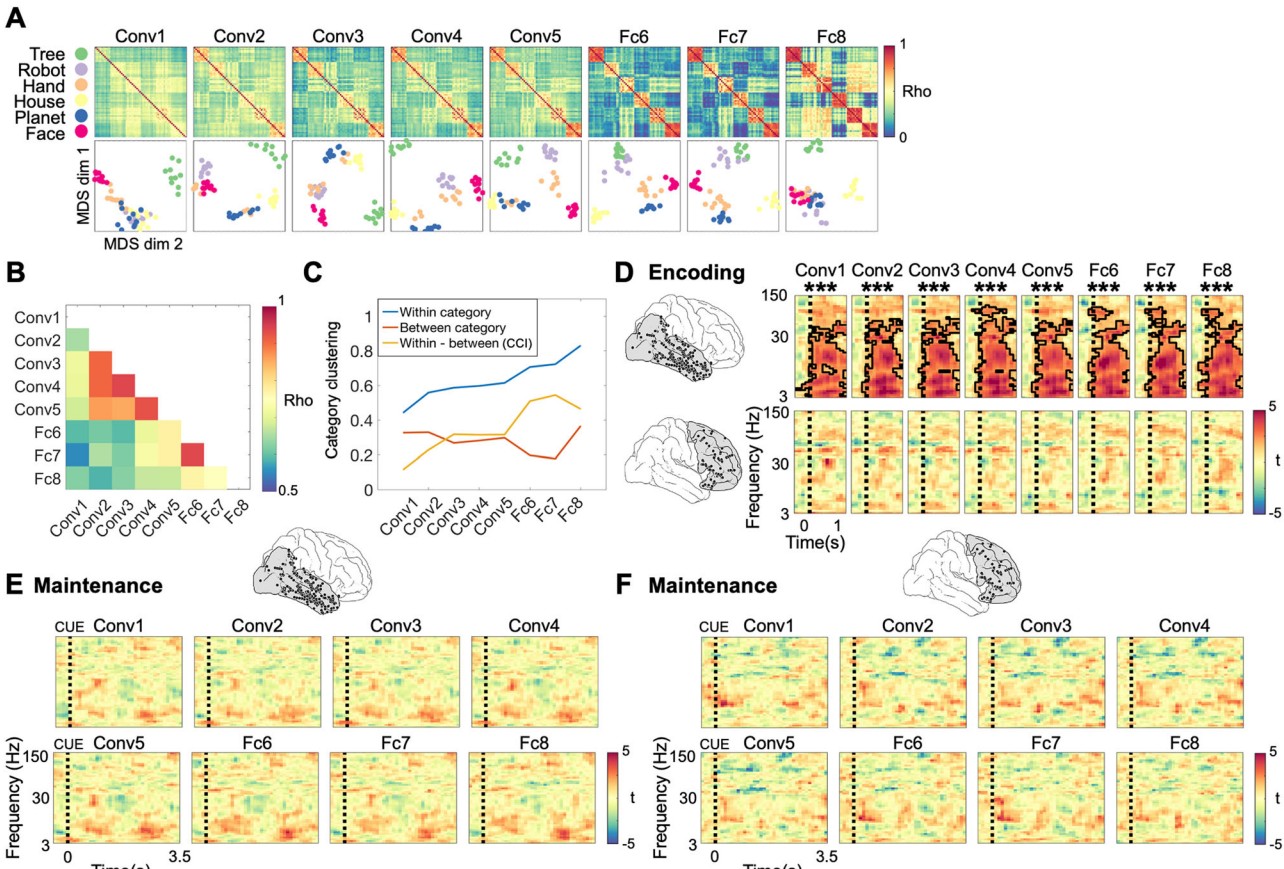

**Fig. 3 | Analysis of representational formats using a feedforward deep neural network. A** Top: Representations in the feedforward network AlexNet. Representational Similarity Matrices (RSMs) reflecting pairwise correlations of unit activations in each layer of the network. Bottom: 2D Multidimensional Scaling (MDS) projections of RSMs at each layer, color-coded according to categories.
**B** Representational consistency plot showing pairwise correlations (Spearman's rho) of RSMs at each network layer. **C** Within-category, between-category and within-category vs. between-category correlations (i.e., Category Cluster Index, CCI) as a function of network layer. **D** Top: Correlations between RSMs from the DNN and neural data, for each AlexNet layer and each encoding time-frequency window in the VVS. Each time-frequency plot shows the correlation values of representations in one particular layer to neural representations. Clusters outlined in black indicate time-frequency periods where correlation values are significantly higher than zero at the group level (two-sided *t*-tests, Bonferroni corrected for 8 layers). Bottom: Same analysis for PFC data. Time zero in all panels indicates the onset of stimulus presentation **E** No matching of VVS RSMs with AlexNet RSMs during the maintenance period. Time zero indicates the onset of the cue. **F** Same analysis as in E for the PFC data. Color scale of all t-maps in **F** and **G** is indicated at the right of each panel. Source data are provided as a Source Data file. ***$p < 0.001$.

layers = −0.00062; $p = 0.72$; Supplementary Fig 5, middle left). Furthermore, BL-NET between-category correlations decreased significantly more across layers than AlexNet between-category correlations (Alexnet vs. BL-NET slope difference = 0.1061; $p = 0$).

Together, these results show that the BL-NET network represents low-level features more dynamically than high-level visual features and that it clusters categorical information more strongly in deep than superficial layers. Contrary to the AlexNet network, this clustering is exclusively due to a reduction of between-category correlations rather than an increase in within-category correlations across network layers.

We next compared neural and BL-NET representations, focusing on the RSMs at the last time-point of each layer (Fig. 4E). During encoding, results were similar to those in the AlexNet analysis: Network representations of all layers matched VVS representations for a wide range of frequencies between 3 and 75 Hz ($p_{corr} = 0.007$; Fig. 4F), and these extended into the high-gamma range (i.e., until 110 Hz) in layer 7. No significant correlations were observed in the PFC (all $p_{corr} > 0.263$; Fig. 4G). During maintenance, no significant fits were observed in the VVS following the retro-cue, again consistent with the AlexNet analysis. Interestingly, however, we observed a significant matching of VVS representations in the theta/alpha frequency range (3–14 Hz) with BL-NET representations in layers 4 ($p_{corr} = 0.035$), 5

($p_{corr} = 0.035$) and 6 ($p_{corr} = 0.014$). These effects occurred in a late maintenance time period from 2.1 s to 3.2 s, close to the presentation of the probe (Fig. 4H, top row). Critically, in the PFC, we observed a significant fit between neural and network RSMs following presentation of the retro-cue, i.e. time-locked to the prioritization process. This effect started 200 ms after the onset of the retro-cue and lasted for 800 ms; It was specifically observed for the last layer of the BL-NET (final layer: $p_{corr} = 0.021$; all other layers: $p_{corr} > 0.43$), and related to neural representations in the beta frequency range (15–29 Hz; Fig. 4H, bottom row).

The specific alignment of the representational geometry of PFC activity with the last layer of BL-NET during the prioritization period suggests that the format of representations has been transformed in this region—from a purely categorical format during encoding into a format that incorporates distinctions among stimuli between categories during maintenance. To corroborate this transformation and characterize the representational formats observed in the PFC more comprehensively, we performed several additional analyses. First, we tested whether the average pairwise neural correlations differed between encoding and maintenance. Higher correlations of items during maintenance may point towards clustering of representations, while lower correlations would reflect the opposite, i.e.

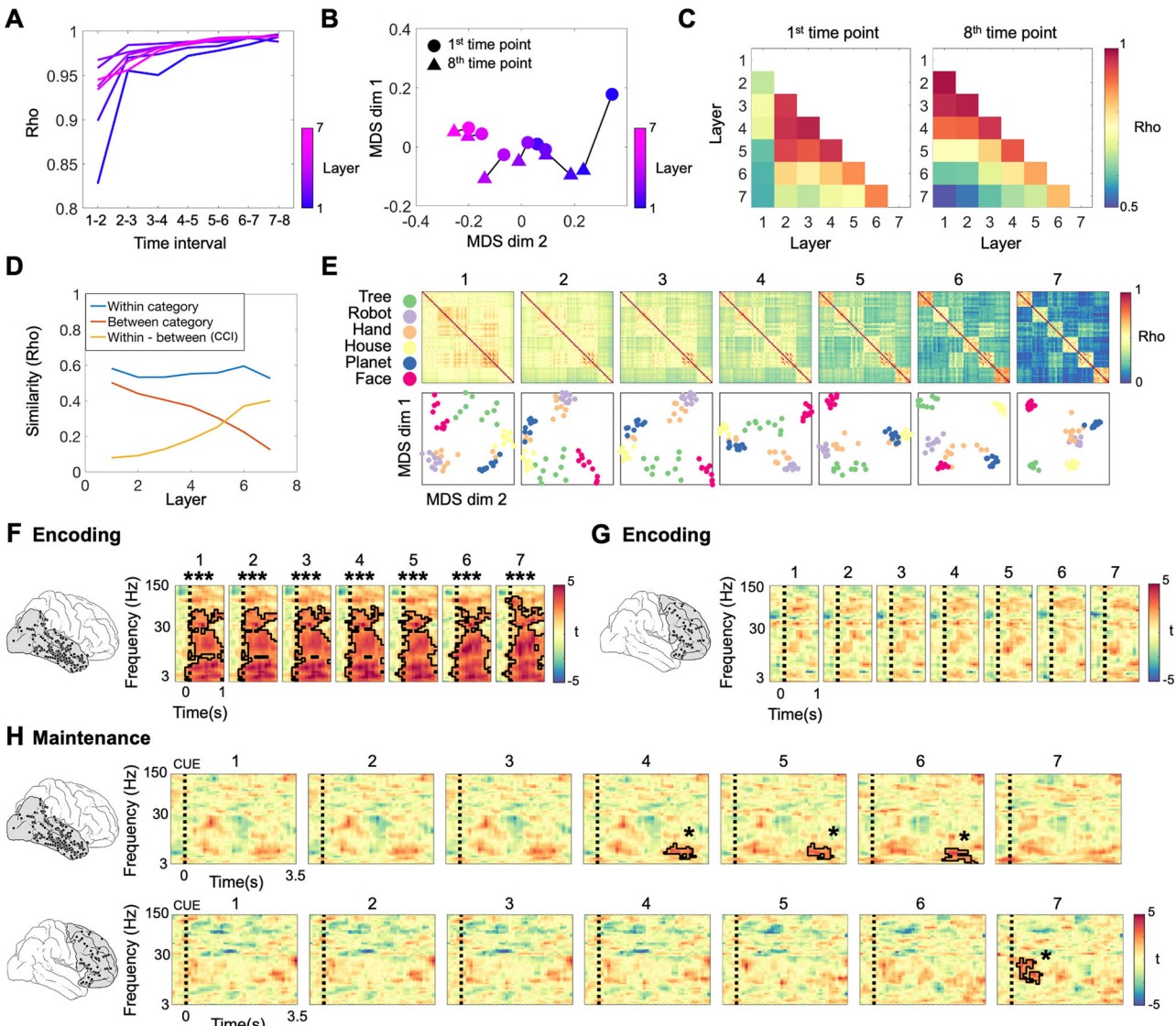

**Fig. 4 | Analysis of representational formats using the BL-NET network.**
**A** Representational consistency at each time interval of the BL-NET network was computed by correlating representations formed at successive time points. Each curve represents one layer of the network, color-coded from early (blue) to deep layers (pink). **B** Two-dimensional projections of the first and last time point of each layer in the BL-NET network showing greater representational distances in the first layer than in all other layers. **C** Pairwise correlations of RSMs corresponding to the first (left) and last (right) time points in each layer of BL-NET. **D** Within-category, between-category and within-category vs. between-category correlations (i.e., Category Cluster Index, CCI) for each layer of BL-NET (last time point). **E** RSMs and corresponding MDS projections for the last time point of all BL-NET layers. In the MDS plots, items are color-coded according to category. **F** Correlations between

BL-NET RSMs (last time point in each layer) and neural RSMs during encoding in VVS. Outlined clusters indicate time-frequency periods where correlation values are significantly higher than zero at the group level (two-sided *t*-tests, Bonferroni corrected for 7 layers). **G** Same analysis as in F for the PFC. **H** Same analysis as in **F** for the maintenance period in the VVS (top) and PFC (bottom). In the VVS, BL-NET representations in layers 4, 5, and 6 matched representations in the theta/alpha frequency range (3-14 Hz) prior to the probe. In the PFC, BL-NET representations in the last layer matched representations in the beta band (16–29 Hz) following presentation of the retro-cue. Color scale of all t-maps in **F**, **G** and **H** is indicated at the right of each panel. Source data are provided as a Source Data file.
***$p < 0.001$; *$p < 0.05$.

representations in a more widely spread representational space. Second, we analyzed the variance of correlations during encoding and prioritization. Higher variances would reflect less uniform (i.e., more distinctly organized and thus lower dimensional) distributions of items, while lower variances would correspond to an opposite pattern. We performed both analyses separately for items of the same category (within-category correlations) and items of different categories (between-category correlations). We found a trend for higher average between-category correlations during maintenance as compared to encoding (t(14) = −2.12, $p = 0.053$), and no significant differences in the average same-category correlations (t(14) = −0.185, $p = 0.856$). Moreover, the variance of between-item correlations decreased significantly

from encoding to maintenance, both for items from different categories (t(14) = 5.87, $p = 4.05e−05$) and from the same category (t(14) = 5.37, $p = 9.89e−05$). We next compared the dimensionality of RSMs during encoding and maintenance. We projected the data in various dimensions using Multidimensional Scaling (MDS), and computed the stress of the MDS projections. Stress indicates the goodness of fit of a particular projection, and thus lower stress values during encoding or maintenance would indicate lower-dimensional representations during that time period. We observed that stress values were systematically lower during encoding as compared to prioritization in a cluster of significant dimensions (from 4 to 33 dimensions; $p = 0.0148$; Supplementary Fig 1A). Taken together, these results

indicate a change from a purely categorical representation during encoding to a representation that matches the fine-grained architecture of the BL-NET during prioritization: PFC representations occur in a smaller representational space, occupy less clustered regions in this space, and rely on a higher-dimensional neural code. Thus, our results point to a transformation of the representational format of PFC activity from encoding to maintenance.

We next investigated the fit of the BL-NET and the AlexNet networks during the prioritization period separately for within-category and between-category correlations. We observed a significant fit of the between-category correlations of RSMs from the BL-NET and neural data in the PFC (t(14) = 3.69, $p$ = 6.76e−05; Supplementary Fig 1B), while this was not true for the AlexNet (t(14) = 1.61, $p$ = 0.13). None of our models could explain the structure of within-category correlations (BL-NET: t(14) = 0.42, $p$ = 0.67; AlexNet: t(14) = −0.18, $p$ = 0.86; Supplementary Fig 1B). The results of the same analysis performed during encoding confirmed that neither BL-NET nor AlexNet are good models of activity in the PFC during this time period (BL-NET within-category correlations: t(14) = −0.37, $p$ = 0.71; BL-NET between-category correlations: t(14) = −1.84, $p$ = 0.086; AlexNet within-category correlations: t(14) = 0.17, $p$ = 0.86; AlexNet between-category correlations: t(14) = −1.31, $p$ = 0.21). These results demonstrate that the fine-grained structure in PFC that is captured by the BL-NET model is due to the geometry of between-category correlations—i.e., that the BL-NET corresponds to the relative representational distances of individual exemplars to exemplars of other categories.

In additional control analyses, we investigated the functional relevance of representations in VVS and PFC during the maintenance period (Supplementary Fig 6), compared the fits of the BL-NET, Alex-Net and the category model in VVS and PFC (Supplementary Note 4; Supplementary Fig 7 and Supplementary Fig 8), dissociated the representational formats of the category model and the BL-NET through simulations and analyses conducted in individual participants (Supplementary Note 5; Supplementary Fig 8 and Supplementary Fig 9) and evaluated the fits of the BL-NET in the PFC with a variant of this network trained with a recently released dataset of images (Ecoset[76]; Supplementary Fig 10).

In our final analysis, we employed the corNET-RT model to account for VVS and PFC representations. Consistent with the BL-NET analyses, we first evaluated the representational consistency across successive time points in each layer of the network. The final layer (IT) showed the lowest correlation across consecutive time points compared to all other recurrent passes in the network (Rho = 0.78; note the first recurrent pass in IT is the fourth overall pass in the network, Methods). This demonstrates that contrary to the BL-NET, corNET-RT represents stimuli more dynamically in its deepest layer. In addition, we observed that representations in layers V2 and V4 clustered together in representational space, while representations in V1 and IT were segregated (Fig. 5B, C). Categorical clustering of representations was found in all layers, as evidenced by significant CCI scores in each layer and at each time point (all $p_{corr}$ > 0.004; Fig. 5D). Similar to BL-NET and contrary to AlexNet, we observed a prominent increase in CCI scores across layers, which was mostly due to a decrease in between-category correlations (average slope across layers = −0.14; $p$ = 0; Fig. 5D and Supplementary Fig 5). However, within-category correlations were also reduced across network layers (average slope across layers = −0.02; $p$ = 1.63e−11; Fig. 5D and Supplementary Fig 5).

We compared corNET-RT RSMs to neural representations, focusing on the last time point in each layer, again consistent with the BL-NET analysis (Fig. 5E). During encoding, we found a significant match of VVS representations across a wide range of frequencies with corNET-RT representations in all layers (3–105 Hz; all $p_{corr}$ < 0.004; Fig. 5F, top row). No significant correlations were found in the PFC (all $p_{corr}$ > 0.053, Fig. 5F, bottom row). During the maintenance phase, corNET-RT representations in IT matched those in the VVS towards the

end of the maintenance period, specifically in the theta-alpha frequency range, consistent with the results observed in the BL-NET analysis (6–11 Hz; $p_{corr}$ = 0.044; Fig. 5G, top row). Critically, we again observed a significant match of corNET-RT representations in IT with PFC representations time-locked to the presentation of the retro-cue and in the beta band (15–29 Hz; $p_{corr}$ = 0.016; Fig. 5G, bottom row). This effect lasted for 500 ms, similar to the results observed in the BL-NET analysis. In addition, we observed a significant correlation with representations in V1 ($p_{corr}$ = 0.036).

We performed control analyses using parameter-matched versions of our recurrent architectures to evaluate the effect of recurrency, while isolating other possible confounding variables (Supplementary Note 6). Results suggested that recurrent computations are indeed crucial for tracking cognitive representations in PFC, because the fit observed with the recurrent networks could not be found in any of the feedforward models we tested. They also show that recurrency may play a relatively less prominent role in the VVS (Supplementary Note 6).

Taken together, these results show that PFC representations following the retro-cue matched those in two recurrent neural network architectures (the BL-NET and the corNET-RT) but not those of a purely feedforward network (the Alexnet), and that these effects were specific to the beta-frequency range and most prominent for late layers of the networks. VVS representations did not show correspondence with representations in recurrent networks following the retro-cue, but prior to the probe.

## Discussion

Our study aimed to unravel representational formats and neural coding schemes in sensory and executive control regions during WM prioritization. Specifically, we analyzed the impact of WM prioritization on stimulus-specific activity patterns in VVS and PFC and assessed their representational formats using feedforward and recurrent DNN models of natural image processing. The VVS exhibited pronounced category-specific representations during encoding which were reinstated during the maintenance period, reflecting a shared (or 'mnemonic') coding scheme across both experimental phases. The PFC exhibited robust category-specific representations during WM encoding as well, but did not show reinstatement of encoding patterns during the maintenance period. Subsequent in-depth analyses showed that this lack of reinstatement in PFC was not due to memory decay or reduced signal to noise ratio, but due to a transformation of representations between different task-dependent formats, in line with a dynamic 'prioritization' coding scheme: Representations in PFC corresponded to a simple categorical model during encoding, but matched only the deepest layer of a recurrent DNN following retro-cue, suggesting a prioritized format in which high-level visual features of images are preponderant. This shift was also reflected at the level of the neurophysiological substrates of WM representations, since PFC representations during encoding were observed in theta, beta and gamma frequency bands but exclusively in beta frequency oscillations during the retro-cue. Taken together, these results demonstrate that WM prioritization relies on a distinct recruitment of specific task-depend representational formats in the PFC.

Recent investigations showed a transformation of visual representations from perceptual to abstract formats during VWM encoding[57,58]. While representations in these studies were based on patterns across the entire brain, we here focused on representations in two brain regions that are critical for VWM storage and control, respectively: VVS and PFC. We note that our initial RSA analysis of category representations (Fig. 2B) could not explain representations during the maintenance period in either of these regions. The EMS analysis (Fig. 2D), however, revealed a distinct set of results in VVS and PFC: While encoding activity patterns reoccurred during the maintenance period in the VVS, this was not the case in the PFC. In the VVS,

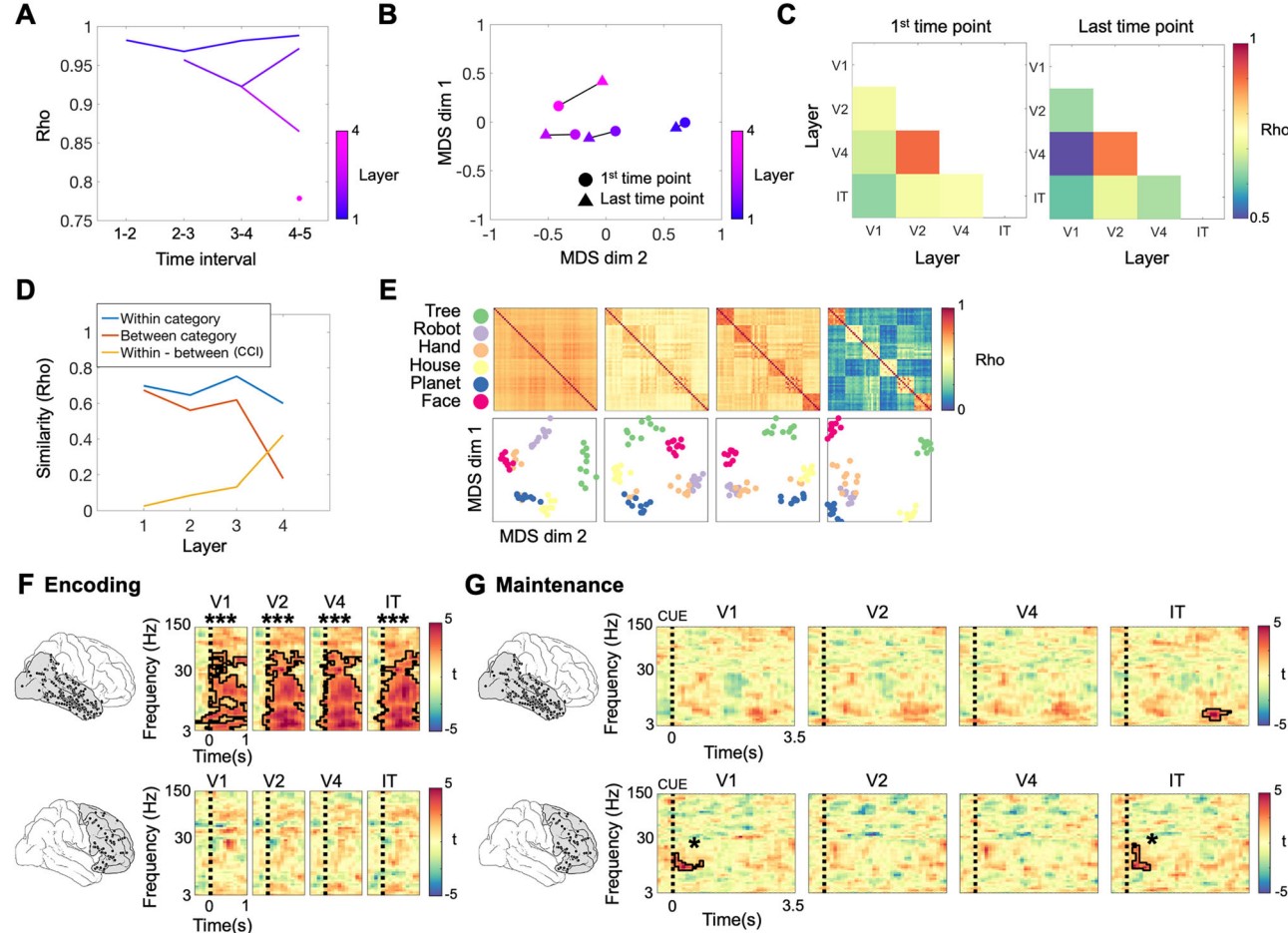

**Fig. 5 | Analysis of representational formats using the corNET-RT network.**
**A** Representational consistency at each time interval of the corNET-RT network. Note that in this architecture, different layers have different numbers of recurrent passes, and deep layers do not receive input until activity has propagated from early layers. Each curve represents one layer, color-coded from early (blue) to deep (pink). **B** MDS projections of first (circle) and last (triangle) time point in each layer show relatively higher temporal dynamics in layer IT (output) compared to the other layers. MDS results have been scaled for visualization. **C** Pairwise correlations of RSMs corresponding to the first (left) and last (right) time points in each layer of corNET-RT. **D** Within-category, between-category and within-category vs. between-category correlations (i.e., Category Cluster Index, CCI) at each layer of the network (final time point). **E** RSMs (top) and corresponding MDS projections (bottom) for each layer of the corNET-RT network (final time-point). Items are color-coded by category. **F** Correlations between corNET-RT RSMs (last time point in each layer) and neural RSMs during encoding in VVS (top) in the 3–150 Hz frequency range. Colors indicate resulting t-maps in the comparison of group-level correlation values against zero (two-sided $t$-tests). Significant regions after multiple comparisons correction are outlined in black (Bonferroni corrected for 4 layers). **G** Same analysis as in **F** for the maintenance period. Top: A match between network and neural representations was observed in the VVS in a late period, close to the presentation of the probe, in the IT layer. Bottom: In the PFC, correlations were significant with representations in the beta frequency range (15–29 Hz) following the onset of the retro-cue with both V1 and IT layer. Color scale of all t-maps in **F** and **G** is indicated at the right of each panel. Source data are provided as a Source Data file. ***$p < 0.001$; *$p < 0.05$.

representations towards the end of the maintenance period matched representations in intermediate and deep layers of two recurrent DNN architectures (BL-NET and corNET-RT), suggesting a transformation during encoding from a purely 'categorical' format into a format that incorporates high-level visual and semantic relationships among stimuli. Thus, despite a relative stability of neural activity patterns (as revealed by the EMS analysis), their representational geometry changes and eventually results in less categorical representations during the maintenance period.

By contrast, in the PFC, encoding activity patterns did not reoccur during the maintenance period, suggesting a more pronounced transformation in this region; however, neural representations following the retro-cue matched representations in deep layers of two recurrent DNN architectures. Notably, in the VVS, all representational signatures that were observed during the second maintenance period (corresponding to the deeper layers of the two recurrent networks) were already apparent during encoding, and this likely explains the

significant encoding-maintenance similarity (EM$_2$S) in this region. Thus, maintenance in VVS corresponds to a partial and selective reappearance of encoding formats, corresponding to a 'mnemonic' coding scheme. By contrast, in the PFC, the representational signatures that were observed during maintenance did not already occur during encoding, and thus the PFC does not show such a mnemonic coding scheme but exhibits a more profound transformation. We refer to the format of PFC representations after the retro-cue as 'prioritized'.

Our results provide a comprehensive description of the representational transformation observed in the PFC during the prioritization period, from a purely categorical to a less categorical and higher-dimensional format that specifically maps with the BL-NET and corNET-RT but not with other DNN models. In detailed analyses of the geometry of PFC representations during encoding and maintenance, we found that PFC representations occur in a smaller representational space during the prioritization period, occupy less clustered regions, and rely on a higher-dimensional neural code. Notably, these

differences were mostly observed for the between-category correlations (Supplementary Fig 1), whose structure cannot be explained by the category model. Considered together with the lack of EM$_2$S in PFC, these results point to a transformation of the representational format of PFC activity from encoding to maintenance, which is particularly due to a transformation of the geometry of the between-category correlations.

An important difference between VVS and PFC relates to the time period at which representations matched those from a recurrent DNN: directly following the retro-cue in PFC, but prior to the probe in the VVS and thus in preparation for the response. This suggests that maintenance in the two different regions likely serves different functional roles.

What could be the functional role of the transformation of category-specific representations in the PFC? Our data is consistent with a capacity-limited view of WM which would benefit from compressed stimulus representations in this region, while still maintaining high-level visual properties of images. This notion has been recently supported by behavioral[61] and neuroimaging[53] studies. In particular, ref. 61. demonstrated that semantic aspects of images are selectively prioritized during WM maintenance in a multi-item WM task, while no selective storage of abstract features of images is present in single-item tests. This fits to our findings in the VVS that contained both perceptually detailed and abstract representations during maintenance of single items, while we additionally found high-level visual representations in the PFC. In the fMRI study of ref. 53, representational abstraction was observed in parietal and visual cortices, but not in prefrontal regions. The differences between our results and those of ref. 53. might relate to the particular stimuli employed (natural images with semantic content versus low-level visual features), and to our use of a paradigm involving prioritization, which preferentially engages the PFC[1,23].

Prioritized information in PFC was specifically detected in the 15–29 Hz frequency range, i.e., within the beta band (13–29 Hz). Previous studies showed a prominent role of prefrontal beta oscillations for top-down control of information in WM[4,35], and oscillatory activity in the beta range has also been associated with transient task-dependent activation of stimulus-specific information during WM maintenance[46]. Our results in PFC are well consistent with these interpretations: The representations we observed are content-specific and locked to the presentation of the retro-cue, which is when the prioritization process takes place. This aligns with previous studies that have reported stimulus-specific activity in the beta range during WM maintenance (e.g., ref. 45; see ref. 46. for review). In contrast to standard delay tasks where beta modulations occur late in the WM delay period[43,45], our study demonstrates brief and cue-locked effects, consistent with previous retro-cue paradigms[77]. Our findings are also consistent with the widely accepted role of the PFC in the top-down control of information stored in other brain regions, in line with previous studies on both episodic and working memory[78]. Indeed, activity in the PFC has been linked to task-dependent executive control over specific contents in several studies (for a review, see ref. 79). This could be achieved by modulating the activation state of distributed perceptual and mnemonic representations[78], for instance through PFC connectivity with the VVS[80]. The transient beta-frequency reactivation we observed in the PFC is suggestive of a top-down signal prompted by presentation of the cue that might affect information processing in downstream regions[81]. Further studies are required to investigate this possibility. Nevertheless, our results confirm previously untested views of PFC functioning by demonstrating its engagement in the transformation of VWM representations during VWM prioritization.

DNNs are increasingly used in cognitive neuroscience to characterize the representational formats and temporal dynamics of perceptual and mnemonic representations in the brain. While different feedforward and recurrent architectures have been applied in the domain of vision, resulting in a wide variety of models employed to fit neural data (e.g., refs. 32,33,59,65,67,82), this approach has only started to be employed in memory research. Pioneering investigations have applied the feedforward neural network AlexNet to study representational formats during visual working memory in humans[57,58]. Notably, these studies did not investigate the representational formats during WM maintenance but focused solely on the encoding period. While theoretical and experimental considerations have strongly argued for the use of recurrent architectures in the domain of visual perception[33,60,83], they have so far not been applied to memory research. The use of recurrent architectures in the context of working memory is particularly important given the relevance of recurrent computations for PFC processing[26,84] and WM functions in general[27,28,30,85]. In our study, we tested a feedforward and two recurrent models in their ability to predict representational distances in human iEEG data. During encoding, both types of models captured the representational geometry of stimuli across all layers and a wide range of frequencies in the VVS, while no fits were observed in the PFC (Figs. 3, 4 and 5). During maintenance, however, the two architectural families strongly differed in their fit to the neural data: the AlexNet was unable to capture representations in either region, while BL-NET and corNET-RT matched representations in both VVS and PFC (Figs. 4H and 5G). Control analyses using parameter-matched versions of the BL-NET without recurrency indicated that recurrent computations are indeed crucial for tracking cognitive representations in PFC, while they appear to play a relatively less critical role in the VVS (Supplementary Note 6). Together, these results demonstrate that only recurrent architectures can explain the representational geometry of stimuli during VWM prioritization in PFC, while a feedforward architecture and a simple model of category information do not provide good fits.

What are the differences in the representational geometries of AlexNet, BL-NET and corNET-RT that can explain the different fits to the neural data we observed? We thoroughly characterized within-category, between-category and within- vs. between-category correlations (i.e., CCI) in all three architectures to investigate their differences in stimuli representation. We found that all networks represented increasingly category-specific information across layers, as assessed by prominent increases in CCI, yet this was achieved through different representational changes. While the AlexNet showed both an increase in within-category correlations and a decrease in between-category correlations, the recurrent models only showed decreases in between-category correlations (Supplementary Fig 5), suggesting that recurrence particularly supports distinct representations of different categories. Again, further studies are needed to unravel the possible neurophysiological basis and cognitive function of these representational transformations.

Computational models of WM have proposed that prioritization requires a transformation of the neural space of activity in which the items are represented, involving, for example, a rotation or "flip" of the format of prioritized content in neural activity space[29,30]. These models have recently received empirical support from studies in monkeys (e.g., ref. 86), suggesting an efficient neural code that organizes and structures neural representations during the prioritization process. Consistently, a recent iEEG study in humans demonstrated a role of PFC in resolving cognitive interference between competing sensory features by transforming their representational population geometry into distinct neural subspaces to accommodate flexible task-switching[87]. Our work contributes to this literature by establishing that the PFC not only supports a transformation of the representational geometry of stimuli but also a differential representation of particular visual formats in the context of VWM prioritization. In particular, we argue that the degree of matching to RSMs derived from DNNs is of heuristic value because these models have previously been

shown to match representations during sensory processing and have been widely applied to analyze representational transformations during various cognitive tasks (see ref. 83).

It has been recently proposed that the selection of network training sets critically influence the matching of DNN and neural representations, and that this influence may be more important than specific architectural constraints[88]. For this reason, in our study we employed two different datasets of images (ImageNet and Ecoset), which provided consistent results (Supplementary Fig 10). Other limitations remain, however: First, the BL-NET and corNET-RT networks were not trained to memorize stimuli but to solve the task of image classification, which may be argued to limit their value as models of WM representation. However, we note that the use of networks trained in a lower dimensional task objective, i.e., image classification, to model cognitive representations embedded in a higher-level cognitive process, i.e., VWM prioritization, has received some theoretical support. Indeed, representational accounts of memory have argued that it is not the cognitive process (e.g., memory versus perception) that defines representations, but rather the content that any given cognitive process requires. Indeed, regions representing particular content in the brain (e.g., low-level visual features in early visual regions) are involved in the representation of these features irrespective of the cognitive process in which they are engaged[89–91]. Since the VVS plays a role both during object recognition and VWM for these objects, it is relevant to investigate the representational format of items during both processes, and DNNs are arguably strong tools to capture these formats[58,92]. Beyond these theoretical considerations, we underscore the widespread practice in our field of using networks pretrained in particular tasks to characterize representations formed in different tasks. Previous studies have employed the AlexNet network to investigate the representational formats of representations during both VWM and long-term memory[57,58,92]. A similar trend is observed in natural language processing, where language models trained in the task of next word prediction have been applied to model language-related brain responses more broadly[93–97].

A second limitation of the models we employed relates to their architecture: BL-NET and corNET-RT do not include top-down connections but only lateral connectivity, and thus cannot account for PFC-VVS interactions. In future work, novel architectures should be employed that mimic brain connectivity more accurately at least at a high-level of description (i.e., containing top-down as well as within-layer connections). Finally, while we decided to focus on the prioritization process and the single-item trials in this study, we aim to further investigate the representation of multiple items in the future. A promising avenue for this purpose is the use of sequential recurrent convolutional networks that receive multiple consecutive images as input and can be employed to track multi-item representations (e.g., ref. 98).

We note that the different models we employed (e.g., BL-NET, AlexNet, category model) do not only represent different hypotheses about how the brain represents visual information, but they also differ in the aspects of the representational geometry they can model. For instance, the category model only codes binary information about category membership, while the DNNs' deep layers in addition reflect more subtle differences among stimuli which encode high-level visual properties of images. The category model is by definition agnostic to any structure in the within-category and between-category correlations (which are all modelled identically, with ones and zeros), while the DNN models propose a very specific geometry for these two types of relationships. Thus, fitting the two models to neural data provides complementary information regarding the geometry of representations. Notably, while the category model and BL-NET are not mutually exclusive (orthogonal), we have shown a dissociation in their levels of fit during encoding and prioritization: The category model explains well representations during encoding but not maintenance, while the reverse is true for the recurrent DNNs.

Many important previous studies on representational transformations during VWM prioritization have been conducted with non-human primates (e.g., refs. 6,86). Our study is the first report on prioritized representations using human intracranial EEG, which provides a level of analysis ideally suited to bridge network level (EEG/MEG) studies on VWM[99] to invasive recordings in monkey studies. In addition, while previous studies have employed analyses on representational subspaces based on single unit data (e.g., ref. 86) or computer simulations[29,30], we employ DNNs and RSA. While both methods have their complementary value and importance, a critical difference is the mapping of DNN onto different processing stages during perception, which adds heuristic value to our findings[83].

In summary, we present evidence of successive representational transformations during VWM encoding and after item prioritization in the VVS and the PFC that critically depend on recurrent computations and abstract representational formats. This result shows that percepts originally formed during encoding are differentially abstracted and reshaped in VVS and PFC to enable flexible task-dependent manipulations during working memory prioritization.

## Methods
### Participants
Thirty-two patients (17 females, 30 ± 10.04 years) with medically intractable epilepsy participated in the study. Data were collected at the Freiburg Epilepsy Center, Freiburg, Germany; the Epilepsy center, Second Affiliated Hospital, School of Medicine, Zhejiang University, Hangzhou, China; and the Center of Epileptology, Xuanwu Hospital, Capital Medical University, Beijing, China. The study was conducted according to the latest version of the Declaration of Helsinki and approved by the ethical committee at the Albert-Ludwigs-Universität Freiburg. All patients provided written informed consent. The number of patients included in the study was determined based on previous literature and is substantially higher than previous iEEG studies on VWM.

### Experimental design
Participants performed a multi-item working memory paradigm involving a retro-cue. They encoded a sequence of 3 images of natural objects from different categories and were asked to remember this content during a subsequent maintenance period. This period consisted of two phases that were separated by a retro-cue. The retro-cue prompted participants to selectively maintain items from particular encoding positions (single-item trials, 50%), or to maintain all items in their order of presentation (multi-item trials, 50%) for a subsequent memory test. Note that with the exception of the behavioral data, we only focused on the single-item trials in this study. In the test, six items were presented, which included all 3 presented items from encoding, and three new exemplars. Of these three new exemplars, one was always from a different category. The other two were either both from categories presented during encoding (50% of trials), or only one of them (50% of the trials, Fig. 1A). In the single-item trials, one of the lure items in the test was from the same category as the cued item in 40% of the trials. This was done to disable inferences from the probe items to the categories of the presented items. Objects pertained to six categories (trees, robots, hands, houses, planets and faces) with ten exemplars each (60 images in total). In order to perform the task correctly, participants needed to remember not only categorical information about the items but also the specific perceptual information identifying each individual exemplar.

Performance in the task was quantified separately for each encoding position. We calculated the proportion of correct responses for positions 1, 2 and 3 independently and averaged these values to

obtain an overall metric of performance in the single and multi-item trials (Fig. 1C). The task was divided into blocks and sessions. Each block consisted of 60 trials. Each session consisted of at least one block, but most participants performed between 1 and 3 blocks in each session (2.59 ± 1.04) and between 1 and 2 sessions (1.19 ± 0.39) in total. The order and frequency of image presentations was pseudorandomized to balance repetitions of images across blocks and sessions. The experiment was programmed in Presentation (Neurobehavioral systems, California, USA), and was deployed on Samsung 12" tablet computers running Microsoft Windows. Patients performed the experiment while sitting in their hospital beds and responded to the memory test utilizing the touch-screen of the tablet.

Two versions of the experiment were implemented for the different patient populations in Germany and China. The two versions had identical stimuli in all categories except for the category "Faces". The German version of the experiment included faces of former German chancellor Angela Merkel and the Chinese version included faces of the actor Jackie Chan. This was made to ensure that the face represented was equally well known to the different patient populations.

## Intracranial EEG recordings

IEEG data were recorded using amplifiers from Compumedics (Compumedics, Abbotsford, Victoria, Australia), and Brain Products GmbH with sampling rates of 2000 Hz and 2500 Hz, respectively. Patients were surgically implanted with intracranial depth electrodes for seizure monitoring and potential subsequent surgical resection. The exact electrode numbers and implantation locations varied across patients and were determined by clinical needs. Online recording data was referenced to a common scalp reference contact which was simultaneously recorded with the depth electrodes. Data was downsampled to 1000 Hz and bipolarized by subtracting the activity of one contact point with that from the nearest contact of the same electrode, resulting in a total of N-1 virtual channels for an electrode with N channels after bipolarization.

## Channel localization

Electrodes employed were standard depth electrodes (Ad-Tech Medical Instrument Corporation, Winsconsin, USA). Electrodes contained variable number of contacts and inter-contact distances. In the data collected at Zhejiang University, Hangzhou, and Medical University, Beijing, each depth electrode was 0.8 mm in diameter and had either 8, 12 or 16 contacts (channels) that were 1.5 cm apart, with a contact length of 2 mm. Channel locations were identified by coregistering the post-implantation computed tomography (CT) images to the pre-implantation Magnetic Resonance Images (MRIs) acquired for each patient, which were afterwards normalized to Montreal Neurological Institute (MNI) space using Statistical Parametric Mapping (SPM; https://www.fil.ion.ucl.ac.uk/spm/). We then determined the location of all electrode channels in MNI space using PyLocator (http://pylocator.thorstenkranz.de/), 3DSlicer (https://www.slicer.org) and FreeSurfer (http://surfer.nmr.mgh.harvard.edu). In a group of patients (data collected in Beijing), we determined MNI coordinates using the pipeline described in ref. 100, and identified the closest cortical or subcortical label for each channel in each patient. In all patients, we removed channels located in white matter, resulting in 588 clean channels across all patients (18.4 ± 11.9 channels per patient).

## ROI selection

We selected two main regions of interest given their well-known involvement in VWM: the ventral visual stream (VVS) and the prefrontal cortex (PFC). The VVS has been widely studied in the context of object recognition during visual perception[101]. Previous work employing iEEG and Deep Neural Networks often applied RSA metrics to activity from distributed electrodes across the whole brain (e.g.[57,]),

and we specifically aimed to extend these studies by investigating region-specific representations in the context of VWM (see also ref. 80).

The role of the PFC in working memory has been linked to executive control processes that enable the task-dependent manipulation and transformation of information[1,4,102]. However, relatively little is known about the representational formats of VWM representations in this region during prioritization. Moreover, no previous study investigated region-specific representational similarity during VWM.

Electrodes located at the following freesurfer locations were labeled as VVS electrodes: 'inferior temporal', 'middle temporal', 'superior temporal', 'bankssts', 'fusiform', 'cuneus', 'entorhinal'. Electrodes with the following labels were categorized as PFC electrodes: 'medial orbitofrontal', 'pars triangularis', 'superior frontal', 'lateral orbitofrontal', 'pars opercularis', 'rostral anterior cingulate', 'rostral middle frontal', 'superior frontal'. Electrodes from both left and right hemispheres were included in our ROIs. This resulted in a total number of 147 electrodes (16 subjects) in PFC and 441 in VVS (28 Subjects). The different number of subjects and channels in our two ROIs implies different levels of statistical power. Since an important objective of our study was to characterize how representations in PFC and VVS differ during VWM maintenance and prioritization, we confirmed that our main findings replicate when matching statistical power in VVS and PFC through several control analyses (Supplementary Fig 11 and Supplementary Note 7).

In additional analyses, we specifically analyzed activity in the lateral prefrontal cortex (LPFC), a brain region that has been associated with attentional prioritization[6], the representation of rules[103] and categories[104] in non-human primates. We excluded all PFC electrodes with MNI x-coordinates smaller than −35 or larger than +35 and z-coordinate < −15. The new selection resulted in a group of 9 subjects with a total number of 38 electrodes in the LPFC, which were located in the following Freesurfer regions: 'rostral middlefrontal', 'pars triangularis', 'caudal middlefrontal', 'pars orbitalis' and 'pars opercularis' (Supplementary Fig 12).

## Preprocessing

We visually inspected raw traces from all channels in each subject independently and removed noisy segments without any knowledge about the experimental events/conditions. All channels located within the epileptic seizure onset zone or severely contaminated by epileptiform activity were removed from further analyses. We divided the data into 9-second epochs (from −2 to 7 s) around the presentation of each stimulus at encoding or the onset of the retro-cue during the maintenance period. After epoching the data, we completely removed epochs containing artifacts that were identified and marked in the non-epoched (continuous) data. We visually plotted spectrograms to verify the presence of artifacts in the frequency domain in the resulting epochs. The number of epochs corresponding to item or cue presentation that were removed varied depending on the quality of the signal in each subject (15.10 ± 13.14 in each session, corresponding to around 6.9% of all epochs in each session).

Preprocessing was performed on the entire raw data using EEGLAB[105], and included high-pass filtering at a frequency of 0.1 Hz and low-pass filtering at a frequency of 200 Hz. We also applied a band-stop (notch) filter with frequencies of 49–51 Hz, 99–101 Hz, and 149–151 Hz.

## Time-frequency analysis

Using the FieldTrip toolbox[106], we decomposed the signal using complex Morlet wavelets with a variable number of cycles, i.e., linearly increasing in 29 steps between 3 cycles (at 3 Hz) and 6 cycles (at 29 Hz) for the low-frequency range, and in 25 steps from 6 cycles (at 30 Hz) to 12 cycles (at 150 Hz) for the high-frequency range. These time-

frequency decomposition parameters were taken following previous research that used iEEG oscillatory power as features for RSA[49,57,107]. The resulting time-series of frequency-specific power values were then z-scored by taking as a reference the mean activity across all trials within an individual session[108]. This type of normalization was applied to remove any common feature of the signal unrelated to the encoding of stimulus-specific information. We z-scored across trials in individual sessions in our final analyses, but similar results were obtained when we z-scored the data by considering the activity of all trials irrespective of the session. We employed the resulting time-frequency data to build representational feature vectors in our pattern similarity analyses (see below).

## Pattern similarity analysis: representational patterns

We employed different representational features in our analyses involving model RSMs (i.e., the category model RSA analysis and the DNN-based RSA analyses), and our analyses involving particular contrasts (i.e., encoding-encoding similarity analysis [EES] and the encoding-maintenance similarity analysis [EMS]; see below). In both types of analyses, representational feature vectors were defined by specifying a 500 ms time window in which we included the time courses of frequency-specific power values in time-steps of 100 ms (5 time points) across all contacts in the respective ROI (VVS or PFC). In the RSM based analyses, we performed this analysis separately for each individual frequency in the 3–150 Hz range, while in the EES and EMS analyses, we analyzed activity patterns across individual frequencies within five different bands (theta, 3–8 Hz; alpha, 9–12 Hz; beta, 13–29 Hz; low-gamma, 30–75 Hz, high-gamma, 75–150 Hz). In the RSM-based analyses, a frequency-specific representational pattern was thus composed of activity of N electrodes x 5 time-points in each 500 ms window. In the EES and EMS analyses, this representational feature vector consisted of *N* electrodes x *M* frequencies x 5 time-points. Note that the number of channels included in the representational feature vectors varied depending on the number of electrodes available for a particular subject/ROI, and the number of frequencies included in each band in the EES and EMS analyses varied as well (theta: 6 frequencies; alpha: 4 frequencies; beta: 17 frequencies; low-gamma: 9 frequencies, high-gamma: 16 frequencies; see section *Time-frequency decomposition* above). These two- or three-dimensional arrays were concatenated into 1D vectors for similarity comparisons. Only subjects with at least 2 electrodes in a particular ROI were included in all RSA analyses, leading to 15 subjects in the PFC and 26 subjects in the VVS.

## Model-based RSA

We employed temporally resolved Representational Similarity Analysis (RSA) to evaluate the dynamics of categorical information in our data following previous work[33,101]. A main assumption of this research is that stimuli from the same categories will have greater neural similarity than stimuli from different categories. To evaluate this hypothesis, we constructed a representational similarity matrix (RSM) in which a value of 1 was assigned to pairs of items of the same category and a value of zero to items of different categories ('category model', Fig. 2A). We also built an 'item model' to track the presence of item-specific information, in which correlations of items of the same category were coded with a 1 and correlations of different items were coded with a zero (Supplementary Fig 3A). Finally, we used RSMs extracted from layers of DNNs as models of representation (see section *Stimulus representations in DNNs* below).

The different model RSM were correlated with time series of neural RSMs in each of our ROIs. Pairwise correlations among stimuli were computed in windows of 500 ms, overlapping by 400 ms, using the representational patterns described in the section above, resulting in an RSA time-frequency map in each of our ROIs. In order to obtain a robust estimate of the multivariate patterns representing individual items in the category model analysis, we averaged the time-frequency

activity across repetitions of items throughout the experiment in each channel independently before building the neural RSMs (note that this was not done in the item model analysis where repeated presentations of exemplars were required). RSM time-series were vectorized by removing the diagonal values and taking only half of the matrix given its symmetry at each time-frequency point. We correlated vectorized model RSMs and neural RSMs at each time-frequency point using Spearman's rho, and evaluated whether the resulting Fisher z-transformed rho-values were different from zero at the group level to determine statistical significance (two-sided tests). Multiple comparisons correction was performed using cluster-based permutation statistics (see below), and—in the DNN analyses—, we Bonferroni corrected the final results to account for the number of layers tested in each network.

## Contrast-based RSA

In order to test the reoccurrence and stability of activity patterns in our two regions of interest during encoding and between encoding and maintenance, we performed two contrast-based pattern similarity analyses, as a complementary analysis to the model-based RSA approach (Fig. 2). In particular, we investigated the presence of category-specific information in our data by contrasting correlations between different items of the same category with correlations between different items from different categories. This was done separately for items presented in different trials during encoding (encoding-encoding similarity, EES) and between encoding and maintenance (encoding-maintenance similarity, EMS). Only items belonging to different trials were included in this analysis to avoid any spurious correlations driven by the autocorrelation of the signal. Similar to the model-based RSA approach, we averaged across item repetitions before conducting the similarity comparisons.

We computed similarities for same-category and different-category item pairs and averaged across all combinations of items in the same condition in each subject independently (rho values were Fisher z-transformed before averaging). The same-category and different-category correlations were then statistically compared at the group level using *t*-tests. In the different-category condition, we excluded item pairs containing stimuli presented in overlapping trials after averaging, again to avoid any possible bias related to the autocorrelation of the signal. As an example, if a Robot exemplar was presented in trials 2, 4 and 8, and a planet was presented in trials 7, 9 and 8, the average correlation corresponding to these items would contain activity of an overlapping trial (8 in the example). The correlation corresponding to these two items would therefore not be included. Note that this was not necessary for the same category correlations.

We quantified the similarity of neural representations by comparing epochs of brain activity separately in VVS and PFC. Note that contrary to the model-based RSA approach (see above) this analysis was not performed at each individual frequency but frequencies were grouped into five frequency bands. This effectively increased the information content (variance) of our representational patterns, making them more suitable to investigate their reoccurrence during encoding and maintenance. Moreover, combining individual frequencies into bands allowed us to reduce the dimensionality of the results when comparing all pairwise combinations of time points in the temporal generalization analysis. We computed the correlation of these representational patterns across all available time-points using a sliding time window approach proceeding in time steps of 100 ms (i.e., with an 80% overlap). This resulted in a temporal generalization matrix with two temporal dimensions on the vertical and horizontal axes (Fig. 2D). Note that values in these matrices reflect both lagged (off-diagonal) and non-lagged (on-diagonal) correlations and were thus informative about the stability of neural representations over time[5].

Pattern similarity maps were computed for each pair of items in a correspondent condition at each time-window and rho values in these maps were Fisher z-transformed for statistical analysis. The temporal generalization maps were averaged across conditions for each subject independently, and the resulting average maps were contrasted via paired t-tests across conditions at the group level.

Please note that in all pattern similarity plots (and also in the DNN-RSA plots, see below), correlations corresponding to each 500 ms window were assigned to the time point at the center of the respective window (e.g., a time bin corresponding to activity from 0 to 500 ms was assigned to 250 ms).

Please note that while the contrast-based and the model-based RSA analyses have been employed as complementary approaches to investigate neural representations[109], they differ in two important aspects. The first distinction relates to the level at which the two methods assess similarities in the representations. While the model-based analysis captures differences in the representational geometry of stimuli (it correlates RSMs of neural data with RSMs of models, a second level analysis), the contrast-based analysis directly correlates neural patterns and is thus sensitive to reoccurrence and transformation of specific neural features. For example, the same representational distances (and thus RSMs) may depend on one particular brain region (i.e., set of electrodes) during encoding and a different brain region during maintenance, leading to significant RSM-based similarities in the absence of encoding-maintenance similarity (EMS). The model-based analyses, on the other hand, correlates representational distances during either encoding or maintenance with distances in particular RSM models, and does not directly compare levels of model fits between encoding and maintenance. Thus, in a strict sense, this approach does not directly test the reoccurrence of the representational geometry, but whether a particular geometry is present during a specific time period. To test for the reoccurrence of a particular representational geometry in the model-based analyses, we directly contrasted the different levels of fit during particular time periods using paired t-tests. A second difference relates to the specific neural features that were included in each analysis. The RSM-based analysis was conducted separately for each individual frequency in the 3–150 Hz range, which allowed for a fine-grained assessment of the contribution of individual frequencies. By contrast, feature vectors in the EMS analyses included power values across several individual frequencies within particular frequency bands, and thus contained higher variance. This was done in order to reduce dimensionality of the representational patterns and facilitate the process of multiple comparisons correction (see below). To corroborate that our results were not affected by differences in the frequency features that we selected in each analysis, we conducted the (RSM-based) category model analysis in the same frequency bands as the EMS analyses (theta, alpha, beta, low-gamma, high-gamma). Our results revealed a significant fit of the category model during encoding in all frequency bands in the VVS, and a more restricted fit in the PFC in the beta band ($p_{corr} = 0.015$, Bonferroni corrected for 5 bands; Supplementary Fig 13A). During maintenance, we did not observe any significant fit in VVS or PFC in any band (VVS: all $p_{corr} = 1$; PFC: all $p_{corr} = 1$; Supplementary Fig 13B).

## RSA at high temporal resolution

We increased the temporal resolution of our sliding time window approach to compare the onset of category-specific information in VVS and PFC (Fig. 2E). In this analysis, power values were computed with the same method and parameters as in the main contrast-based analysis, but at an increased temporal resolution (10 ms). Feature vectors were constructed in 500 ms time windows and the 50 time-points included in each window were averaged separately for electrodes and frequencies, resulting in a two-dimensional representational pattern. We included all individual frequencies in the 3–150 Hz range (a total of 52). These two-dimensional frequency x electrode

vectors were concatenated into one-dimensional arrays for similarity analyses. We employed a sliding time-window approach with incremental steps of 10 ms resulting in an overlap of 490 ms between two consecutive windows, focusing only on matching time points (non-lagged correlations). We performed this analysis separately for the VVS and the PFC and assessed the statistical significance of the resulting time-series in each region. At each time point, we compared the group-level Fisher z-transformed rho values against zero. We also directly compared the values between PFC and VVS at the group level. Given that not all subjects had implanted electrodes in both of our two ROIs, we performed unpaired t-tests at each time-point. We corrected for multiple comparisons by applying cluster-based permutation statistics in the temporal dimension in all the pattern similarity analyses (see section *Multiple Comparisons Correction* below).

## Feedforward and recurrent DNN models

We compared VWM representations in the iEEG data with those formed in two types of convolutional deep neural network (DNN) architectures: feedforward and recurrent DNNs. We used AlexNet[62], a widely applied network in computational cognitive neuroscience to model visual perception and WM, as our feedforward model[36,57]. We also employed two recurrent convolutional DNNs: BL-NET, which has been recently applied to model human reaction times in a perceptual recognition task[71], and corNET-RT, a network recently developed to model information processing in the primate ventral visual stream[72]. AlexNet is a deep convolutional feedforward neural network composed of five convolutional layers and 3 fully connected layers that simulates the hierarchical structure of neurons along the ventral visual stream. AlexNet was trained in the task of object identification, i.e., the assignment of object labels to visual stimuli, using the ImageNet dataset[110]. When learning to identify images, AlexNet develops layered representations of stimuli that hierarchically encode increasingly abstract visual properties: Early layers reflect low-level features of images such as edges or textures while deeper layers are sensitive to more complex visual information, such as the presence of objects or object parts. Several studies demonstrated the validity of AlexNet as a model of neural representations during biological vision, showing that it can capture relevant features of information processing in the VVS of humans during perceptual and mnemonic processing[36,58,64]. We computed RSMs at every convolutional and fully connected layers of the network, following previous work[57,58].

The BL-NET is a deep recurrent convolutional neural network consisting of 7 convolutional layers with feedforward and lateral recurrent connections, followed by 7 batch normalization and RELU layers. Every unit in the BL-NET network receives lateral input from other units within feature maps. BL-NET has demonstrated high accuracy in the task of object recognition[71] after being trained with two large-scale image datasets (i.e., ImageNet and Ecoset[71,76];). We tested the network trained with these two different datasets in our analyses. Given that the output of each layer, which combines activity of lateral and feedforward connections, is computed at every single time-step in the RELU layers of the model, we selected these specific layers to compute the RSMs in our main analyses[71]. We obtained similar results when we compared the activations extracted from the convolutional layers (after batch normalization).

The corNET-RT network is another prominent example of recurrent architectures that have been employed to model neural activity in the VVS of primates. It comprises four layers designed to capture information processing in the main four VVS regions: V1, V2, V4, and IT. Like the BL-NET, corNET-RT exclusively incorporates lateral and not across area connectivity. Each layer of the network consists of an input and output convolutional layer, group normalization and RELU non-linearities. Unlike the BL-NET, the number of recurrent steps in each layer is not fixed but varies from 5 (in layer V1) to 2 (in layer IT). RSMs were computed specifically for the convolutional layers (we selected

the output convolutional modules in each layer), although similar results were observed when RSMs were computed from the outputs of the non-linear layers.

BL-NET and corNET-RT are two of the most prominent task-performing convolutional DNN models for image classification that have introduced recurrence as a main architectural feature. These networks have shown improvements in performance as compared to parameter-matched feedforward networks in the complex task of object recognition[71,72]. Theoretical accounts and experimental findings have proposed that recurrent DNNs can better explain neural activity in the VVS and behavioral data than feedforward networks[33,59,69,72,111]. While previous studies characterized neural representations in humans using recurrent models in the domain of visual perception[33], no study so far has used these types of architectures to model VWM, and no study has applied them to iEEG data.

Note that the BL-NET and the corNET-RT networks have different unrolling schemes across time, which affects how activity propagates through the networks. In BL-NET, feedforward and recurrent processing happen in parallel: a feedforward pass takes no time, while each recurrent step takes 1 time point. Thus, each layer receives a time-varying feedforward input. In corNET-RT, on the other hand, the onset of responses at deep layers is delayed when recurrence is engaged in earlier layers. These two approaches have been referred to as unrolling in 'biological' time (corNET-RT) vs 'engineering' time (BL-NET, see refs. 71,72).

### Stimulus representations in DNNs

In order to analyze how the different DNN architectures represented the stimuli in our study, we presented the networks with our images and computed unit activations at each layer. We calculated Spearman's correlations between the DNN features for every pair of pictures, resulting in a $60 \times 60$ representational similarity matrix (RSM) in each layer[47]. The AlexNet unit activations were computed using the Matlab Deep Learning Toolbox. Images were scaled to fit the 227 x 227 input layer of the network. The unit activations in the BL-NET network were extracted using the pipeline described in https://github.com/cjspoerer/rcnn-sat. Images were scaled to $128 \times 128$ pixels, and normalized to values between −1 and 1 to fit the input layer of the network as it was originally trained. The number of recurrent passes in the BL-NET architecture was set to 8 time-steps in each layer. We extracted the unit activations at each of these time points and computed RSMs, resulting in a total of 7 (layers) × 8 (time-points) = 56 RSMs. corNET-RT activations were extracted using TorchLens[112], and we corroborated the results using the 'TorchVision' toolbox[73]. Images were z-scored to the mean and standard deviation of the ImageNet database and scaled to $224 \times 224$ pixels to match the training parameters of the network.

To visualize the representations of stimuli in our networks, we employed Multidimensional Scaling (MDS). MDS is a dimensionality reduction technique which exploits the geometric properties of RSMs, projecting the high-dimensional network activation patterns into lower-dimensional spaces. To apply the MDS algorithm to our RSMs, we subtracted the correspondent values in the matrix from 1 to obtain a distance metric and projected the data into two dimensions (Figs. 3A, 4E and 5E).

Importantly, all three architectures we employed were trained with the ImageNet dataset, in which none of the categories included in our study ('house', 'robot', 'hand', 'face', 'planet', and 'tree') are present as object labels. For this reason, we did not focus our analysis on network classification performance but characterized categorical representations that were formed across layers, computing within-category, between-category correlations and their difference (CCI scores, see below). Moreover, we performed an additional control analysis involving a variant of BL-NET trained with the Ecoset dataset[76],

which contains part of our stimuli labels (i.e., labels 'house', 'robot' and 'tree') to corroborate our main results (Supplementary Fig 10).

The segregated representation of images according to their classes in deep layers of convolutional DNNs is a well-documented phenomenon[113,114]. This segregation however can be achieved by (1) grouping together items belonging to the same category, (2) separating items belonging to different categories, or (3) a combination of these two processes. To distinguish among these possibilities, we separately computed within-category and between-category correlations in all DNN layers. Results are presented in Figs. 3C, 4D and 5D, and in Supplementary Fig 5. In addition to computing the representational geometry of stimuli across all layers of the networks, we quantified the similarity between RSMs across layers using Spearman's Rho (a "second-level" similarity metric; see ref. 74). We applied MDS to visualize the similarity structure of the initial and last time points in each layer of BL-NET and Cornet-RT (Figs. 4B and 5B).

To quantify the amount of category information in the different layers of the networks, we computed a Category Cluster Index (CCI), defined as the difference of average within-category and between-category correlations in the DNN representations of the stimuli. Both within and across category correlation averages were computed after removing the diagonal of the RSM matrices (which only contains values of 1 by definition) and duplicated values due to the symmetry of the RSMs. CCI approaches 1 if representations in all categories are perfectly clustered and 0 if no categorical structure is present in the data[74]. We computed CCI at each layer of the AlexNet (Fig. 3C), and for each time point in each layer of the BL-Net and the corNET-RT networks (Figs. 4D and 5D). To assess whether the observed CCI values were significant, we implemented a permutation procedure. We built a distribution of CCI values expected by chance by shuffling the trial labels of the network RSMs 1000 times and recomputing CCI values. We considered significant CCI values that exceeded the 95th percentile of these null distributions.

In order to better characterize categorical representations in our networks and directly compare them, we performed a linear fit of within-category and between-category correlations across layers (Supplementary Fig 5). We specifically focused on the last time point in each layer in our recurrent architectures. We computed the correlation of the activations corresponding to every pair of items in each layer and performed a linear least-squares fit with the resulting values (270 within-category correlations and 1500 between-category correlations were computed in each layer). To evaluate whether correlations increased or decreased linearly across layers, we compared the distribution of slopes taken from the linear fit against zero (representing the null hypothesis of an average flat line) in each individual network. In addition, we compared these distributions across networks using paired t-tests.

Please note that given that two versions of the experiment were created for German and Chinese participants (with Angela Merkel and Jackie Chan as face stimuli, respectively), we passed through the networks two different datasets of images. For visualization of network RSMs and corresponding MDS plots (Figs. 3A, 4E and 5E), we employed the German version of the stimuli. In the analyses focusing on the network representations, we generated independent statistics for the two stimuli sets and then averaged them. This applies to the plots showing the representational consistency of networks across layers and time-points, within- and between-category correlations and CCI scores (Figs. 3B, C, 4A–D, and 5A–D).

### Representational similarity analyses based on DNNs: modeling neural representations with deep neural networks

We compared the representations formed in the DNN architectures with the iEEG representations using RSA. Neural RSMs were constructed following the procedure described in the section *Model-based RSA* above.

Similar to the category model analysis, we performed a time-frequency resolved analysis of fits of neural and DNN-based RSMs. In this analysis, neural RSM time series (same time windows as described above) were computed with feature vectors comprising information of each individual frequency independently (e.g., for 3 Hz, 4 Hz, … 150 Hz). The resulting RSM time-series were correlated with network RSMs at each individual layer. Individual frequencies were extracted using the same parameters as in the category model and the contrast-based analyses (see section *Time-frequency analysis*). The resulting time-series of correlation values were stacked into time-frequency maps of model fits. Significance was determined by contrasting the observed Fischer-Z transformed rho values against zero at the group level. Results were corrected for multiple comparisons using cluster-based permutation statistics and we additionally applied Bonferroni corrections across layers (see below).

We separately analyzed the fit of particular DNN models to the within-category and the between-category correlations in the neural data (Supplementary Fig 1, Supplementary Fig 3). In these analyses, we excluded the within-category or the between-category correlations from both the neural data and the models before vectorizing the RSMs and computing the correlations.

### Trial-based DNN analyses: correct vs incorrect

Since the number of incorrect trials was substantially lower than the number of correct trials in our data, directly comparing DNN correlations of the RSMs of incorrect vs. correct trials would be unbalanced. We thus computed a single-trial metric of model fits by correlating each row in the model RSMs and the neural RSMs independently[92]. Because of the unbalanced trial numbers, we then analyzed the match of these representations at all time-frequency points and for all DNN layers, instead of selecting the time periods where we had observed the original effects (which were mainly driven by the correct trials, given their larger number). This resulted in one time-frequency map of model fits for each trial. We averaged these trial-specific fits separately for correct and incorrect trials and in the two ROIs and evaluated whether there was a difference between correct and incorrect trials at all time-frequency bins. We only included subjects with at least 5 trials in each condition, leading to a total of 18 participants for VVS and 13 in PFC (paired *t*-tests were applied to the average time-frequency maps across conditions). Given the relevance of categorical information for PFC and VVS representations, trials in which the correct category of the cued item was reported were considered as correct, and trials in which subjects failed to retrieve the correct category were considered incorrect. This led to a total number of $13.22 \pm 9.12$ incorrect trials and $65.23 \pm 30.64$ correct trials in the VVS analysis (Mean ± STD), and of $11.77 \pm 4.94$ incorrect trials and $60.26 \pm 26.55$ correct trials in the PFC analysis (Mean ± STD).

### Stress analysis

We performed Multidimensional Scaling (MDS) on the RSMs during encoding and maintenance at various levels of dimensionality and computed the stress value of the MDS projections. Stress (a.k.a. Stress-1) is a metric of the goodness of fit of a particular MDS projection that reflects how well a lower-dimensional embedding—in a specific dimension—reflects the structure of the high-dimensional data. Stress values are low if the data can be relatively well embedded in lower dimensions, and high if the embedding is less accurate. We performed the MDS analysis for all dimensions in the 1–60 range (corresponding to the size of the average RSM). For every subject, we converted the RSMs into distance matrices (1-correlation), performed MDS and computed stress in each time period. Given that the stress metric is sensitive to the number of dimensions of the distance matrix, we randomly removed items in RSMs to match the number of the condition with less items (some subjects had less than 60 trials during the maintenance period because some trials were removed during artifact

rejection). This was done 100 times to corroborate that the results were not affected by the specific random selection of trials. We subsequently compared the group level stress values during encoding and maintenance for every dimension independently, and assessed regions of contiguous dimensions with significant differences between the two time periods. We applied cluster-based permutations statistics to control for multiple comparisons correction (see below).

### Multiple comparisons corrections

We performed cluster-based permutation statistics to correct for multiple comparisons in the pattern similarity analyses (Fig. 2), in the RSA-DNN analyses (Figs. 3–5), and in the analysis of different levels of stress during encoding and maintenance (Supplementary Fig 1).

In the pattern similarity analyses, we applied cluster-based permutation statistics both for the temporal generalization analysis (Fig. 2D, F), and for the temporally resolved analysis (Fig. 2E). For both analyses, we contrasted same and different category correlations at different time-points using *t*-tests, as in the main analysis, after shuffling the trial labels 1000 times. We considered significant a time-point if the difference between these surrogate conditions was significant at $p < 0.05$ (two-tailed tests were employed). At every permutation, we computed clusters of significant values defined as contiguous regions in time where significant correlations were observed and took the largest cluster at each permutation. Please note that in the temporal generalization analysis, time was defined in two dimensions and clusters were formed by grouping significant values across both of these dimensions, while in the temporally resolved analysis (Fig. 2E), correlations were computed at matching time-points and clusters were formed along one temporal dimension. In both analyses, the permutation procedure resulted in a distribution of surrogate *t*-values under the assumption of the null hypothesis. We only considered significant those contiguous time pairs in the empirical (non-shuffled) data whose summed *t*-values exceeded the summed *t*-value of 95% of the distribution of surrogate clusters (corresponding to a corrected $P < 0.05$; see ref. 115).

We also performed cluster-based permutation statistics in the analysis at high temporal resolution in which we directly compared similarity values between VVS and PFC. In this analysis, we computed clusters of significant EES differences between the two regions for every time-point by applying unpaired *t*-tests. We repeated this analysis 1000 times after shuffling the region labels and kept the summed value of the largest cluster at every permutation. We only considered significant those clusters in the empirical data above the 95th percentile of the shuffled distribution.

In the RSA-DNN analyses (and also in the category model RSA analysis, Fig. 2), we applied cluster-based permutation statistics. To determine the significance of the correlations between neural and model RSMs, we recalculated the model RSMs at each layer of the network after randomly shuffling the labels of the images. The surrogate model similarity matrices were then correlated with the neural similarity matrix 1000 times at all time-frequency pairs. As in the original analysis, we computed the correlations after removing the diagonal of the RSMs and only took half of the matrices given their symmetry. We identified clusters of contiguous windows in the time-frequency domain where the group-level correlations between neural and network RSMs were significantly different from zero at $p < 0.05$ (two-sided test) and selected the maximum cluster size of summed *t*-values for every permutation. This resulted in a distribution of surrogate *t*-values. The statistical significance was then determined by comparing the correlation values for the empirical data with the distribution of correlation values for the surrogate data (clusters whose summed *t*-values exceeded the 95% of the null distribution were considered significant).

In addition to cluster-based permutations, we also corrected our results for multiple comparisons using the Bonferroni method in the

contrast-based RSA analyses (Fig. 2) and in the model-based RSA analyses (Figs. 2, 3, 4, and 5). In the contrast-based analyses, given that we tested five different frequency bands, we only considered $p$-values significant that were below an alpha of 0.05/5. In the model-based analyses, we adjusted the significance threshold according to the correspondent number of layers in each network that was tested (AlexNet = 8; BL-NET = 7, corNET-RT = 4). The same correction by number of layers was applied in the CCI analysis (Figs. 3C, 4D and 5D).

To correct for multiple comparisons in the correct versus incorrect trial level DNN fit analysis, we shuffled the condition labels (correct versus incorrect) 1000 times. At each permutation, we calculated the summed $t$-values of the significant differences between conditions with shuffled labels, resulting in a distribution of summed $t$-values under the null hypothesis. We then ranked the observed $t$-value with respect to this distribution to assess statistical significance.

In the analysis of different levels of stress during encoding and maintenance, we randomly shuffled the condition labels (encoding or maintenance) in each subject independently 1000 times and recomputed the condition differences. At each permutation, we summed the $t$-values of the largest cluster of significant dimensions, resulting in a distribution of $t$-values expected by chance. We ranked the observed $t$-values with respect to this null distribution to assess statistical significance.

### Reporting summary

Further information on research design is available in the Nature Portfolio Reporting Summary linked to this article.

## Data availability

Anonymized intracranial EEG data supporting the findings of this study have been deposited in the Open Science Framework (https://osf.io/mw8cf/). Source data are provided with this paper.

## Code availability

Custom-written Matlab and Python code supporting the findings of this study are available at https://github.com/dpachec/WM.

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

## Acknowledgements

We would like to acknowledge DFG funding via the ORA project "WMREPS Hidden brain states underlying efficient representations in working memory" (project number 396894956). N.A. also acknowledges DFG funding via the SFB 1280, project number 316803389. This project was a collaboration with Mark Stokes (Oxford) and Elkan Akyürek (Groningen) and is dedicated to the memory of Prof. Stokes.

## Author contributions

Conceptualization, M.-C.F., D.P.-E. and N.A.; methodology, D.P.-E., H.Z., L.K., G.X. and N.A.; data collection, M.-C.F., L.K., C.R., A.B., L.Y., S.W. and J.L.; data analysis: D.P.-E. and N.A.; writing—original draft, D.P.-E.; writing —review and editing, D.P.-E. and N.A.; funding acquisition, N.A.; resources, L.K., H.Z., P.R., A.B., A.S.-B., L.Y., S.W. and G.X.; supervision N.A.

## Funding

## Competing interests

The authors declare no competing interest.
