## [Peer Review File · Nature Communications]

Maintenance and transformation of representational formats during working memory prioritizationREVIEWER COMMENTS

Reviewer #1 (Remarks to the Author):

In this study, Estefan and colleagues address an important question related to visual information is represented in the human brain during visual working memory encoding and maintenance. During visual working memory (VWM), individuals may choose to prioritize some content over others. The question examined here is how such prioritization may change the representations of WM in the ventral visual stream (VVS) and the prefrontal cortex (PFC). The authors test the hypothesis that WM prioritization may involve task-dependent transformations of representational patterns in executive control regions which can be disentangled from mnemonic coding in perceptual regions. The authors examine this by capturing and analyzing intracranial EEG (iEEG) signals from human epilepsy patients, and importantly relate the neural activity captured through these recordings with two different DNN architectures to examine how different aspects of images are represented. One DNN architecture (AlexNet) is based purely on feedforward connections while the second (BL-NET and corNET-RT) involves lateral recurrent connections in each layer. The hypothesis is that since WM maintenance should involve some recurrent computations to stabilize the representations, then neural activity during WM maintenance should be more similar to the representations in the recurrent rather than strictly feedforward architectures. The authors test this, and leverage the temporal resolution of iEEG to also investigate which specific frequency bands may be relevant for these processes.

Overall, this is a good study with conclusions that will be relevant for our understanding of WM maintenance and prioritization in the human brain. The authors find that category specific information is represented in the VVS and PFC during encoding, but maintenance only demonstrates reinstatement of this information in the VVS. However, when linking this neural data with the DNN architectures, the authors find that in fact PFC activity appears to take on a representational format that is more abstract, represented in higher levels of the DNN, during the maintenance period, but only in the recurrent DNN. Thus they conclude that only recurrent DNN architectures are capable of capturing maintenance, and that the PFC changes its representations to reflect abstract information during the maintenance period. These data are consistent with prior literature, and add evidence to the premise that brain regions responsible for executive control may change their representations to reflect higher level WM information. The analyses are rigorously conducted, and the conclusions are well supported. There are a few suggestions that the authors might consider to strengthen the overall manuscript.

The authors perform an RSA analysis and find category representations in the VVS and PFC during encoding but not during maintenance. However, in the contrast analysis (within versus between categories), the authors observe significant encoding-maintenance similarity in the VVS during maintenance. The presumption is that the representations have changed during maintenance.

Thus, when correlating the neural data with the DNN's, the authors find significant correlations during maintenance, but only in deeper layers of the recurrent networks. The conclusion is therefore that VVS representations have transformed.

Conversely, for the PFC, the authors find that the neural data is correlated with the DNNs, also in deeper layers of the recurrent network, during maintenance. In contrast to the VVS, however, there is no evidence of encoding-maintenance similarity. The authors also take this as evidence that the representations in the PFC have also transformed. Yet, in this case, the transformation of PFC representations appears different than the transformations of the VVS representations (since the VVS also demonstrates significant EES), and perhaps more complete? The explanation offered by the authors, as far as I can understand, is that the PFC representations have transformed to match the high level features of the stimulus. But it appears that this is also happening for VVS, since the significant correlations are also only with the deeper layers. Is this correct, that the VVS is also transformed to match the higher level features? It would be helpful to clarify how one should think about the two different transformations that are happening in both regions since one of the main conclusions of the study is that the PFC transforms its representations to a different format while the VVS does not exhibit such a deep transformation.

On a related note, what are these 'abstract' representations that appear to fit only with deeper layers of the DNNs. Presumably, these are categorical representations, since they are represented in higher, conceptual levels of the DNN. If that were the case, then it would be helpful to also know whether neural activity, and the corresponding DNN's, can also distinguish individual exemplars within categories. Here, the hypothesis would be that superficial layers of DNNs (in both the feedforward and recurrent architectures) should be sufficient to distinguish individual exemplars, and this would match activity in the VSS during encoding.

Also to help clarify these issues, it would also be helpful to understand how these representations are transformed, and how they may be linked to the DNNs, during incorrect choices. Do we see the absence of any representations in the neural data during these trials, and are they similar to or different from the transformations that occur during correct trials. Are any categories (or exemplars) represented during this time, and are they related to their incorrect choice?

There are two maintenance periods. I assume that all of these analyses are during the second maintenance period after the cue is presented. This should be clarified, but I think most people would probably assume this as well. However, it would be interesting to know if there are any meaningful representations during the first maintenance period. Are all three stimuli represented (perhaps serially) during this time?

The authors also examine the representations of stimulus information in the separate DNN architectures. In deeper layers, categorical distinctions are more pronounced, although in the feedforward model this is because within category similarity increases whereas in the recurrent models across category similarity decreases. In both cases, categorical distinction is apparent in the deeper layers. Is this surprising? Previous studies of such DNNs in image classification have demonstrated that conceptual (or categorical) information is often better represented in deeper layers.

Reviewer #2 (Remarks to the Author):

Estefan and colleagues examine neural representations during a visual working memory task using intracranial EEG recordings and representational similarity analysis. Various categorical and neural network based representational analyses are conducted, revealing that memory representations in the PFC involve a transformation of format, which is revealed only interrogated with a recurrent deep neural net. Overall, this study is examining important questions about how working memory representations are transformed during maintenance, and using state-of-the art RSA approaches to do so. Further remarks are below, mainly relating to the analysis description and statistical methodology.

1. The first analysis is an involves a "simple categorical model" that examines an item-by-item similarity matrix and correlates this with a categorically coded similarity matrix. This reveals significant categorical coding during encoding VVS and PFC, but no such coding during maintenance. Next, to "evaluate whether transformed activity patterns from encoding reoccur during the maintenance period, we performed a category-specific pattern similarity analysis." How does this "category-specific" analysis differ from the "simple categorical" model. From the text, hard to tell--and the analyses are described similarly: "this analysis involved contrasting correlations of items belonging to the same category with correlations of items from different categories". And this line: "EES and EMS analyses test for re-occurrence of category-related neural activity patterns from different encoding periods" hardly clarifies the situation, since the "simple categorical" analysis couldn't work if category-related neural activity didn't "reoccur". A figure is then referred to (2A, 2C) but now the latter technique is called a "mnemonic scheme". (One must zoom in 300% or 400% to identify the little images on the left of the graph).

2. Having found no evidence for categorical coding in the PFC during maintenance, the manuscript turns to a series of DNN analyses, one using feedforward model (AlexNet), and two analyses with recurrent DNNs. This is indeed an important and worthwhile approach, but the presentation is not statistically convincing. The goal here is not to simply test the predictive performance (but see below for remarks on that) of several neural networks on iEEG activity during maintenance, but rather to say something about mechanism, namely recurrence. One issue, however, is that recurrence is not the only thing that differs between AlexNet and BL-net/cor-NET-RT. They have

other differences, including number of layers, training regimes, parameters, etc. One should like to say: BL-Net outperforms AlexNet due to its recurrent properties-- but this requires an "all other things being equal" comparison, which is not the case here. Ideally, one would have a model with the same architecture, same training set, training parameters, etc. where one had recurrence and one no recurrence.

3. If the recurrent models better model PFC activity during maintenance than the other models (categorical, Alexnet, etc.) then one should like to see a statistical evidence via model comparison. For example, AlexNet did not match neural representations in VVS, all ps > .056. But BL-Net was deemed a success with three p-values < .05 (.035, .035, .014). Is this really different level of predictive performance, though? The three analyses are presented independently, and plotted in successive grids of (very small) heatmaps in separate figures. Here is a case where an aggregate measure (to avoid contrasting every cell of the heatmaps) of cross-validated performance could be statistically compared between all models, starting with the simple categorical, with either a non-parametric test or an ANOVA-style analysis. Otherwise, we don't really know if the differences are reliable.

4. Comparisons between PFC and VVS are made implicitly at times, but it's important to keep in mind that there are different sample sizes (N=28, N=16), and therefore differences in power.

5. We read: "... studies (Liu et al., 2020, 2021) established a crucial role of transformed representational formats, particularly abstract representational formats devoid of specific sensory information, in VWM maintenance. Based on these insights, we hypothesized that the PFC may contain categorical information in an abstract representational format after presentation of the retro-cue."

This is not entirely clear, since a categorical format could be entirely abstract, and therefore picked up by the "simple categorical model". If "abstract" here simply means "non-sensory", then it is still the case that items in the same category share all non-sensory features. I think some work defining terms would help the reader here.

Reviewer #3 (Remarks to the Author):

In this study, using multi-item working memory task with a retro-cue, Pacheco Estefan and colleagues examined how executive systems select relevant items within working-memory space in the brain and how this selection process affects neural representational formats in ventral visual stream (VVS) and prefrontal cortex (PFC) in human participants. Based on the observation that only in PFC, the category specific information which was originally present in neural activity during

encoding did not re-occur during maintenance (post retro-cue), the authors suggested that in PFC, the task representation underwent significant transformation between before and after the presentation of the retro-cue, changing from signals representing category information to a higher-order abstract representation, putatively reflecting the prioritization process. The authors also reported that the recurrent DNN representations matched PFC representations in beta band following the presentation of the retro-cue, which, the authors suggested, reflected the top-down control over the visual WM.

Neural activity data appear to be recorded in sufficient volume and preprocessing is adequate. The method itself was clearly explained and easy to understand. However, as you can detail in the comments below, the main argument of this paper is not well supported for several reasons: First, the authors did not present sufficient evidence to support the aforementioned claim at the level of both neural and DNN-based analyses. Second, there appear to be fundamental flaws in some of the key analyses (e.g., ignoring the effect of the factor, retro-cue type on neural, especially prefrontal, activity, and using DNNs which did not actually perform the behavioral task), potentially invalidating the interpretation of the key observations. Third, and most importantly, no evidence was presented at all in both neural and model analyses as to what exactly the “abstract representation format” of category information and "prioritization process" in the prefrontal cortex are.

The conclusions drawn from the results are scarce, and the previous studies using similar behavioural paradigms should incorporate the key findings of this manuscript, with more detailed information on what has happened in the post-retro-cue epoch. (e.g., Everling et al., *Nat. Neurosci.*, 2002; Panichello and Buschman, *Nature*, 2021; Piwek et al., *PLoS Comput. Biol.*, 2023; Meyers et al., *TICS*, 2017). Below is a list of major comments and specific comments:

Major comments:

1. The paper argued that PFC contains categorical information in an abstract representational format after the presentation of the retro-cue. However, what exactly is the "abstract representational format" of categorical information was not explained at all even at the level of neural activity.

Rather, the authors only showed that the correlation between rDNN RSM and PFC RSM (in beta band) is significant at the deep layer of these models (Figures 4 and 5), and argued that this was the key evidence supporting the presence of “abstract” category information in PFC in the maintenance period (which seemed to be different from category specificity observed in the earlier, encoding period). This correlation, however, does not explain what the aforementioned "abstract representational format" of categorical information is, nor does it provide evidence that the information representation of PFC immediately after the retro-cue had an abstract representational format which was transformed from and thus different from the category representation, per se in the encoding epoch.

In neural information processing, abstraction usually means dimensionality reduction (e.g., Bernardi et al., *Cell*, 2020); the process of abstraction extracts the essential data structure by

reducing the amount of information to less than originally provided. In this study, as the authors noted, category information observed in the encoding phase did correspond to the abstract format of representation (i.e., representing visual objects not by their low-level visual features, but by semantic categories). However, as a main argument of this paper, the authors stated that following the retro-cue presentation, PFC started to represent “categorical information in an abstract representational format” (lines 240-241). The same claim was made repeatedly in the paper, i.e., that after the retro-cue, PFC began to represent different and perhaps higher-order information than the category representation observed in the encoding period. Then, what exactly does “categorical information in an abstract representational format” mean? The authors should clarify this point with additional neural analyses before moving on to analysis using artificial neural networks.

2. It is insufficient to analyze neural activity, especially prefrontal activities, by extracting only the single-item trials condition and focusing only on the factor, visual object type. According to the task explanation in Methods, the retro-cue used in this experiment represents four types of rules: multi-item trials ("All") and single-item trials (1, 2, or 3). It is well known that neural activity in the PFC strongly represents rules (e.g., Wallis et al., *Nature* 2001), varies with task difficulty (Barch et al., *Neuropsychologia*, 1997) and with the motivation of the individual. In this study, the retro-cue may have indicated the difficulty of the trial besides the behavioural rule for that trial itself. This is because even in the single-item condition, the recency effect and the primacy effect may have caused a difference in difficulty among the three object conditions. These higher-order representations of information including representation of rule and difficulty are expected to be more pronounced in PFC than in VVS. In other words, in comparing VVS and PFC, this paper completely ignores the main effect and interaction associated with the factor, retro-cue, which are expected to be particularly strong in PFC. In this sense, it is not surprising that in the present study, the same pattern of neural activity as in the encoding phase was reinstated only in VVS and not in PFC during the time immediately after the retro-cue presentation. As such, the authors should use full task variables and clarify at the level of neural activity why the reinstatement did not occur in PFC alone.

3. In relation to the above point, the most important argument of this paper was that the activity immediately after the retro-cue presentation in PFC represents an abstract representational format (i.e., not just a categorical representation, but something higher-order than that). However, the analysis time window that the authors focused on to obtain this conclusion was the period immediately after the presentation of the retro-cue (0 – 500 ms after retro-cue onset, e.g., Figure 4H), and during this period, as mentioned above, there should be strong neural responses to the various information that the retro-cue have carried. It is therefore not surprising that the activity during this time period differed from that during the encoding period.

4. Compared to previous publications, (e.g., Panichello and Buschman, *Nature*, 2021; Piwek et al., *PLoS Comput. Biol.*, 2023; Meyers et al., *TICS*, 2017), there seem to be little new findings in the

present manuscript. For example, Panichello and Buschman (2021) recorded neural signals in the macaque lateral prefrontal cortex (LPFC) during a retro-cueing task, contrasting delay-period activity between before and after the retro-cue onset. They reported that in the delay period prior to the presentation of retro-cue, the individual stimuli were maintained in independent subspaces of neural population activity, whereas in the delay period following the retro-cue presentation, the prioritized items (i.e., items indicated by the retro-cue) was rotated into a common subspace, potentially allowing a common readout mechanism. Another study (Piwek et al., 2023) modelled the emergence of these representational changes using a recurrent neural networks (RNNs) that was actually trained to perform an equivalent retro-cue task. These results have reinforced the currently influential theory concerning neural mechanisms underlying retro-cue processing (Meyers et al., TiCS, 2017). In addition, neural mechanisms for attentional filtering (prioritization) in PFC has also been extensively studied (e.g., Everling et al., 2002). The authors should clearly state how the present manuscript differs from these previous reports.

5. In drawing main conclusions of the paper (Figures 4 and 5), the authors only used DNNs trained on the object classification task (Method). These DNNs were not trained on the retro-cue task which was used to collect neural activity. In other words, the DNNs were exposed only to the object image input, while the neural activity was fed two types of information, object image and retro-cue. Therefore, because these DNNs do not receive retro-cue input to induce prioritization in the first place, the use of such DNNs can be insufficient for examining the transformation process of category signals induced by the retro-cue. Can one really compare two systems with different input and analyze the transformation process? If so, please specify the rationale for that.

6. The RSM in layer 7 of BL-net (Figure 4E, rightmost panel) and that in layer IT of corNET-RT (Figure 5E, rightmost panel) look very similar to the RSM (model RSM) in Figure 2A which was based on a simple model of category information that the authors used to analyze neural activity. In other words, the RSM in layer 7 of BL-net and layer IT of corNET-RT appear to be simply a noisy version of the model RSM in Figure 2B.

This brings up one question: isn't calculating the correlation between rDNN RSM and neural RSM in Figure 4H (layer 7, rightmost panel) and Figure 5G (layer IT, rightmost panel) essentially doing the same thing as calculating the correlation between the model RSM and neural RSM in Figure 2B? In fact, the resultant fit presented in Figure 2B and Figures 4H/5G look very similar to each other, except that there was a period of significant correlation only in Figures 4 H and 5G. Please elaborate further on this. Also, could you please quantify how similar the time-frequency correlation maps for the maintenance period were between Figure 2B and Figure 4H/5G (rightmost panels)?

If these two analyses (i.e., calculating the correlation between rDNN RSM and neural RSM in Figure 4H/5G, and calculating the correlation between the model RSM and neural RSM in Figure 2B) are essentially doing the same thing, then, doesn't the observed significant cluster in beta band in PFC (Figure 4H rightmost panel) merely indicate a presence of a variant of category specificity as already shown in Figure 2 (for encoding period only)?

7. Regarding the electrode locations, medial, lateral and orbital prefrontal electrodes are all analyzed together as “prefrontal electrodes”. It is known that category information, rule, and attentional prioritization are mainly processed in the lateral prefrontal cortex. I would be very interested to see how the results would look like if the electrodes were limited to only those in the lateral frontal cortex.

Specific comments:

1. In Fig. 2, the authors should label the task periods, the meaning of time zero, and the units of values in a colorbar within the figures. Readers should not have to guess or look for this information.

2. In Figure 1C, the electrodes are mapped on the transparent brain, which is very difficult to see. Please illustrate them in a different format or at least add multiple coronal slices from anterior end to posterior end of the electrode locations.

3. Lines 170-172: the authors directly compared proportion correct rates between the two conditions which had different chance levels. Because the single-item and multi-item conditions have very different chance levels (0.167 and 0.0083, respectively), this result does not necessarily reflect the difference in performance between the two tasks.

4. Lines 348-349: Ecoset-trained BL net was only applied to the time-frequency clusters that showed significant results in the ImageNet-trained BL net. The same analysis should be performed for all time-frequency ranges for the eco-set BL net as in Figure 4 and the results should be presented.

5. Lines 398 and 403: In the discussion section, the authors stated that in the encoding phase, “PFC exhibited robust category-specific representations during WM encoding” (L403), but the representation in PFC “matched only high-level visual and abstract formats from a recurrent DNN following the retro-cue.”

Again, what is meant by high-level visual and abstract formats is still not clearly mentioned here. Since this is the most important argument of the paper, it should be specifically indicated with some evidence.

REVIEWER COMMENTS

Reviewer #1 (Remarks to the Author):

In this study, Estefan and colleagues an important question related to visual information is represented in the human brain during visual working memory encoding and maintenance. During visual working memory (VWM), individuals may choose to prioritize some content over others. The question examined here is how such prioritization may change the representations of WM in the ventral visual stream (VVS) and the prefrontal cortex (PFC). The authors test the hypothesis that WM prioritization may involve task-dependent transformations of representational patterns in executive control regions which can be disentangled from mnemonic coding in perceptual regions. The authors examine this by capturing and analyzing intracranial EEG (iEEG) signals from human epilepsy patients, and importantly relate the neural activity captured through these recordings with two different DNN architectures to examine how different aspects of images are represented. One DNN architecture (AlexNet) is based purely on feedforward connections while the second (BL-NET and corNET-RT) involves lateral recurrent connections in each layer. The hypothesis is that since WM maintenance should involve some recurrent computations to stabilize the representations, then neural activity during WM maintenance should be more similar to the representations in the recurrent rather than strictly feedforward architectures. The authors test this, and leverage the temporal resolution of iEEG to also investigate which specific frequency bands may be relevant for these processes.

Overall, this is a good study with conclusions that will be relevant for our understand of WM maintenance and prioritization in the human brain. The authors find that category specific information is represented in the VVS and PFC during encoding, but maintenance only demonstrates reinstatement of this information in the VVS. However, when linking this neural data with the DNN architectures, the authors find that in fact PFC activity appears to take on a representational format that is more abstract, represented in higher levels of the DNN, during the maintenance period, but only in the recurrent DNN. Thus they conclude that conclude that only recurrent DNN architectures are capable of capturing maintenance, and that the PFC changes its representations to reflect abstract information during the maintenance period. These data are consistent with prior literature, and add evidence to the premise that brain regions responsible for executive control may change their representations to reflect higher level WM information. The analyses are rigorously conducted, and the conclusions are well supported. There are a few suggestions that the authors might consider to strengthen the overall manuscript.

We thank the reviewer for the constructive and positive assessment of our manuscript and the suggestions for improvement. In response, we have addressed all raised concerns and conducted a series of control analyses as recommended by the referee.

The authors perform an RSA analysis and find category representations in the VVS and PFC during encoding but not during maintenance. However, in the contrast analysis (within versus between categories), the authors observe significant encoding-maintenance similarity in the VVS during maintenance. The presumption is that the representations have changed during maintenance. Thus, when correlating the neural data with the DNN's, the authors find significant correlations during maintenance, but only in deeper layers of the recurrent networks. The conclusion is therefore that VVS representations have transformed. Conversely, for the PFC, the authors find that the neural data is correlated with the DNNs, also in deeper layers of the recurrent

network, during maintenance. In contrast to the VVS, however, there is no evidence of encoding-maintenance similarity. The authors also take this as evidence that the representations in the PFC have also transformed. Yet, in this case, the transformation of PFC representations appears different than the transformations of the VVS representations (since the VVS also demonstrates significant EES), and perhaps more complete? The explanation offered by the authors, as far as I can understand, is that the PFC representations have transformed to match the high level features of the stimulus. But it appears that this is also happening for VVS, since the significant correlations are also only with the deeper layers. Is this correct, that the VVS is also transformed to match the higher level features? It would be helpful to clarify how one should think about the two different transformations that are happening in both regions since one of the main conclusions of the study is that the PFC transforms its representations to a different format while the VVS does not exhibit such a deep transformation.

We thank the reviewer for this question, which addresses a crucial aspect of our study. Indeed, as described by the reviewer, we found that both VVS and PFC representations matched the representations in a category model during encoding but not during maintenance, indicating that transformations occur in both regions. These transformations have similarities and also differ in several aspects, which we discuss below.

First, we note that representations from encoding reoccur during maintenance in VVS but not PFC. In the VVS, all representational signatures that were observed during maintenance (corresponding to the deeper layers of the two recurrent networks) were already apparent during encoding – and this likely explains the significant encoding-maintenance similarity (EMS) in this region. However, some of the encoding signatures (categorical representations, representations in the feedforward network, and representations in lower layers of the recurrent networks) did not reappear in VVS. Thus, it seems that maintenance in VVS corresponds to a partial and selective re-appearance of encoding formats, which we call the “mnemonic coding scheme”. By contrast, in the PFC, the representational signatures that were observed during maintenance (again corresponding to deep layers of the recurrent networks) did *not* already occur during encoding, and thus the PFC does *not* show such a “mnemonic coding scheme”. Thus, as the reviewer points out, the transformation indeed seems to be more complete (or deep) in this region. We have described this difference more explicitly in the revised manuscript (p. 15) when we write: “*In the VVS, all representational signatures that were observed during maintenance (corresponding to the deeper layers of the two recurrent networks) were already apparent during encoding, and this likely explains the significant encoding-maintenance similarity (EMS) in this region. Thus, maintenance in VVS corresponds to a partial and selective re-appearance of encoding formats, corresponding to a mnemonic coding scheme. By contrast, in the PFC, the representational signatures that were observed during maintenance did not already occur during encoding, and thus the PFC does not show such a mnemonic coding scheme but exhibits a more profound transformation. We refer to the format of PFC representations after the retro-cue as ‘prioritized’*”.

Second, the time periods during maintenance when representational signatures could be matched by deep layers of the recurrent DNNs differed between the two regions. In the PFC, this effect occurred *early* during the maintenance period, immediately following

the retro-cue and thus PFC representational signatures during maintenance are putatively related to the prioritization process. This is why we call this a “prioritization coding scheme”. By contrast, in the VVS, the effect occurred *late* during the maintenance period, just before the presentation of the probes and thus in preparation for the response. Maintenance in the two different regions thus likely serves different functional roles. Again, this is now described explicitly in the manuscript on p.16: “*An important difference between VVS and PFC pertains to the time period at which representations matched those from a recurrent DNN: directly following the retro-cue in PFC, but prior to the probe in the VVS and thus in preparation for the response. This suggests that maintenance in the two different regions likely serves different functional roles.*”

Third, as described by the reviewer, we found that during maintenance, representations in both regions matched representations in deep layers of our recurrent models (layers 4-5-6-7 in BL-NET and layer IT in CorNet-RT; Figures 4H and 5G). This raises the question whether the two regions indeed display the same format of representation (at different time periods) during maintenance. We performed additional analyses to characterize the representational transformation and the final representational formats in our two ROIs more comprehensively, which we now present in Supplementary Figure 1. First, to confirm representational transformations between encoding and maintenance, we correlated the RSMs during these two time periods, separately in each region. We focused on the respective clusters where we observed significant fits in the original analysis: In the PFC, we selected one of the encoding clusters that corresponded to the category model (in the beta range), and the cluster during the prioritization period showing significant matching to the BL-NET model (Supplementary Figure 1B, bottom row). Consistently, in the VVS, we selected the cluster matching the category model during encoding, and an intersection cluster that included all significant time frequency bins of the BL-NET analysis during maintenance (layers 4, 5 and 6; Supplementary Figure 1B, top row). For each subject, we extracted the average neural RSMs in these different periods and correlated them between encoding and maintenance. Results revealed that RSMs between encoding and maintenance were not significantly correlated in either region (although the VVS showed a trend: $t(25) = 2.01$, $p = 0.055$; PFC: $t(14) = 0.71$, $p = 0.48$), indicating transformations of representational formats. Second, to further characterize these transformations, we computed the average of all pairwise between-item correlations in the neural RSMs. Between-item correlations in the VVS increased significantly from encoding to maintenance ($t(25) = -3.67$, $p = 0.0011$). A similar effect was observed at a trend level in the PFC ($t(14) = -2.12$, $p = 0.09$), but the changes were significantly more pronounced in VVS than PFC ($t(9) = -2.34$, $p = 0.043$). Thus, representations of individual items are more clustered during maintenance than during encoding; however, the overall stronger transformation in the PFC than in the VVS cannot merely be explained by stronger clustering of individual-item representations, which was even more pronounced in the VVS. We included these results in the new Supplementary Figure 1. We also refer to them in the results section p. 8: “*In several additional control analyses, we comprehensively characterized the representational formats in VVS and PFC (Supplementary Figure 1)*”.

Fourth, to investigate whether the representational transformations in the two regions resulted in a similar format, given that we observed a significant matching to the BL-NET in both regions, we directly contrasted the average RSMs in our two ROIs during the maintenance period. We again focused on the subset of subjects with electrodes in

both regions ($N = 10$) and on the clusters observed in each region in the original BL-NET analysis. We found that the RSMs during maintenance were not correlated ($t(9) = -0.07, p = 0.94$), suggesting substantial differences in representational geometries. We next separately computed correlations with the BL-NET model focusing either only on the within-category correlations or only on the between-category correlations. For the within-category analysis, we observed that BL-NET representations matched neural representations only in the VVS ($t(26) = 3.85, p = 0.00071$), but not in PFC ($t(15) = 0.42, p = 0.67$; Supplementary Figure 1C, top). In the between-category analysis, the BL-NET significantly matched neural RSMs in both regions (PFC: $t(15) = 3.96, p = 0.0024$; VVS = $t(26) = 3.48, p = 0.001$; Supplementary Figure 1C, bottom). Taken together, these findings reveal that the representational transformations in both regions resulted in a dissociation of the coding scheme of VVS and PFC in terms of their representational content, with a more prominent reliance on within-category correlations in the VVS than the PFC.

We believe our results allow for a comprehensive description of the similarities and differences of the representational transformations in VVS and PFC. Our two ROIs represented content differently regarding the recurrence of information from the encoding period (VVS: yes, PFC: no), the timing during maintenance (VVS: prior to probe, PFC: following retro-cue), and the resulting representational format (VVS: matching to within-category BL-NET representations, PFC: only matching to between-category BL-NET representations). Thus, our new analyses, reported in the new Supplementary Figure 1, show that while both regions transformed their representations to match high-level stimulus features, they differed in critical aspects related to the representational content and dynamics of these representations.

On a related note, what are these 'abstract' representations that appear to fit only with deeper layers of the DNNs. Presumably, these are categorical representations, since they are represented in higher, conceptual levels of the DNN. If that were the case, then it would be helpful to also know whether neural activity, and the corresponding DNN's, can also distinguish individual exemplars within categories. Here, the hypothesis would be that superficial layers of DNNs (in both the feedforward and recurrent architectures) should be sufficient to distinguish individual exemplars, and this would match activity in the VSS during encoding.

We thank the reviewer for raising this point. First, we note that the DNNs models we employed were trained to categorize objects based on their visual features and are thus sensitive to categorical information, in particular in their deep layers. To quantify the degree of categorical organization in VVS and PFC during the time periods of significant BL-NET fits, we computed a metric of category clustering (within-category vs. between-category correlations). This analysis indeed revealed that categorical information is an important dimension of representations in PFC during item prioritization ($t(14) = 3.3, p = 0.0053$), but not in VVS in the late maintenance period ($t(25) = 1.64, p = 0.113$). In addition, the VVS relied more on within-category distinctions than the PFC (see our response to the previous comment). Notably, however, categorical distinctions were not sufficient to explain PFC representations during the prioritization period, because the category model by itself did not show significant fits with the neural data (Figure 2B).

Following the suggestion of the reviewer, we performed additional analyses to investigate item-specific representations in our data. First, we constructed an ‘item model’ RSM and evaluated how well it could explain the representational geometry in our data. In the RSM of this model, correlations between repeated presentations of the same items were coded with a 1, correlations between different items of the same category were coded with a zero, and correlations between different categories were excluded (Supplementary Figure 3A). This model matched well with representations in the VVS during encoding in the low theta band (3-5 Hz; $p = 0.009$), and with representations in the PFC in the high beta band (25-29Hz; $p = 0.02$, Supplementary Figure 3B, left). During maintenance, the item model matched representations in the VVS at trend level (all clusters: $p_{\text{corr}} > 0.07$, Supplementary Figure 3B, right); note that this analysis had reduced statistical power, since only 11 participants with VVS electrodes had enough (i.e., > 12 , with at least one in each category) repetitions of prioritized items. We could not run a corresponding analysis for the PFC, because only four subjects had enough item repetitions in this region during maintenance. Thus, item-specific information could be identified at the neural level (independent of DNNs) during encoding, but to a lesser degree during maintenance.

To evaluate whether the DNNs could distinguish individual exemplars within categories, as suggested by the reviewer, we performed an additional analysis focusing on the AlexNet and the BL-NET networks. RSMs were created at each layer of these architectures across all pairs of trials (no averaging was conducted across repetitions). We excluded all between-category correlations and computed correlations between neural RSMs and model RSMs at all time-frequency points in VVS and PFC (Supplementary Figure 3C). In the VVS, we observed significant fits with item-level AlexNet representations of layer 1 ($p_{\text{corr}} = 0.008$), layer 5 ($p_{\text{corr}} = 0.04$), layer 6 ($p_{\text{corr}} = 0.016$) and layer 7 ($p_{\text{corr}} = 0.024$, Supplementary Figure 3D, top) and with item-level BL-NET representations of layer 6 ($p_{\text{corr}} = 0.028$) and layer 7 ($p_{\text{corr}} = 0.007$, Supplementary Figure 3D, top). Thus, neural representations indeed matched exemplar-specific representations in the DNNs. Notably, the presence of item-specific information was not only observed in superficial layers in our architectures, but also in intermediate and deep layers, which underscores the relevance of high-level visual information for the representation of individual exemplars. In the PFC, we observed a significant fit with AlexNet representations in layer 1 during encoding ($p_{\text{corr}} = 0.024$, Supplementary Figure 3D, bottom), while no significant fits with BL-NET representations were observed (all $p_{\text{corr}} > 0.28$, Supplementary Figure 3D, bottom). During maintenance, we focused solely on the BL-NET network because Alexnet did not show any significant fits in the original analysis. In the VVS, results revealed significant matching with exemplar-specific BL-NET representations in layers 5 and 6 in a time-frequency period that overlapped with the original fits observed in the main analysis (layer 5: $p_{\text{corr}} = 0.014$; layer 6: $p_{\text{corr}} = 0.014$, Bonferroni corrected for seven layers, Supplementary Figure 3E, top). No significant clusters were observed in the PFC (all $p > 0.25$, Supplementary Figure 3E, bottom). These analyses reveal a further difference in the resulting representational formats between VVS and PFC (see preceding question): during maintenance, exemplar-specific representations can be found in VVS but not PFC.

Together, these results show that exemplar-specific representations could be extracted from the iEEG data during encoding in both VVS and PFC and that the VVS maintained this information during the maintenance period. Interestingly, the significant fits we observed in the VVS towards the end of the maintenance period overlapped with

the clusters found in the original analysis, pointing towards a possible functional relevance of item-specific reactivations before the presentation of the probe.

We have included these additional results in the new Supplementary Note 2. We also modified the methods section of the manuscript (p. 37), to describe the methodologies employed in these additional analyses.

Also to help clarify these issues, it would also be helpful to understand how these representations are transformed, and how they may be linked to the DNNs, during incorrect choices. Do we see the absence of any representations in the neural data during these trials, and are they similar to or different from the transformations that occur during correct trials. Are any categories (or exemplars) represented during this time, and are they related to their incorrect choice?

We thank the referee for this suggestion. Since the number of incorrect trials was substantially lower than the number of correct trials, a null effect for the incorrect trials would be hard to interpret. Similarly, directly comparing DNN correlations of the RSMs of incorrect vs. correct trials would be unbalanced given that the RSMs of correct trials are so much larger. We thus followed a previous paper which compared DNN fits of correct vs. incorrect long-term memory retrieval in fMRI data and which computed a single-trial metric of model fits, by correlating each row in the model RSMs and the neural RSMs independently (Davis et al., Cereb Cortex 2021). Because of the unbalanced trial numbers, we then analyzed the match of these representations at all time-frequency points and for all DNN layers, instead of selecting the time periods where we had observed the original effects (which were mainly driven by the correct trials, given their larger number). This analysis resulted in one time-frequency map of model fits for each trial. We averaged these trial-specific fits separately for correct and incorrect trials and in the two ROIs and evaluated whether there was a difference between correct and incorrect trials at all time-frequency bins. To correct for multiple comparisons, we performed the same analysis 1,000 times by permuting the condition labels. At each permutation, we calculated the summed t-values of the significant differences between conditions with shuffled labels, resulting in a distribution of summed t-values under the null hypothesis. We then ranked the observed t-value with respect to this distribution to assess statistical significance. We only included subjects with at least 5 trials in each condition, leading to a total of 18 participants for VVS and 13 in PFC (paired t-tests were applied to the average time-frequency maps across conditions). Given the relevance of categorical information for PFC and VVS representations, trials in which the correct category of the cued item was reported were considered as correct, and trials in which subjects failed to retrieve the correct category were considered incorrect. This led to a total number of $M=13.22 \pm 9.12$ incorrect trials and $M=65.23 \pm 30.64$ correct trials in the VVS analysis (Mean \pm STD), and of $M=11.77 \pm 4.94$ incorrect trials and $M=60.26 \pm 26.55$ correct trials in the PFC analysis (Mean \pm STD).

These analyses revealed significant differences between correct and incorrect trials in VVS but not in PFC (Supplementary Figure 6): In the VVS, fits were higher for correct as compared to incorrect trials in BL-NET layers 1 ($p_{\text{corr}} = 0.014$), 3 ($p_{\text{corr}} = 0.028$) and 4 ($p_{\text{corr}} = 0.021$; Bonferroni corrected for 7 network layers). This was observed in three clusters of significant time frequency bins in the alpha-beta range (6-21Hz), which started after the onset of the cue and lasted for ~1s in each of the three layers. Notably, these clusters were

observed in the same time-frequency period where we observed significant EMS in the VVS. In the PFC, no significant clusters were observed in any layer (all $p > 0.35$). In summary, these results demonstrate the functional relevance of VVS representations during the maintenance period for task performance, which were substantially degraded in the incorrect trials as compared to the correct trials. Notably, the prioritization process in PFC appears to affect both correct and incorrect trials in the PFC, since no difference between these two types of trials was observed in this region. In the VVS, on the other hand, the representations of incorrect trials were distorted, and these effects occurred in a time period which partially overlaps with the significant EMS observed in this region.

Following the suggestion of the reviewer, we evaluated whether category and item-specific information could be extracted in the time period where we observed significant differences between correct and incorrect trials in the VVS. We computed an average RSM in the clusters observed in layers 1,3, and 4 in the VVS and correlated these average RSMs with the category model and the item model. We did not observe any significant correlations with either model in these clusters (all $p > 0.61$). These results are consistent with the lack of fit of the category model during the period of significant EMS originally reported in our manuscript.

We have included these results in the new Supplementary Figure 6, and we mention them in the results section (p. 8), where we write: *“In additional control analyses, we investigated the functional relevance of representations in VVS and PFC during the maintenance period (Supplementary Figure 6)”*.

There are two maintenance periods. I assume that all of these analyses are during the second maintenance period after the cue is presented. This should be clarified, but I think most people would probably assume this as well. However, it would be interesting to know if there are any meaningful representations during the first maintenance period. Are all three stimuli represented (perhaps serially) during this time?

All analyses reported are indeed from the second maintenance period (M2). We clarified this in the revised manuscript (p. 7-8). To investigate the representations of items during the first maintenance period, we performed the following analyses:

(1) We computed the fit of all DNNs during M1. Since there is no ‘cued’ item during this period, and to assess a possible sequential representation of each item as proposed by the reviewer, we computed the fits separately for each of the presented items (position 1, 2 and 3). These analyses did not reveal a significant fit during the M1 period for any of the items presented in the sequence (position 1: all $p_{\text{corr}} > 0.77$; position 2: all $p > 0.2$; position 3: all $p_{\text{corr}} > 0.51$).

(2) We performed a maintenance-maintenance similarity analysis (MMS) in which we calculated M1-M1 similarity between trials in which all items that were presented during encoding matched, versus trials in which all items are different. Results in this analysis revealed a lack of significant fits in all five frequency bands we tested (VVS: all $p_{\text{corr}} > 0.47$; PFC: all $p_{\text{corr}} > 0.71$, Bonferroni corrected for five frequency bands). While this analysis is sensitive to a “compound” representation of all presented items, it is not well suited to detect a possible sequential representation of the items.

(3) Finally, consistent with our original approach, we also computed an encoding-maintenance similarity analysis in the M1 period (EM₁S analysis) for five frequency bands (Theta, Alpha, Beta, Low Gamma, High Gamma). We computed EM₁S separately for items at the first, second and third position, to assess a possible relevance of the sequential order of representation of the items. In the VVS, we observed significant EM₁S for items encoded at the third position in the theta range ($p_{\text{corr}} = 0.025$, Bonferroni corrected for 5 bands); for items at the first and the second position, this effect only reached significance before Bonferroni correction (first position, $p = 0.028$; second position: $p = 0.027$). Similarly, we found that items encoded at the third position showed significant EM₁S in the beta range ($p_{\text{corr}} = 0.005$), while we observed EM₁S at a trend level in position 2 ($p_{\text{corr}} = 0.06$), and no effects in position 1 ($p = 0.134$). We also observed significant EM₁S in the gamma band for items in position 2 ($p_{\text{corr}} = 0.015$), but not in position 1 or 3, which did not reach significance at an uncorrected level ($p > 0.11$). Following up on these results, we performed an analysis in which we grouped together the low frequencies (theta, alpha and beta; 3-29Hz) and assessed EM₁S in the VVS. Results revealed marked EM₁S in this broad frequency range in the VVS for items encoded in all positions (position 1: $p_{\text{corr}} = 0.005$; position 2: $p_{\text{corr}} = 0.015$; position 3: $p_{\text{corr}} = 0.005$), confirming that representations of encoded items are maintained in the VVS during the M1 period, before the presentation of the cue. Our EM₁S analysis in the PFC did not show any significant result in any of the five frequency bands (all $p_{\text{corr}} > 0.53$).

Together, these findings highlight that neural patterns encoding category-specific representations were reinstated during the maintenance period in the VVS in the 3-29Hz range, while this was not the case in the PFC. We report these new results in Supplementary Figure 2 and Supplementary Note 1. In addition, we have modified Figure 2 of the manuscript and included separate labels for the two EMS analyses: “EM₁S” and “EM₂S”.

The authors also examine the representations of stimulus information in the separate DNN architectures. In deeper layers, categorical distinctions are more pronounced, although in the feedforward model this is because within category similarity increases whereas in the recurrent models across category similarity decreases. In both cases, categorical distinction is apparent in the deeper layers. Is this surprising? Previous studies of such DNNs in image classification have demonstrated that conceptual (or categorical) information is often better represented in deeper layers.

When trained to classify images, i.e., to assign object labels to pictures, DNNs form layered representations at various degrees of visual abstraction. In that sense, the emergence of higher-level visual representations and the segregated representation of images according to their classes in deep layers of convolutional DNNs is neither novel nor surprising. This segregation however can be achieved (1) by grouping together items belonging to the same category, (2) by separating items belonging to different categories, or (3) by a combination of these two processes. To characterize how our networks represented stimuli, we separately computed within-category and between-category correlations in all DNN layers and observed a remarkable dissociation in the representations of our feedforward and recurrent models. Recurrent architectures exhibited decreasing between-category correlations for higher layers without any

increases in within-category correlations; in contrast, both phenomena were observed in our feedforward model. While we acknowledge that this might not be a universal property of recurrent architectures, we felt it was important to show that this is the case in our data, given the relevance of this dimension of representation (category), and the relationship between same and different categories in our task. To clarify the rationale of including this separate analysis, we modified the methods section of the manuscript (Methods section, p. 42) where we write:

“The segregated representation of images according to their classes in deep layers of convolutional DNNs is a well-documented phenomenon (Cichy & Kaiser, 2019; Kriegeskorte, 2015). This segregation however can be achieved by (1) grouping together items belonging to the same category, (2) separating items belonging to different categories, or (3) a combination of these two processes. To distinguish among these possibilities, we separately computed within-category and between-category correlations in all DNN layers. Results are presented in Figures 3C, 4D and 5D, and in Supplementary Figure 5”.

Reviewer #2 (Remarks to the Author):

Estefan and colleagues examine neural representations during a visual working memory task using intracranial EEG recordings and representational similarity analysis. Various categorical and neural network based representational analyses are conducted, revealing that memory representations in the PFC involve a transformation of format, which is revealed only interrogated with a recurrent deep neural net. Overall, this study is examining important questions about how working memory representations are transformed during maintenance, and using state-of-the-art RSA approaches to do so. Further remarks are below, mainly relating to the analysis description and statistical methodology.

We thank the Referee for the constructive comments and suggestions for improvement. Below, we present our responses and indicate where new information can be found in the manuscript.

1. The first analysis is an involves a “simple categorical model” that examines an item-by-item similarity matrix and correlates this with a categorically coded similarity matrix. This reveals significant categorical coding during encoding VVS and PFC, but no such coding during maintenance. Next, to “evaluate whether transformed activity patterns from encoding reoccur during the maintenance period, we performed a category-specific pattern similarity analysis.” How does this “category-specific” analysis differ from the “simple categorical” model. From the text, hard to tell—and the analyses are described similarly: “this analysis involved contrasting correlations of items belonging to the same category with correlations of items from different categories”. And this line: “EES and EMS analyses test for re-occurrence of category-related neural activity patterns from different encoding periods” hardly clarifies the situation, since the “simple categorical” analysis couldn’t work if category-related neural activity didn’t “reoccur”. A figure is then referred to (2A, 2C) but now the latter technique is called a “mnemonic scheme”. (One must zoom in 300% or 400% to identify the little images on the left of the graph).

We apologize for the lack of clarity regarding the description of the contrast-based (EMS) vs. model-based (RSA) analyses in our manuscript. We note that while both analyses study the presence of category representations in the iEEG data, they differ in two important aspects and have been recently employed as complementary approaches to investigate neural representations (Reagh and Ranganath, *Nat Commun* 2023).

The first distinction relates to the level at which the two methods assess similarities in the representations. While the model-based analysis captures differences in the representational geometry of stimuli (it correlates RSMs of neural data with RSMs of models, a second level analysis), the contrast-based analysis directly correlates neural patterns and is thus sensitive to reoccurrence and transformation of specific neural features. For example, the same representational distances (and thus RSMs) may depend on one particular brain region (i.e., set of electrodes) during encoding and a different brain region during maintenance, leading to significant RSM-based similarities in the absence of encoding-maintenance similarity (EMS). The model-based analyses correlate representational distances during either encoding or maintenance with distances in particular RSM models, and does not directly compare levels of model fits between encoding and maintenance. Thus, in a strict sense, this approach does not directly test the reoccurrence of the representational geometry, but whether a particular geometry is

present during a specific time period. In our analyses of the VVS, we observed significant EMS without matching of the category-model RSM during maintenance. This indicates that the neural features during encoding and maintenance of items from the same category are closely matched, while the representational geometry of stimuli is not sufficiently aligned with the category model during maintenance. While in our original analyses, we assessed model fits at every time point during encoding and maintenance, in additional analyses presented in this revision we now compared the representational geometry of the neural data at different time points by contrasting the different levels of fit. These analyses are thus informative about the “reoccurrence” of a particular representational geometry (see below).

A second difference relates to the specific neural features in each analysis. The RSM-based analysis was conducted separately for each individual frequency in the 3-150Hz range, which allowed for a fine-grained assessment of the contribution of individual frequencies. By contrast, feature vectors in the EMS analysis included power values across several individual frequencies within particular frequency bands, and thus contained higher variance. This was necessary in order to avoid the occurrence of 3D-clusters in the EMS analyses (encoding time x maintenance time x frequency), which make cluster-correction for multiple comparisons very difficult. To corroborate that our results were not affected by differences in the frequency features that we selected in each analysis, we conducted the (RSM-based) category model analysis in the same frequency bands as the EMS analysis (theta, alpha, beta, gamma, low gamma). Our results in the RSM-based band-specific analyses were consistent with the frequency-resolved analysis reported in Figure 2B: A significant fit of the category model during encoding in all frequency bands was observed in the VVS, and a more restricted fit in the PFC in the beta band ($p_{\text{corr}} = 0.015$, Bonferroni corrected for 5 bands; Supplementary Figure 13A). During maintenance, we did not observe any significant fit in VVS or PFC in any band (VVS: all $p > 0.3$; PFC: all $p > 0.275$; Supplementary Figure 13B).

In general, the application of both contrast-based (EMS) and model-based (RSM) analyses provided consistent (though not identical) results. During encoding, both analyses confirmed the relevance of categorical information in VVS and PFC, demonstrating it at the level of the representational geometry of stimuli (RSM-based analysis) and of the neural features (contrast-based analysis, EMS). Consistent results in the two methods were also observed during maintenance in the PFC (a lack of fit of the category model, and an absence of EMS in this region). Critically, however, the two approaches yielded different results in the VVS during maintenance: The category model explained representations in the VVS during encoding but not during maintenance, while the EMS approach revealed significant maintenance of neural activity patterns from the encoding stage (Figure 2D). These results suggest that the VVS patterns during maintenance only contain partial information from the encoding period, and that the representational geometries were transformed (see also our response to the first comment of Reviewer 1). To further corroborate this interpretation, we extracted a mean neural RSM in the time-frequency periods where we observed the most pronounced EMS effect, i.e., from 700ms after cue onset to 1.9s in the theta (3-8Hz) frequency range. We compared this average RSM with the average RSM observed during encoding in the theta band (the full encoding time period was taken during which significant fits were observed). Visual inspection of the two RSMs revealed marked differences in their representational

geometry, with a more regular structure in the off-diagonal values for the encoding as compared to the maintenance RSM, indicating the presence of category-specific representations. To quantify these differences, we correlated the two neural RSMs in each subject independently and contrasted the group-level rho values against zero. Results revealed that the two RSMs were not correlated ($t(25) = 1.19$, $p = 0.245$), suggesting a change in representational geometries from encoding to maintenance. In addition, we computed the Category Clustering score (CCI; within – between correlations) during the two time periods. Results revealed that the CCI was significantly higher during encoding as compared to maintenance ($t(25) = 3.66$, $p = 0.0012$), consistent with our original results (Figure 2B), and the interpretation of a representational transformation into a less categorical format in the VVS from encoding to maintenance.

We clarified the description of the two approaches in the Methods section, specifying their differences (p. 39) Following the reviewer’s suggestion, we emphasized that both methods can be informative about the reoccurrence of neural representations, although at different levels. While the model-based analysis can be employed to test for the presence of a particular representational geometry during the two stages, it does not test for the reoccurrence of neural features. The contrast-based analysis, on the other hand, directly correlates neural patterns at different time points.

In p. 37 we now write: *“Please note that while the contrast-based and the model-based RSA analyses have been employed as complementary approaches to investigate neural representations (Reagh & Ranganath, 2023), they differ in two important aspects. The first distinction relates to the level at which the two methods assess similarities in the representations. While the model-based analysis captures differences in the representational geometry of stimuli (it correlates RSMs of neural data with RSMs of models, a second level analysis), the contrast-based analysis directly correlates neural patterns and is thus sensitive to reoccurrence and transformation of specific neural features. For example, the same representational distances (and thus RSMs) may depend on one particular brain region (i.e., set of electrodes) during encoding and a different brain region during maintenance, leading to significant RSM-based similarities in the absence of encoding-maintenance similarity (EMS). The model-based analyses, on the other hand, correlates representational distances during either encoding or maintenance with distances in particular RSM models, and does not directly compare levels of model fits between encoding and maintenance. Thus, in a strict sense, this approach does not directly test the reoccurrence of the representational geometry, but whether a particular geometry is present during a specific time period. To test for the reoccurrence of a particular representational geometry in the model-based analyses, we directly contrasted the different levels of fit during particular time periods using paired t-tests. A second difference relates to the specific neural features that were included in each analysis. The RSM-based analysis was conducted separately for each individual frequency in the 3-150Hz range, which allowed for a fine-grained assessment of the contribution of individual frequencies. By contrast, feature vectors in the EMS analysis included power values across several individual frequencies within particular frequency bands, and thus contained higher variance. This was done in order to reduce dimensionality of the representational patterns and facilitate the process of multiple comparisons correction (see below). To corroborate that our results were not affected by differences in the frequency features that we selected in each analysis, we conducted the (RSM-based)*

category model analysis in the same frequency bands as the EMS analysis (theta, alpha, beta, gamma, low gamma). Our results revealed a significant fit of the category model during encoding in all frequency bands in the VVS, and a more restricted fit in the PFC in the beta band ($p_{corr} = 0.015$, Bonferroni corrected for 5 bands; Supplementary Figure 13A). During maintenance, we did not observe any significant fit in VVS or PFC in any band (VVS: all $p_{corr} = 1$; PFC: all $p_{corr} = 1$; Supplementary Figure 13B).”

We also modified panels A and C of Figure 2 and changed the subheadings to “Model-based RSA - Category Model RSM” and “Contrast-based RSA – Category-specific similarity” following the reviewer’s remark.

2. Having found no evidence for categorical coding in the PFC during maintenance, the manuscript turns to a series of DNN analyses, one using feedforward model (AlexNet), and two analyses with recurrent DNNs. This is indeed an important and worthwhile approach, but the presentation is not statistically convincing. The goal here is not to simply test the predictive performance (but see below for remarks on that) of several neural networks on iEEG activity during maintenance, but rather to say something about mechanism, namely recurrence. One issue, however, is that recurrence is not the only thing that differs between AlexNet and BL-net/cor-NET-RT. They have other differences, including number of layers, training regimes, parameters, etc. One should like to say: BL-Net outperforms AlexNet due to its recurrent properties-- but this requires an "all other things being equal" comparison, which is not the case here. Ideally, one would have a model with the same architecture, same training set, training parameters, etc. where one had recurrence and one no recurrence.

We thank the reviewer for raising this important point. We acknowledge that the employed architectures differ in several aspects, including the number and type of layers, trainable parameters, information processing dynamics, etc. For example, BL-NET employs convolutional, batch normalization, and ReLU layers, while AlexNet incorporates additional layers like dropout, max pooling, and fully connected layers. Furthermore, CorNet-RT exhibits distinct information propagation across layers compared to BL-NET due to the absence of inter-layer information flow before information has been processed in each individual layer. Despite these variations, it is crucial to recognize the shared aspects of these DNNs. All three are convolutional DNNs that learn visual filters in a hierarchical manner across layers. They share the same objective function (classification), learning algorithm (backpropagation), and training data (ImageNet). Our choice of AlexNet as the feedforward model was due to its widespread adoption in cognitive neuroscience research, particularly within the domain of iEEG, which allows for direct comparison with prior studies (e.g., Liu et al., *PNAS* 2020; Liu et al., *Sci Adv* 2021). Regarding our recurrent models, it is worth emphasizing that our work represents the first attempt to leverage recurrent architectures for modeling VWM in iEEG research. In that sense, we opted for convolutional recurrent networks that are widely known in the cognitive and computational neuroscience community and have been employed to explain important features of visual perception (Spoerer et al., *Plos Comp Biol* 2020; Kubilius, *bioRxiv* 2018). Notably, these two architectures have very different features and information processing dynamics, yet we have shown that they converge in their results.

We agree with the reviewer regarding the need to control potentially confounding factors when isolating the specific effect of recurrence. Matching the number of trainable parameters is a common approach for performing controlled comparisons between networks (Spoerer et al., *Plos Comp Biol* 2020). In general, adding recurrent connections to a feedforward model increases its number of trainable parameters. We thus applied three parameter-matched variants of BL-NET, all lacking lateral recurrence but including additional parameters: BK-NET, which increases the size of the convolutional kernels; BF-NET, which increases the number of feature maps; and BD-NET, which increases the depth of the network (number of layers). Additionally, for completeness, we included the non-recurrent version of CorNet-RT, named CorNet-Z, which contains a different number of parameters but maintains the number of layers of CorNet-RT.

Focusing on representations in the respective last layer in each of these control architectures (layer 7 in BK-NET and BF-NET, layer 13 in BD-NET, and layer 4 in CorNet-Z), we did not observe any significant fit with representations in non-recurrent networks in the PFC (BK-NET all $p_{\text{corr}} > 0.196$; BF-NET, all $p_{\text{corr}} > 0.063$; BD-NET, all $p_{\text{corr}} > 0.21$; CorNet-Z all $p_{\text{corr}} > 0.9$). In the VVS, on the other hand, we did observe a significant fit in BF-NET layer 4 ($p_{\text{corr}} = 0.042$), while all other layers did not show significant fits (all $p_{\text{corr}} > 0.056$). This was observed in a time period from 2.1s to 2.9s, overlapping in time with the clusters observed in layers 4, 5 and 6 of the original BL-NET analysis. The other control models did not show significant fits to VVS representations in any layer (all $p_{\text{corr}} > 0.066$). The CorNet-Z model did not show a significant fit in any layer (all $p_{\text{corr}} > 0.23$). Results of these control analyses are presented in the new Supplementary Note 6.

These control analyses, together with the main findings reported in our study (significant fits in PFC locked to the retro-cue for both BL-NET and CorNet-RT, Figure 4H and 5G), suggests that recurrent computations play an important role in information processing dynamics in the PFC. On the other hand, they also suggest that VVS representations during maintenance can be explained by computations in feedforward models as well. In our revised manuscript, we added these important results in p. 14, where we write: *We performed control analyses using parameter-matched versions of our recurrent architectures to evaluate the effect of recurrency, while isolating other possible confounding variables (Supplementary Note 6). Results suggest that recurrent computations are indeed crucial for tracking cognitive representations in PFC, because the fit observed with the recurrent networks could not be found in any of the feedforward models we tested. They also show that recurrency may play a relatively less prominent role in the VVS (Supplementary Note 6)*".

We also highlight this point in the discussion section (p.18): *"During maintenance, [...], the two architectural families strongly differed in their fit to the neural data: the AlexNet was unable to capture representations in either region, while BL-NET and corNET-RT matched representations in both VVS and PFC (Figure 4H and 5G). Control analyses using parameter-matched versions of the BL-NET without recurrency indicated that recurrent computations are indeed crucial for tracking cognitive representations in PFC, while they appear to play a relatively less critical role in the VVS (Supplementary Note 6). Together, these results demonstrate that only recurrent architectures can explain the representational geometry of stimuli during VWM prioritization in PFC, while a feedforward architecture and a simple model of category information do not provide good fits"*.

3. If the recurrent models better model PFC activity during maintenance than the other models (categorical, Alexnet, etc.) then one should like to see a statistical evidence via model comparison. For example, AlexNet did not match neural representations in VVS, all p s > .056. But BL-Net was deemed a success with three p -values < .05 (.035, .035, .014). Is this really different level of predictive performance, though? The three analyses are presented independently, and plotted in successive grids of (very small) heatmaps in separate figures. Here is a case where an aggregate measure (to avoid contrasting every cell of the heatmaps) of cross-validated performance could be statistically compared between all models, starting with the simple categorical, with either a non-parametric test or an ANOVA-style analysis. Otherwise, we don't really know if the differences are reliable.

We performed additional analyses to compare the predictive performance of BL-NET, AlexNet and the Category model using an aggregate measure of model fits following the suggestion of the reviewer. In each analysis, we selected only one layer of our DNNs and one particular time-frequency period, and we averaged levels of fits in each model. We performed statistical comparisons separately for the PFC and the VVS, given the distinct temporal features our two ROIs showed during the maintenance period. Results are presented in the new Supplementary Note 4 and Supplementary Figure 7.

In the PFC, we selected the last layer of our DNNs and focused specifically on the prioritization period (0-800ms after cue onset during maintenance) in the high-beta frequency range (21-29Hz). We constructed an ANOVA model with “RSM model” as a factor and included the average fits during this period for each of our different model RSMs. Results revealed a main effect of RSM model, indicating significantly different matching to the different models in PFC ($F(14) = 4.9$, $p = 0.0149$). Post-hoc analyses revealed that this effect was mostly driven by differences in fits between BL-NET and AlexNet ($t(14) = 3.01$, $p_{\text{corr}} = 0.0238$), while no significant differences were observed between BL-NET and the Category model ($t(14) = 0.91$, $p_{\text{corr}} = 0.64$), nor between AlexNet and the Category model ($t(14) = -2.07$, $p_{\text{corr}} = 0.133$, Tukey-Kramer corrected for three comparisons). We note that we considered a prioritization time period of the same length of that of stimulus presentation during encoding (i.e., 800ms after cue onset), but we observed similar results in the more restricted period of cue presentation (0-500ms; Main effect: $F(14) = 4.05$, $p = 0.029$, BL-NET vs. AlexNet = $t(14) = 2.66$, $p_{\text{corr}} = 0.045$; BL-NET vs. Category model: $t(14) = 0.78$, $p_{\text{corr}} = 0.72$; AlexNet vs. Category Model: $t(14) = -1.91$, $p_{\text{corr}} = 0.17$; Tukey-Kramer corrected for three comparisons).

In a complementary analysis with a more comprehensive and data-driven approach, we contrasted BL-NET and AlexNet fits during a more extended time period of item prioritization and across all frequencies (0-1s after cue onset during maintenance, 3-150Hz range). We used the same temporal resolution as in the original analyses, i.e., 100ms steps. Cluster-based permutation statistics were applied to correct for multiple comparisons by shuffling the condition labels (network identity). This analysis revealed that representations in the last layer of BL-NET provided significantly better fits than representations in the last layer of AlexNet in a cluster from 200 to 800ms after the onset of the cue in the 20-27Hz frequency range ($p_{\text{corr}} = 0.01$; Supplementary Figure 7). We performed the same analysis now contrasting BL-NET layer 7 with AlexNet layer Fc7 instead of Fc8. We chose the penultimate fully connected layer instead of the very last

layer of AlexNet because this layer contains a relatively higher number of features, which increases variance and has been selected in other studies (e.g., Tang et al., *PNAS* 2018). Again, we observed a cluster of significant time-frequency bins from 400ms and until 800ms in the high beta range (21-26Hz, $p_{\text{corr}} = 0.018$).

In the VVS, we selected the end of the maintenance period (2-3s after cue onset) in the alpha frequency range (9-12Hz) and compared BL-NET (layer 5), AlexNet (layer 5), and the Category model, again using an ANOVA. Results revealed a significant main effect of model in this time-frequency period, indicating different degrees of matching to the 3 models in VVS as well ($F(25) = 6.65$, $p = 0.0027$). Post-hoc comparisons indicated that for this ROI, differences in the model fits were driven by differences between BL-NET and the Category model ($t(25) = 2.71$, $p_{\text{corr}} = 0.03$) and between AlexNet and the Category model ($t(25) = 2.65$, $p_{\text{corr}} = 0.035$), while the difference between BL-NET and AlexNet was not significant in VVS ($t(25) = -0.16$, $p_{\text{corr}} = 0.98$; Tukey-Kramer corrected for 3 comparisons). Notably, the lack of significant difference between the AlexNet and the BL-NET is consistent with the results reported above that VVS representations could also be fit by representations in the BF-NET, suggesting they rely less on recurrency than those in PFC.

Together, these results demonstrate that our models differently explained the data, and that differences were more pronounced between BL-NET and AlexNet in the PFC during item prioritization, and between BL-NET and the Category model in the VVS during the late maintenance period.

4. Comparisons between PFC and VVS are made implicitly at times, but it's important to keep in mind that there are different sample sizes ($N=28$, $N=16$), and therefore differences in power.

We thank the reviewer for raising this point. An important objective of our manuscript is to characterize how representations in PFC and VVS differ during VWM maintenance and prioritization, and for this purpose it is indeed crucial to consider the differences in sample size and, as a result, statistical power in our two ROIs. One of the main analyses in which we implicitly compared VVS and PFC representations is the EMS analysis, in which we observed different results in the two regions. Indeed, the significant EMS across all frequency bands in the VVS and a lack of this effect in PFC was one of the key pieces of evidence which led us to conclude that PFC representations were transformed more strongly than those in the VVS. We matched the number of subjects in the two regions by randomly removing a total of 11 subjects in the VVS analysis 1,000 times in order to assess the stability of the effect in the VVS under similar levels of statistical power. This led to a total of $N=15$ subjects in the VVS analysis (note that one subject with only one electrode had already been excluded in the PFC analysis). We focused on the theta band where the most prominent EMS effects were observed in the original analysis. To avoid statistical noise and random fluctuations, we took the average EMS difference (same minus different categories) in the time period of 1.5s after cue offset during maintenance and for all time points during encoding (where most of the effects were observed in the analysis including all subjects) at every random subsampling of subjects. We then contrasted this average difference value against zero at the group level to assess statistical significance. These analyses confirmed significant category-level EMS effects in the VVS with participant numbers matched to those in the PFC in 95.8% of the random subsamples (Supplementary Note 7 and Supplementary Figure 11A).

In addition to the EMS analysis, we also correlated representations in neural and model RSMs (BL-NET analysis) during encoding and maintenance by randomly subsampling from the total number of subjects in the VVS 1,000 times. We specifically focused on the levels of fits in the last layer of the network during encoding and maintenance. During encoding, we selected the time-frequency cluster observed in layer 6, which covers the whole time period of image presentation (0-800ms), and frequencies in the 3-150Hz range. We took the average levels of fit in this cluster and contrasted them against zero at the group level. Results revealed that the average fits with the BL-NET during the encoding period was significant in 100% of the random VVS subsamples (Supplementary Figure 11B). During maintenance, we specifically focused on the cluster observed in layer 6 of BL-NET in the VVS analysis, and again took the average levels of fit in this cluster at every subsample. Results revealed that fits with maintenance were significant in this analysis in 100% of the random subsamples (Supplementary Figure 11C).

We included these additional analyses in Supplementary Note 7 and Supplementary Figure 11, and modified the manuscript to highlight the differences in sample size between our ROIs in the Methods section (p.33), where we write: *“The different number of subjects and channels in our two ROIs implies different levels of statistical power. Since an important objective of our study was to characterize how representations in PFC and VVS differ during VWM maintenance and prioritization, we confirmed that our main findings replicate when matching statistical power in VVS and PFC through several control analyses (Supplementary Figure 11 and Supplementary Note 7)”*.

5. We read: "... studies (Liu et al., 2020, 2021) established a crucial role of transformed representational formats, particularly abstract representational formats devoid of specific sensory information, in VWM maintenance. Based on these insights, we hypothesized that the PFC may contain categorical information in an abstract representational format after presentation of the retro-cue." This is not entirely clear, since a categorical format could be entirely abstract, and therefore picked up by the "simple categorical model". If "abstract" here simply means "non-sensory", then it is still the case that items in the same category share all non-sensory features. I think some work defining terms would help the reader here.

We thank the reviewer for this suggestion and apologize for the lack of clarity. We acknowledge that the particular phrasing we employed in this sentence is misleading because categorical information is abstract by definition. In our original hypothesis, we expected prioritized representations to map with deep layers of DNN models and/or with the category model RSMs. What best distinguishes these two models, however, is not their level of 'abstraction', but other aspects, such as the presence of subtle distinctions among stimuli that reflect high-level visual information, and a particular representational geometry of within-category and between-category correlations in the DNN layers but not the category model (see also our response to the first comment of reviewer 3). Therefore, we now label the format of representations in the PFC after the cue 'prioritized' instead of 'abstract'. We changed the manuscript accordingly throughout. We also modified the lines highlighted by the reviewer (p. 8), which now read:

“Based on these insights, we hypothesized that the PFC might represent stimuli in a representational format devoid of low-level sensory information that maps to deep DNN layers during the prioritization period.”

Reviewer #3 (Remarks to the Author):

In this study, using multi-item working memory task with a retro-cue, Pacheco Estefan and colleagues examined how executive systems select relevant items within working-memory space in the brain and how this selection process affects neural representational formats in ventral visual stream (VVS) and prefrontal cortex (PFC) in human participants. Based on the observation that only in PFC, the category specific information which was originally present in neural activity during encoding did not re-occur during maintenance (post retro-cue), the authors suggested that in PFC, the task representation underwent significant transformation between before and after the presentation of the retro-cue, changing from signals representing category information to a higher-order abstract representation, putatively reflecting the prioritization process. The authors also reported that the recurrent DNN representations matched PFC representations in beta band following the presentation of the retro-cue, which, the authors suggested, reflected the top-down control over the visual WM.

Neural activity data appear to be recorded in sufficient volume and preprocessing is adequate. The method itself was clearly explained and easy to understand. However, as you can detail in the comments below, the main argument of this paper is not well supported for several reasons: First, the authors did not present sufficient evidence to support the aforementioned claim at the level of both neural and DNN-based analyses. Second, there appear to be fundamental flaws in some of the key analyses (e.g., ignoring the effect of the factor, retro-cue type on neural, especially prefrontal, activity, and using DNNs which did not actually perform the behavioral task), potentially invalidating the interpretation of the key observations. Third, and most importantly, no evidence was presented at all in both neural and model analyses as to what exactly the "abstract representation format" of category information and "prioritization process" in the prefrontal cortex are.

The conclusions drawn from the results are scarce, and the previous studies using similar behavioural paradigms should incorporate the key findings of this manuscript, with more detailed information on what has happened in the post-retro-cue epoch. (e.g., Everling et al., Nat. Neurosci., 2002; Panichello and Buschman, Nature, 2021; Piwek et al., PLoS Comput. Biol., 2023; Meyers et al., TiCS, 2017). Below is a list of major comments and specific comments:

We thank the referee for their comprehensive and detailed review of our work and the suggestions for improvement. We performed several additional analyses to address the reviewer's concerns, and added further evidence supporting our main conclusions. We also analyzed the effects of the additional factors mentioned by the reviewer, further elucidated and described the representational format and the prioritization process in PFC, and clarified the novelty of our findings as compared to previous studies. Below, we provide a point-by-point response to the issues raised by the reviewer and indicate where changes have been made in the manuscript.

Major comments:

1. The paper argued that PFC contains categorical information in an abstract representational format after the presentation of the retro-cue. However, what exactly is the "abstract representational format" of categorical information was not explained at all even at the level of neural activity. Rather, the authors only showed that the correlation between rDNN RSM and PFC RSM (in beta band) is significant at the deep layer of these models (Figures 4 and 5), and argued

that this was the key evidence supporting the presence of “abstract” category information in PFC in the maintenance period (which seemed to be different from category specificity observed in the earlier, encoding period). This correlation, however, does not explain what the aforementioned “abstract representational format” of categorical information is, nor does it provide evidence that the information representation of PFC immediately after the retro-cue had an abstract representational format which was transformed from and thus different from the category representation, per se in the encoding epoch.

In neural information processing, abstraction usually means dimensionality reduction (e.g., Bernardi et al., Cell, 2020); the process of abstraction extracts the essential data structure by reducing the amount of information to less than originally provided. In this study, as the authors noted, category information observed in the encoding phase did correspond to the abstract format of representation (i.e., representing visual objects not by their low-level visual features, but by semantic categories). However, as a main argument of this paper, the authors stated that following the retro-cue presentation, PFC started to represent “categorical information in an abstract representational format” (lines 240-241). The same claim was made repeatedly in the paper, i.e., that after the retro-cue, PFC began to represent different and perhaps higher-order information than the category representation observed in the encoding period. Then, what exactly does “categorical information in an abstract representational format” mean? The authors should clarify this point with additional neural analyses before moving on to analysis using artificial neural networks.

We thank the reviewer for this remark. First, we acknowledge that our terminology may not accurately reflect the different representational formats we observed (see also our response to the last comment of reviewer 2). We agree that categorical representations, which we observed during encoding in both VVS and PFC, are by definition abstract. We note that we do not claim that the fit we observed in the PFC with the deepest layers of our recurrent DNNs reflects ‘more abstract’ – or ‘higher order’ – representations than during the encoding period, and apologize that this was not clear in our manuscript. Importantly, however, we did find that the format of representations differs between encoding and prioritization in the PFC, matching different computational models, and in the revised manuscript we have characterized these differences in more detail, both at the level of neural geometries and regarding their correspondence to the DNN layers. We also agree with the reviewer that the recurrent DNN’s deep layer fit by itself is not sufficient evidence of a transformation of the representational format between encoding and prioritization, and clarify the additional evidence supporting that conclusion. We will further elaborate on these two issues below.

First, we would like to underscore the substantial differences in the representational geometries of the category model and our recurrent DNNs architectures. The category model solely distinguishes items based on their categories, and codes items belonging to the same category with a 1 and items of different categories with a 0. Therefore, the category model assumes that the geometry of within-category and between-category correlations is not structured (or is the result of noise – we addressed this point below, see comment #6). The DNN layers, on the other hand, capture nuanced relationships among stimuli which relate to the visual properties of images. DNNs trained for image classification extract class information from visual inputs, and representations in these architectures reflect fine-grained visual features of images at various different

levels. In particular, the deep layers are sensitive to high-level visual information, e.g., the presence of objects or object parts, as has been extensively described (Yamins & DiCarlo, *Nat Neurosci* 2016; Kriegeskorte, *Annu Rev Vis Sci* 2015). This high-level visual information is what we referred to as “abstract” in our original manuscript, but we removed this potentially misleading term (actually, as we describe below, the dimensionality of this high-level visual information during the prioritization period is higher than the dimensionality of the purely categorical information during encoding). The RSMs derived from the category model and from recurrent DNNs thus do not only constitute different hypotheses about the geometry of VWM representations in the brain, but model different aspects of these geometries. Both models can be fit to the full geometry of the neural data and are not necessarily orthogonal, but only the DNNs can be informative about the representational geometry of individual items within and between categories.

Our data shows that the category model is a good model of PFC representations during encoding but not during prioritization, while the reverse is true for the BL-NET. This suggests that the format of prioritized information in PFC is transformed from encoding to maintenance, and is not well explained by purely categorical information during the prioritization period. This transformation of representational formats is also supported by the lack of encoding-maintenance similarity in the PFC, which differs from the significant encoding-maintenance similarity in VVS (which we labeled as “mnemonic coding scheme”).

To further characterize the transformation in PFC, we performed several additional analyses, focusing on the comparisons of neural RSMs during encoding and prioritization. First, we extracted an average neural RSM during the time periods where we observed the significant fits with the category model (encoding) and BL-NET layer 7 (prioritization phase, Supplementary Figure 1B). We then directly correlated these two RSMs and observed they are not significantly correlated ($t(14) = 0.717$, $p = 0.485$). These results are conceptually similar to our previous analysis of encoding-maintenance similarity (which was not significant in PFC, as mentioned above). However, they are analytically complementary, because they are based on correlations of neural RSMs during the time periods that matched the category model (during encoding) and the BL-NET model (during maintenance). They thus demonstrate that the two representational geometries (i.e., formats) are indeed different, rather than only showing differences in the neural features that constitute these RSMs, which are correlated in the encoding-maintenance similarity analysis. These results thus provide additional support to our conclusion of a transformation of representational formats in PFC.

In addition, we performed three novel analyses to characterize the neural coding scheme in PFC and its change from encoding to maintenance. First, we tested whether average correlations within these RSMs differed between encoding and maintenance. Higher correlations of items during maintenance may point towards clustering of representations in representational space, while lower correlations would reflect the opposite, i.e. representations in a more widely spread space. We found a trend for higher between-category correlations during maintenance as compared to encoding ($t(14) = -2.12$, $p = 0.053$), suggesting that PFC representations occur in a smaller representational space. This effect was not observed for items from the same category ($t(14) = -0.185$, $p = 0.856$).

Second, we assessed the variance among correlations, again during encoding and maintenance. Higher variances would reflect less uniform (i.e., more distinctly organized and thus lower dimensional) distributions of items, while lower variances would correspond to an opposite pattern. We conducted analyses 1 and 2 separately for items of the same category and for items of different categories. Importantly, since representations of individual items are not considered at all in the category model, only the DNNs provide a model of them. This analysis showed that the variance of between-item correlations decreased significantly from encoding to maintenance, both for items from different categories ($t(14) = 5.87$, $p = 4.05e-05$) and from the same category ($t(14) = 5.37$, $p = 9.89e-05$). This result indicates that despite the overall (by trend) higher correlations among PFC representations during maintenance – i.e., representations in a smaller representational space – PFC representations during maintenance are more uniformly distributed in this space.

Third, we performed multi-dimensional scaling (MDS) on the RSMs during encoding and maintenance at various levels of dimensionality and computed the stress value of the MDS projections. Stress (a.k.a. Stress-1) is a metric of the *goodness of fit* of a particular MDS projection that reflects how well a lower-dimensional embedding – with a specific dimensionality – reflects the structure of the original (high-dimensional) data. Stress values are low if the data can be relatively well embedded in lower dimensions, and high if the embedding is less accurate. We performed the MDS analysis for all dimensions in the 1-60 range (corresponding to the size of the trial-averaged RSM in our data). For every subject, we converted the RSMs into distance matrices (1-correlation), performed MDS and computed stress in each time period. Given that the stress metric is sensitive to the number of dimensions of the distance matrix, we randomly removed items in RSMs to match the number of the condition with less items (some subjects had less than 60 trials during the maintenance period because some trials were removed during artifact rejection). This was done 100 times to corroborate that the results were not affected by the specific random selection of trials. We subsequently compared the group level stress values during encoding and maintenance for every dimension independently, and assessed regions of contiguous dimensions with significant differences between the two time periods. We applied cluster-based permutations statistics to control for multiple comparisons correction, by randomly shuffling the condition labels (i.e., encoding and maintenance) in each subject independently 1,000 times and recomputing the condition differences. At each permutation, we summed the t-values of the largest cluster of significant dimensions, resulting in a distribution of t-values expected by chance. We ranked the observed t-values with respect to this null distribution to assess statistical significance. This analysis revealed that stress values were systematically lower during encoding as compared to prioritization in a cluster of significant dimensions (from 4 to 33 dimensions; $p = 0.0148$; Supplementary Figure 1A), indicating a higher dimensionality of representational formats during prioritization than encoding.

Taken together, these results further characterize the change from a purely categorical representation during encoding to a representation that matches the fine-grained architecture of the BL-NET during prioritization: PFC representations occur in a smaller representational space (higher overall correlation), occupy less clustered regions in this space (lower variance of correlations), and rely on a higher-dimensional representational geometry.

Our findings indicate that the BL-NET provides additional information on the representational geometry in PFC beyond a purely categorical representation. However, this could rely on the fine-grained differences of either the within-category correlations or the between-category correlations of individual exemplars (or both). We next attempted to distinguish between these different scenarios, and conducted additional analyses that further characterize the representational format during the prioritization period using the DNNs. We correlated neural RSMs in the PFC with the RSMs extracted from the last layers of BL-NET and AlexNet during prioritization, separately for within-category and between-category correlations. We observed a significant fit of the between-category correlations of RSMs from the BL-NET and neural data in the PFC ($t(14) = 3.69$, $p = 0.0024$), while this was not true for the AlexNet ($t(14) = 1.61$, $p = 0.13$). None of our models could explain the structure of within-category correlations (BL-NET: $t(14) = 0.42$, $p = 0.678$; AlexNet: $t(14) = -0.18$, $p = 0.86$). The results of the same analysis performed during encoding confirms our initial observation that neither BL-NET or AlexNet are good models of activity in the PFC during this time period (BL-NET within-category correlations: $t(14) = -0.37$, $p = 0.71$; BL-NET between-category correlations: $t(14) = -1.84$, $p = 0.086$; AlexNet within-category correlations: $t(14) = 0.17$, $p = 0.86$; AlexNet between-category correlations: $t(14) = -1.31$, $p = 0.21$). These results demonstrate that the fine-grained structure in PFC that is captured by the BL-NET (but not the category model) is due to the geometry of between-category correlations, but not within-category correlations – i.e., that the BL-NET corresponds to the relative representational distances of individual exemplars to exemplars of *other* categories.

In addition to the fine-grained correlations between individual exemplars from different categories, the BL-NET differs from the category model because the distances between categories are not uniform in BL-NET. We thus tested whether the representational geometry of PFC activity could be explained by BL-NET when averaging the within-category correlations in a 6x6 RSM. In this analysis, we averaged exemplars in each category both for the BL-NET model and for the neural RSMs, and correlated the resulting 6x6 matrices during both encoding and prioritization. Note that in the resulting RSMs, the same-category correlations are not by definition 1 as in the category model, but reflect the average correlations among all exemplars within one category in BL-NET and PFC data. Thus, the categories are not equidistant from each other as they are in the category model. Results revealed that the BL-NET model is a good model of this categorical organization during prioritization ($t(14) = 3.76$, $p = 0.0021$) but not during encoding ($t(14) = 1.36$, $p = 0.196$) – again supporting our conclusion that particularly the between-category relationships can be well captured by BL-NET.

In summary, our results in these additional analyses provide a comprehensive description of the representational transformation observed in the PFC during the prioritization period, from a purely categorical to a less categorical and higher-dimensional format that specifically maps with the BL-NET but not with other DNN models. We report five new findings which support this interpretation: 1) Higher average values of between-category correlations during maintenance (observed at trend level); 2) lower variance of between category correlations during maintenance; 3) lower stress values of MDS projections during encoding as compared to maintenance; 4) significant matching to between-category but not within-category correlations of BL-NET, only during maintenance; and 5) fit with averaged between-category distances of BL-NET, only during

maintenance. We highlight that most of these differences in the neural RSMs between encoding and prioritization were observed for the between-category correlations, whose structure cannot be explained by the category model.

On a more general note, we acknowledge that alternative analytical strategies – e.g., analyses of representational subspaces as in Panichello and Buschman, *Nature*, 2021 – provide complementary ways to characterize representational formats. However, we argue that the degree of matching to RSMs derived from DNNs is of heuristic value because these models have previously been shown to match representations during sensory processing and have been widely applied to analyze representational transformations during various cognitive tasks. This is further elaborated in our response to comments # 4 and 5.

We have updated the results section of our manuscript and included these additional analyses. On p. 11-12, we now write:

“The specific alignment of the representational geometry of PFC activity with the last layer of BL-NET during the prioritization period suggests that the format of representations has been transformed in this region – from a purely categorical format during encoding into a format that incorporates distinctions among stimuli between categories during maintenance. To corroborate this transformation and characterize the representational formats observed in the PFC more comprehensively, we performed several additional analyses. First, we tested whether the average pairwise neural correlations differed between encoding and maintenance. Higher correlations of items during maintenance may point towards clustering of representations, while lower correlations would reflect the opposite, i.e. representations in a more widely spread representational space. Second, we analyzed the variance of correlations during encoding and prioritization. Higher variances would reflect less uniform (i.e., more distinctly organized and thus lower dimensional) distributions of items, while lower variances would correspond to an opposite pattern. We performed both analyses separately for items of the same category (within-category correlations) and items of different categories (between-category correlations). We found a trend for higher average between-category correlations during maintenance as compared to encoding ($t(14) = -2.12$, $p = 0.053$), and no significant differences in the average same-category correlations ($t(14) = -0.185$, $p = 0.856$). Moreover, the variance of between-item correlations decreased significantly from encoding to maintenance, both for items from different categories ($t(14) = 5.88$, $p = 4.03e-05$) and from the same category ($t(14) = 5.37$, $p = 9.86e-05$). We next compared the dimensionality of RSMs during encoding and maintenance. We projected the data in various dimensions using multidimensional scaling (MDS), and computed the stress of the MDS projections. Stress indicates the goodness of fit of a particular projection, and thus lower stress values during encoding or maintenance would indicate lower-dimensional representations during that time period. We observed that stress values were systematically lower during encoding as compared to prioritization in a cluster of significant dimensions (from 4 to 33 dimensions; $p = 0.0148$; Supplementary Figure 1A). Taken together, these results indicate a change from a purely categorical representation during encoding to a representation that matches the fine-grained architecture of the BL-NET during prioritization: PFC representations occur in a smaller representational space, occupy less clustered regions in this space, and rely on a higher-dimensional neural code.

Thus, our results point to a transformation of the representational format of PFC activity from encoding to maintenance.

We next investigated the fit of the BL-NET and the AlexNet networks during the prioritization period separately for within-category and between-category correlations. We observed a significant fit of the between-category correlations of RSMs from the BL-NET and neural data in the PFC ($t(14) = 3.69$, $p = 6.76e-05$; Supplementary Figure 1B), while this was not true for the AlexNet ($t(14) = 1.61$, $p = 0.13$). None of our models could explain the structure of within-category correlations (BL-NET: $t(14) = 0.42$, $p = 0.67$; AlexNet: $t(14) = -0.18$, $p = 0.86$; Supplementary Figure 1B). The results of the same analysis performed during encoding confirmed that neither BL-NET nor AlexNet are good models of activity in the PFC during this time period (BL-NET within-category correlations: $t(14) = -0.37$, $p = 0.71$; BL-NET between-category correlations: $t(14) = -1.84$, $p = 0.086$; AlexNet within-category correlations: $t(14) = 0.17$, $p = 0.86$; AlexNet between-category correlations: $t(14) = -1.31$, $p = 0.21$). These results demonstrate that the fine-grained structure in PFC that is captured by the BL-NET model is due to the geometry of between-category correlations – i.e., that the BL-NET corresponds to the relative representational distances of individual exemplars to exemplars of other categories”.

In addition, we included a more detailed description of the differences in the formats of the category model and our recurrent DNNs in the Discussion section of the updated manuscript (p. 16): “Our results provide a comprehensive description of the representational transformation observed in the PFC during the prioritization period, from a purely categorical to a less categorical and higher-dimensional format that specifically maps with the BL-NET but not with other DNN models. In detailed analyses of the geometry of PFC representations during encoding and maintenance, we found that PFC representations occur in a smaller representational space during the prioritization period, occupy less clustered regions, and rely on a higher-dimensional neural code. Notably, these differences were mostly observed for the between-category correlations (Supplementary Figure 1), whose structure cannot be explained by the category model. Considered together with the lack of EMS in PFC, these results point to a transformation of the representational format of PFC activity from encoding to maintenance, which is particularly due to a transformation of the geometry of the between-category correlations”.

In addition, we removed the term “abstract” when referring to the prioritized representational format in the PFC throughout the manuscript (p.9, p.11, p.15, p.17 and p.18).

2. It is insufficient to analyze neural activity, especially prefrontal activities, by extracting only the single-item trials condition and focusing only on the factor, visual object type. According to the task explanation in Methods, the retro-cue used in this experiment represents four types of rules: multi-item trials (“All”) and single-item trials (1, 2, or 3). It is well known that neural activity in the PFC strongly represents rules (e.g., Wallis et al., Nature 2001), varies with task difficulty (Barch et al., Neuropsychologia, 1997) and with the motivation of the individual. In this study, the retro-cue may have indicated the difficulty of the trial besides the behavioural rule for that trial itself. This is because even in the single-item condition, the recency effect and the primacy effect may have caused a difference in difficulty among the three object conditions. These higher-order representations of information including representation of rule and difficulty are expected to be

more pronounced in PFC than in VVS. In other words, in comparing VVS and PFC, this paper completely ignores the main effect and interaction associated with the factor, retro-cue, which are expected to be particularly strong in PFC. In this sense, it is not surprising that in the present study, the same pattern of neural activity as in the encoding phase was reinstated only in VVS and not in PFC during the time immediately after the retro-cue presentation. As such, the authors should use full task variables and clarify at the level of neural activity why the reinstatement did not occur in PFC alone.

3. In relation to the above point, the most important argument of this paper was that the activity immediately after the retro-cue presentation in PFC represents an abstract representational format (i.e., not just a categorical representation, but something higher-order than that). However, the analysis time window that the authors focused on to obtain this conclusion was the period immediately after the presentation of the retro-cue (0 – 500 ms after retro-cue onset, e.g., Figure 4H), and during this period, as mentioned above, there should be strong neural responses to the various information that the retro-cue have carried. It is therefore not surprising that the activity during this time period differed from that during the encoding period.

We thank the reviewer for raising these points, which we address together below. We note that we did not specifically select the time period of the retro-cue presentation for our EMS analysis in the VVS and PFC, but conducted it for all possible combinations of time points during encoding and maintenance. Similarly, we performed the DNN fit analyses at all possible time periods during maintenance, and found the effects specifically during the prioritization period in PFC. We agree with the reviewer on the need to assess the influence of other task variables during this period and use full task variables. We performed several additional analyses.

First, we evaluated whether behavioral performance for items at different positions differed in our task. We constructed a 2x2 repeated measures ANOVA to test the effect of position and type of trial (single or multi-item) on the overall memory performance. Performance at each position was computed as the number of correct responses divided by the total number of responses. Note that, for the multi-item trials, this is a different metric than the one employed in the original analyses reported in Figure 2D, where correct trials were those in which all three items were retrieved in their correct order (note that this metric has been updated in the revised manuscript). This implies that, contrary to the original analysis, chance levels between single and multi-item trials are equal in this analysis. Results of the ANOVA model revealed that indeed both position and type of trial had a significant effect on memory performance (position: $F(31) = 18.52$, $p = 4.72e-07$, type of trial: $F(31) = 10.28$, $p = 0.0031$). Items encoded in the first position were remembered significantly better than items in the second position ($t(31) = 5.11$, $p_{\text{Tukey}} = 4.55e-05$), and in the third position ($t(31) = 4.65$, $p_{\text{Tukey}} = 0.0001$). No significant difference was observed between items in the second and the third position ($t(31) = 0.09$, $p_{\text{Tukey}} = 0.99$; Supplementary Figure 4A). Moreover, performance for the single item trials was significantly better than performance for the multi-item trials ($t(31) = 3.21$, $p_{\text{Tukey}} = 0.0031$). In addition to the main effects of position and type of trial, we also observed a significant interaction between these two factors ($F(31) = 4.515$, $p = 0.0148$). We therefore conducted two separate one-way ANOVAs to assess the effect of position separately for single and multi-item trials. In the single-item condition, we found a main effect of item position ($F(31) = 3.279$, $p = 0.044$). Post-hoc analyses revealed a trend for higher performance of items

encoded in positions 1 vs 2 ($t(31) = 2.45$, $p_{\text{Tukey}} = 0.0509$), and no significant differences between positions 1 and 3 ($t(31) = 1.36$, $p_{\text{Tukey}} = 0.0509$), or 2 and 3 ($t(31) = -1.26$, $p_{\text{Tukey}} = 0.22$). In the multi-item trials, we also found a main effect of item position ($F(31) = 38.23$, $p = 1.52e-11$), and post-hoc analyses revealed higher performance for items encoded in positions 1 vs 2 and 1 vs 3 (1 vs 2: $t(31) = 6.06$, $p_{\text{Tukey}} = 3.0e-06$; 1 vs 3: $t(31) = 7.24$, $p_{\text{Tukey}} = 1.157e-07$), and higher performance for position 2 vs 3 ($t(31) = 2.93$, $p_{\text{Tukey}} = 0.017$). Thus, we observed primacy but no recency effects.

Given that the positions in which the items were encoded affected their probability of recall, we investigated whether the neural representations were also affected. First, to assess the presence of 'rule' representations, we evaluated whether items encoded in the same position had greater similarity than items encoded in different positions during the maintenance period (maintenance-maintenance similarity analysis, MMS). In this analysis, the original position of the cued item was taken, and contrasts were conducted separately for items of the same category (excluding same-exemplar correlations) and for items of different categories. We used the same temporal resolution as in the original analysis (i.e., 500ms windows sliding in 100ms; Figure 2D). Second, we assessed the reinstatement of category-specific similarity for items encoded at different positions (Encoding-maintenance similarity analysis, EMS).

(A) MMS analysis: Representation of position information in PFC but not VVS

During maintenance, the within-category analysis revealed that position information was significantly encoded in PFC representations in the theta frequency range ($p_{\text{corr}} = 0.02$; Bonferroni corrected for 5 bands; Supplementary Figure 4D). This was observed in a time period from 700ms to 1.6s after the presentation of the cue. The other frequency bands did not show a significant effect, even at an uncorrected level (all $p > 0.289$). We did not find any significant effect of item position in any band in the between-category correlations analysis (all $p_{\text{corr}} > 0.52$). In the VVS, none of the analyses revealed a significant effect of position (within-category: all $p > 0.362$; between-category: all $p_{\text{corr}} > 0.3$). Notably, these results show that during the maintenance period, the PFC encodes information related to the position of the items, while this information is not present in the VVS.

(B) EMS analysis: Reinstatement of category-specific information from items at individual positions in VVS but not PFC

To evaluate whether category-specific information was more prominent for items encoded at particular positions, we performed EMS analyses contrasting within-category vs. between-category correlations, separately for items encoded in positions 1, 2, and 3. The EMS analysis revealed a similar pattern of results as the main analysis in the VVS and in the PFC (Supplementary Figure 4B and 4C). In the VVS, we found significant EMS for position 1 in all bands except for the beta band (all $p_{\text{corr}} < 0.03$, beta: $p_{\text{corr}} = 0.2$) while position 2 showed significant effects in the theta and alpha bands (theta: $p_{\text{corr}} = 0.005$, alpha, $p_{\text{corr}} = 0.015$, other bands: all $p_{\text{corr}} > 0.44$; Supplementary Figure 4B). Position 3 showed significant EMS in the gamma ($p_{\text{corr}} = 0.02$) and high gamma bands ($p_{\text{corr}} = 0.01$), while the other bands did not show an effect (all $p_{\text{corr}} > 0.1$). In the PFC, we did not observe EMS in any of the positions and bands we tested (all $p_{\text{corr}} > 0.3$; Supplementary Figure 4C), consistent with the lack of EMS observed in the original analysis in this region (Figure 2D).

(C) Coding of position information in PFC lacks category specificity

To evaluate whether the behavior of the subjects (better performance for items in the first position) was reflected at the levels of EMS at each position, we directly contrasted category-specific EMS using a one-way within-subjects ANOVA with the factor “Position”, essentially corresponding to an interaction analysis of category and position information. We subtracted within-category and between-category correlations for each of the encoding positions and averaged these differences during the full encoding time period (0-800ms) and the prioritization time period (0-800ms after cue onset). Results revealed a lack of significant differences in levels of EMS for items encoded at different positions: No effect of position was observed in any frequency band in VVS (Encoding: all $F(25) < 1.48$; all $p_{\text{corr}} > 1$, Prioritization: all $F(25) > 4.27$; all $p_{\text{corr}} > 0.097$) or in PFC (Encoding: all $F(14) < 1.75$; all $p_{\text{corr}} > 0.95$; Prioritization: all $F(14) < 1.79$; all $p_{\text{corr}} > 0.92$). In the PFC, we also performed this interaction analysis in the frequency band and time period where we observed significant position MMS effects (3-8Hz, 700ms to 1.6s after the presentation of the cue). Results revealed no significant effect of position in EMS in this time period either ($F(14) = 0.231$, $p = 0.795$).

Taken together, these results demonstrate that the position at which the items were encoded significantly influenced PFC representations during maintenance. Notably, the effect of position was observed in an overlapping time period but a different frequency range (i.e., theta, 3-8Hz) as compared to the results observed in the DNN analyses (i.e., beta frequency range, 15-29Hz, 0.2-1s after the presentation of the retro-cue). These results thus suggest that PFC representations of different task variables rely on separable neural signatures. Importantly, we also note that EMS does not interact with position information in the PFC, suggesting that the lack of EMS in PFC is not due to the influence of a position code. Indeed, our results demonstrate that PFC representations encode stimulus features and task variables via separable neural signatures, embedding categorical, high-level visual and task representations in a multiplexed neural coding scheme. We added these results in the new Supplementary Note 3, and supplementary Figure 4.

4. Compared to previous publications, (e.g., Panichello and Buschman, Nature, 2021; Piwek et al., PLoS Comput. Biol., 2023; Meyers et al., TiCS, 2017), there seem to be little new findings in the present manuscript. For example, Panichello and Buschman (2021) recorded neural signals in the macaque lateral prefrontal cortex (LPFC) during a retro-cueing task, contrasting delay-period activity between before and after the retro-cue onset. They reported that in the delay period prior to the presentation of retro-cue, the individual stimuli were maintained in independent subspaces of neural population activity, whereas in the delay period following the retro-cue presentation, the prioritized items (i.e., items indicated by the retro-cue) was rotated into a common subspace, potentially allowing a common readout mechanism. Another study (Piwek et al., 2023) modelled the emergence of these representational changes using a recurrent neural networks (RNNs) that was actually trained to perform an equivalent retro-cue task. These results have reinforced the currently influential theory concerning neural mechanisms underlying retro-cue processing (Meyers et al., TiCS, 2017). In addition, neural mechanisms for attentional filtering (prioritization) in PFC has also been extensively studied (e.g., Everling et al., 2002). The authors should clearly state how the present manuscript differs from these previous reports.

We thank the reviewer for providing these relevant references, all of which have been incorporated into the revised version of our manuscript. We clarified the novel contribution

of our study compared to the existing literature. In particular, four aspects make our study a novel contribution to the field: 1) Our study is the first report on prioritized representations using human intracranial EEG. Many important previous studies on representational transformations during VWM prioritization – mentioned by the reviewer – have been conducted with non-human primates (e.g., Everling et al., *Nat Neurosci* 2002, Panichello and Buschman, *Nature* 2020), and iEEG recordings are ideally suited to bridge network level (EEG/MEG) studies to invasive recordings in monkey studies. 2) Our study investigates VWM prioritization using DNNs with RSA, while most previous studies have employed analyses on representational subspaces based on single unit data (e.g., Panichello and Buschman, *Nature* 2021). While both methods have their complementary value and importance, a critical difference is the mapping of DNN onto different processing stages during perception, which adds heuristic value to our findings (Yamins and DiCarlo, *Nat Rev Neurosci* 2016). 3) Our study is the first to employ a recurrent architecture using RSA in iEEG, demonstrating the relevance of recurrency for explaining cognitive representations. 4) Compared to computational studies, we record directly from the human brain and provide empirical evidence of the role of distinct representational formats and the relevance of beta-frequency oscillations emphasized in several previous models of WM (Miller et al., *Neuron* 2018; Spitzer & Haegens, *eNeuro* 2017).

Below, we specifically discuss the differences and novel aspects of our study in relation to the references provided by the reviewer. In the study by Everling et al., the focus is on how the prefrontal cortex filters unwanted representations during cued attention. While this research examines attention before items are encoded in WM, our study specifically addresses WM prioritization—how mnemonic representations, as opposed to perceptual ones, are modified by internal attention. A second important difference relates to the use of animals – monkeys –, in Everling et al., vs. human subjects in our study. Humans might use different strategies during VWM compared to monkeys and are not extensively trained. Training in non-human primates might influence WM representations, and their formats may thus differ from those in humans who conducted this task for only a few trials. Moreover, it is likely that the representations of natural images (used in both Everling et al., and our study) is different in monkeys and humans, particularly because of the integration of these visual representations into language and semantic networks in humans. This might not only affect how these representations are initially formed, but also their format during WM maintenance. Finally, the study by Everling et al. recorded brain activity at the level of single neurons, which differs substantially from the population-level activity in intracranial EEG recordings that we investigate. The mesoscopic level of description of iEEG is particularly well suited to bridge the literature on VWM prioritization in animals and human studies conducted with fMRI and EEG/MEG.

The study of Panichello and Buschman in *Nature* (2021), also conducted with monkeys and single-unit data, explores prioritization post-encoding using a retro-cue, similar to our paradigm. This study is highly relevant to our work and we had already included it as a reference in our original manuscript. Panichello and Buschman found that prioritization affects the structure of representational subspaces in monkey LPFC, which is assessed at the level of the firing rate of single units. Using sophisticated population geometry analyses, they report a rotation of the prioritized stimulus properties in the subspace of neuronal activity to align with relevant task dimensions and facilitate behavioral redout. Although this study investigates the representational geometry of

stimuli during item prioritization, similar to our study, the methodology it uses to quantify this geometry is different than ours and focuses on a different neurophysiological substrate. In particular, Panichello and Buschman look at the microscale and the firing rate of individual units, while we are focusing on the mesoscopic level of iEEG. Moreover, they use population geometry analysis versus model-based RSA that we employ. This allows them to test a different hypothesis than ours, which is centered on the representation of abstract task variables, while we focus on the visual representational formats of prioritized VWM representations. Moreover, while the stimuli employed by Panichello and Buschman are abstract colors and simple shapes, we employ natural images, which might trigger completely different responses in the VVS, particularly in higher level regions. Finally, we apply DNN models of natural image processing to process stimuli and assess the presence of visual representational formats, which is again a novel contribution beyond the study of Panichello and Buschman.

Our study also provides findings that extend and complement the computational study of Piwek et al. Our study provides experimental data extracted directly from the human brain, showing the transformations of VWM representations during item prioritization in the PFC grounded in biological computations rather than simulated results. Another key difference relates to the complexity of the task-performing models we employed, capable of object recognition in natural conditions, unlike the simpler, abstract models used in the Piwek et al. study. Methodologically, while both studies involve geometry analysis, we focus on mesoscopic signals rather than artificial unit firing rates and are particularly interested in the visual formats of WM, rather than characterizing the subspace of activity through population geometry analysis (see also differences with the Panichello and Buschman study).

We have summarized these main differences in a table (see below). Despite these critical differences, we acknowledge the shared focus on the question of item prioritization in WM, a pivotal aspect of current WM research. All studies discussed here align with the influential theory proposed in Myers et al., TICS, 2020. Specifically, they support a main argument of this theory: that task representations adapt their format to enhance behavioral readout.

In the Discussion section of the updated manuscript, we provide a more detailed description of the novel contributions of our study with respect to previous literature (p. 22): *“Many important previous studies on representational transformations during VWM prioritization have been conducted with non-human primates (e.g., Everling et al., 2002, Panichello and Buschman, Nature, 2020). Our study is the first report on prioritized representations using human intracranial EEG, which provides a level of analysis ideally suited to bridge network level (EEG/MEG) studies on VWM (e.g., Brookes et al., 2011) to invasive recordings in monkey studies. In addition, while previous studies have employed analyses on representational subspaces based on single unit data (e.g., Panichello and Buschman, 2021), or computer simulations (Piwek et al., 2023; Wan et al., 2022), we employ DNNs and RSA. While both methods have their complementary value and importance, a critical difference is the mapping of DNN onto different processing stages during perception, which adds heuristic value to our findings (Yammins and DiCarlo, Nat Rev Neurosci, 2016).”*

Additionally, we have incorporated the references suggested by the reviewer in the introduction (p. 3-4).

Study	Analysis Method	Type of data	Empirical vs Computational	Attention vs Memory prioritization
Everling et al., 2002	Firing rate analyses	Single units	Empirical	Attention
Panichello & Buschman, 2020	Population geometry analyses	Single units	Empirical	Prioritization
Piwek et al., 2023	Population geometry analyses	Simulations	Computational	Prioritization
Pacheco Estefan et al., 2024	RSA	iEEG	Empirical	Prioritization

5. In drawing main conclusions of the paper (Figures 4 and 5), the authors only used DNNs trained on the object classification task (Method). These DNNs were not trained on the retro-cue task which was used to collect neural activity. In other words, the DNNs were exposed only to the object image input, while the neural activity was fed two types of information, object image and retro-cue. Therefore, because these DNNs do not receive retro-cue input to induce prioritization in the first place, the use of such DNNs can be insufficient for examining the transformation process of category signals induced by the retro-cue. Can one really compare two systems with different input and analyze the transformation process? If so, please specify the rationale for that.

We thank the reviewer for this comment. It would be interesting to compare PFC representations in networks trained with the objective function of item prioritization, but we would like to clarify why we believe that representations in DNNs trained on object classification could nevertheless be useful models for representations during VWM. We note that the use of a low-dimensional task objective (e.g., image classification) to monitor cognitive representations during higher-level functions (e.g., visual working memory maintenance) has received theoretical support (Cowell et al., *eNeuro* 2019, Murray et al., *Annu Rev Neurosci* 2007). One prominent theory is the representational-hierarchical model (Murray et al., *Annu Rev Neurosci* 2007), which posits that different operations (e.g., those that support memory and those that support perception) can act on representations throughout all levels of the VVS and medial temporal lobe (MTL) hierarchy. According to this theory, it is not the cognitive process (e.g., memory versus perception) that defines how regions along this pathway contribute to cognition, but rather the representational content required for any given cognitive process, be it perception or memory (see also Cowell et al., *eNeuro* 2019). Indeed, representational accounts of memory have argued that regions representing particular content in the brain (e.g., low-level visual features in early visual regions) are involved in the representation of these features irrespective of the cognitive process in which they are engaged (Cowell et al, *eNeuro* 2019; Barense and Lee, *Neuron* 2021). Since the VVS plays a role both during object recognition and VWM for these

objects, it is relevant to investigate the representational format of items during both processes, and DNNs are arguably strong tools to capture these formats (Yamins et al., *PNAS* 2014; Davis et al., *Cereb Cortex* 2021; Liu et al., *Sci Adv* 2021; see also Xie and Zaghoul, *Curr Biol* 2020).

In addition to these theoretical considerations, we underscore the widespread practice in our field of using networks pretrained in a particular task to characterize representations formed in a different task. In previous work, the DNN network AlexNet, trained for image classification, has been employed to investigate the representational formats of representations during both VWM and long-term memory (Davis et al., *Cereb. Cortex* 2021, Liu et al., *Sci Adv* 2021). A similar trend is observed in natural language processing, where language models trained in the task of next word prediction have been applied to model language-related brain responses more broadly (Schrimpf et al., *PNAS*, 2021; Goldstein et al., *Nat Neurosci* 2022; Goldstein et al., *Nat Commun* 2024; Caucheteux et al., *Sci Rep* 2022; Tuckute et al., *Nat Hum Behav*, 2024). These studies did not assume that these models capture all the complexity of VWM or language, but that they are appropriate models to explain the formats of representations used in these tasks. For example, in the case of VWM, the assumption is that convolutional DNNs provide a way to assess the presence of specific visual information and particular visual representational formats in the neural data.

Notably, DNN models do not have a precise anatomical correspondence with brain regions. Some studies have proposed a hierarchical correspondence of representations between DNNs and biological brains, linking superficial DNN layers to early visual regions and deep layers to higher-level regions (Yamins et al., *PNAS* 2014, Kuzovkin, *Nat Commun Biol* 2018; Eickenberg et al., *Neuroimage* 2017; Dwivedi et al., *Plos Comp Biol* 2021; Cichy et al., *Sci Rep* 2016; Guclu and van Gerven, *J Neurosci* 2015). However, this has been recently questioned in a study showing that deep DNN layers better explain representations in all brain regions, when these are relevant to the task (Sexton and Love, *Sci Adv* 2022). Previous iEEG studies have employed DNNs to track representations across brain locations without any specific anatomical commitment, extracting representations from brain-wide patterns (e.g., Liu et al., *PNAS* 2020, Liu et al., *Sci Adv* 2021). In our study, we go beyond these established practices in iEEG research to investigate the specific representational formats of prioritized content in two regions that are crucial for VWM: PFC and VVS. However, we were agnostic regarding which particular network layers should be mapped to which particular brain regions, and explored the fit of all network layers to all regions.

Our approach assumes that representations during VWM have different representational formats which can be sufficiently well explained by neural architectures trained to classify natural images. DNNs have been described as the best available models of VVS activity during visual perception (Yamins and DiCarlo, *Nat Rev Neurosci* 2016). Despite possible changes in neural representations related to the different demands in our task (VWM as opposed to visual perception), our study shows that DNNs trained on image classification can significantly explain the variance of the neural data related to the visual properties of stimuli during VWM encoding and maintenance.

We note that at least partially, the task of object recognition is embedded in our paradigm, because subjects are recognizing objects when they are presented with the

images during encoding. We acknowledge that object recognition is only a subprocess of VWM encoding. Notably, cognitive demands are very different during item prioritization, where higher-order task variables might critically affect representations, particularly in PFC, as the reviewer points out (see the previous comment).

Since we agree this is an important point, we have extended the section in the Discussion where we addressed it in the original manuscript (p. 19), which now reads: “[...], we note that the use of networks trained in a lower dimensional task objective, i.e., image classification, to model cognitive representations embedded in a higher-level cognitive process, i.e., VWM prioritization, has received some theoretical support. Indeed, representational accounts of memory have argued that it is not the cognitive process (e.g., memory versus perception) that defines representations, but rather the content that any given cognitive process requires. Indeed, regions representing particular content in the brain (e.g., low-level visual features in early visual regions) are involved in the representation of these features irrespective of the cognitive process in which they are engaged (Barense & Lee, 2021; Cowell et al., 2019; Murray et al., 2007). Since the VVS plays a role both during object recognition and VWM for these objects, it is relevant to investigate the representational format of items during both processes, and DNNs are arguably strong tools to capture these formats (Davis et al., 2021; Liu et al., 2021). Beyond these theoretical considerations, we underscore the widespread practice in our field of using networks pretrained in particular tasks to characterize representations formed in different tasks. Previous studies have employed the AlexNet network to investigate the representational formats of representations during both VWM and long-term memory (Davis et al., 2021; Liu et al., 2020, 2021). A similar trend is observed in natural language processing, where language models trained in the task of next word prediction have been applied to model language-related brain responses more broadly (Caucheteux et al., 2022; Caucheteux & King, 2022; Goldstein et al., 2022, 2024; Schrimpf et al., 2021; Tuckute et al., 2024)”.

6. The RSM in layer 7 of BL-net (Figure 4E, rightmost panel) and that in layer IT of corNET-RT (Figure 5E, rightmost panel) look very similar to the RSM (model RSM) in Figure 2A which was based on a simple model of category information that the authors used to analyze neural activity. In other words, the RSM in layer 7 of BL-net and layer IT of corNET-RT appear to be simply a noisy version of the model RSM in Figure 2B. This brings up one question: isn't calculating the correlation between rDNN RSM and neural RSM in Figure 4H (layer 7, rightmost panel) and Figure 5G (layer IT, rightmost panel) essentially doing the same thing as calculating the correlation between the model RSM and neural RSM in Figure 2B? In fact, the resultant fit presented in Figure 2B and Figures 4H/5G look very similar to each other, except that there was a period of significant correlation only in Figures 4 H and 5G. Please elaborate further on this. Also, could you please quantify how similar the time-frequency correlation maps for the maintenance period were between Figure 2B and Figure 4H/5G (rightmost panels)?

If these two analyses (i.e., calculating the correlation between rDNN RSM and neural RSM in Figure 4H/5G, and calculating the correlation between the model RSM and neural RSM in Figure 2B) are essentially doing the same thing, then, doesn't the observed significant cluster in beta band in PFC (Figure 4H rightmost panel) merely indicate a presence of a variant of category specificity as already shown in Figure 2 (for encoding period only)?

On a conceptual level, we would like to emphasize that the category model only codes binary information about category membership, while the DNNs' deep layers in addition reflect more subtle differences among stimuli which encode high-level visual properties of images (see also our response to comment #1). The two models therefore not only represent different hypotheses about how the brain represents visual information, but they also differ in the aspects of the representational geometry they can model. A key distinction is that the category model is agnostic to any structure in the within-category and between-category correlations (which are all modelled identically with ones and zeros respectively), while the DNN models propose a very specific geometry for these two types of relationships. Thus, fitting the two models to PFC activity provides complementary information regarding the geometry of representations in this region. Notably, while the models are not mutually exclusive (orthogonal), we have shown a dissociation in their levels of fit during encoding and prioritization: The category model explains well representations during encoding but not maintenance, while the reverse is true for the recurrent DNN models.

Nevertheless, since the DNNs are trained to identify categorical information, one would expect that representations of items from the same categories are more similar than those from different categories, as in the category model. This is why the correlations of PFC activity and the category model on the one hand, and the DNNs on the other hand, are indeed similar. Following the suggestion of the referee, we quantified the similarity of the category model and the deep layer rDNN fits, focusing specifically on the BL-NET model (layer 7). While we observed numerical differences in the levels of fits in particular time-frequency periods (Supplementary Figure 8), these did not survive correction for multiple comparisons (shuffling model labels 1,000 times: all $p > 0.6$). In addition, we correlated the concatenated time-frequency maps of BL-NET fits and category model fits in each participant independently. We observed a mean correlation of 0.635 ± 0.035 (Mean \pm STD), and R^2 of 0.4 ± 0.04 (Mean \pm STD), corresponding to the shared variance of the time frequency maps (Supplementary Figure 8). These results suggest that categorical information is an important dimension of representation during the prioritization period in the PFC. We note however that in this comparison, the fits of the category model and the BL-NET are performed on the *same* neural data, and therefore some shared explained variance in the time-frequency maps is expected.

How can these results be reconciled with the fact that categorical information is not sufficient to explain PFC representations during the prioritization period? We tested two hypotheses, following the reviewer remarks: First, it may be that a noisy version of the category model would provide a better fit with PFC representations than a version without noise; second, the specific representational geometry of individual exemplars in the BL-NET matches the geometry in the PFC consistently across our group of subjects.

We tested the first hypothesis by simulation analyses. We performed two types of simulations: 1) We correlated the category model under different levels of noise with BL-NET representations, and 2) we correlated the category model under different levels of noise with PFC representations. In both analyses, we started with the category model and added Gaussian noise with a mean of 0 and standard deviations of 0, 0.1, 0.5, 1, 3, 5, and 10. In the first analysis, we computed the correlation of the resulting noise models with BL-NET representations (layer 7) 1,000 times (Supplementary Figure 9A). We found that

BL-NET representations were significantly more similar to PFC representations during the prioritization period than they were to the noisy version of the category model representations. The correlations of BL-NET and PFC representations resulted in a t-value of 5.88 at the group level (red line in histogram plots, Supplementary Figure 9A). Adding noise to the category model did never reach a similar level, in any of the permutations and the tested levels of noise. Notably, results in this simulation demonstrate that the BLNET RSM is not a noisy version of the category model, because adding random noise to the category model never leads to a similar level of fit with the neural data as the BL-NET model.

In the second simulation analysis, we assessed how well the noisy versions of the category model fitted the neural data in PFC separately for within-category and between-category correlations. In each subject, we took the average RSM in the cluster where we observed significant effects in the original BL-NET analysis during the maintenance period (layer seven, beta range). **The number of significant correlations were no different from what is expected by chance at all levels of noise** (within-category correlations: n = 4.8%, 4.9%, 4.7%, 4.7%, 4.6%; between-category correlations: n = 4.4%, 5.2%, 4%, 5.5%, 4.6%, 5.1%). These results show that the representational geometry of within and between-category items in the PFC during encoding and prioritization cannot be explained by random noise.

To test the second hypothesis, we investigated how consistently the BL-NET reflects the representational geometry of stimuli in PFC. If BL-NET is a good model of PFC representations, we would expect to see that the representational geometries are closely related in most of our subjects. We thus computed the correlation of neural and BL-NET RSMs during the prioritization period in each subject independently. We focused on the cluster of significant time-frequency bins in the original BL-NET analysis (layer seven, beta range). We also computed a distribution of correlation values expected by chance for every subject, by shuffling the labels of the model RSMs 1,000 times. Results are plotted in Supplementary Figure 9B, which indicates the percentage of times in each subject that the observed correlations with BL-NET were higher than the correlations expected under the null hypothesis. Notably, the fit was substantially higher with BL-NET in the large majority of cases (>88% in 11 out of 15 subjects). The fact that we observed this consistent representational geometry in our group of subjects makes it highly unlikely that the group level correlation we observed in the main analysis is driven just by noise.

We would also like to highlight the results presented in response to the first comment of the reviewer, where we specifically tested the fit of BL-NET and AlexNet for the between-item correlations, an analysis that is not possible to do with the category model. Our results in this analysis revealed that BL-NET is a good model of the between-category correlations in PFC ($t(14) = t = 5.58, p = 6.76e-05$), demonstrating that it explains an aspect of the variance that the category model cannot explain by definition. Together, these additional analyses show that while the geometry of representations in PFC is categorically structured, the fine-grained structure of between-category similarities cannot be captured by noise, but by the very specific representational geometry of the BL-NET model. We have added these results to the new Supplementary Note 5. In addition, we have included the following lines in the Discussion section of the main manuscript (p. 20):

“We note that the different models we employed (BL-NET, AlexNet, Category model) do not only represent different hypotheses about how the brain represents visual information, but they also differ in the aspects of the representational geometry they can model. For instance, the category model only codes binary information about category membership, while the DNNs’ deep layers in addition reflect more subtle differences among stimuli which encode high-level visual properties of images. The category model is by definition agnostic to any structure in the within-category and between-category correlations (which are all modelled identically, with ones and zeros), while the DNN models propose a very specific geometry for these two types of relationships. Thus, fitting the two models to neural data provides complementary information regarding the geometry of representations. Notably, while the Category Model and BL-NET are not mutually exclusive (orthogonal), we have shown a dissociation in their levels of fit during encoding and prioritization: The category model explains well representations during encoding but not maintenance, while the reverse is true for the recurrent DNNs”.

7. Regarding the electrode locations, medial, lateral and orbital prefrontal electrodes are all analyzed together as “prefrontal electrodes”. It is known that category information, rule, and attentional prioritization are mainly processed in the lateral prefrontal cortex. I would be very interested to see how the results would look like if the electrodes were limited to only those in the lateral frontal cortex.

Thanks for the suggestion. We repeated all PFC analyses during the maintenance period by including only electrodes in lateral frontal regions. For this purpose, we excluded electrodes with MNI x-coordinates smaller than -35 or larger than +35. We also excluded electrodes with MNI coordinates < -15 in the superior-inferior axis (z-coordinate), which effectively removed all orbitofrontal electrodes. The new selection resulted in a group of 9 subjects with a total number of 38 electrodes, which were located in the following Freesurfer regions: *rostral middlefrontal, pars triangularis, caudal middlefrontal, pars orbitalis, pars opercularis* (Supplementary Figure 12A).

We correlated again the BL-NET and the PFC RSMs during the maintenance period. We observed very similar results as in our original analysis reported in Figure 4H (in the same direction, and even more pronounced): a significant match between BL-NET representations and LPFC representations in the deepest layer, locked to the presentation of the cue in the beta band ($p_{\text{corr}} = 0.007$, Bonferroni corrected for 7 layers; Supplementary Figure 12B), while no significant fits were found in any other layers (all $p_{\text{corr}} > 0.33$). Consistent with our original analyses, we did not observe any significant fit in the analysis of the category model (all $p > 0.47$; Supplementary Figure 12C), and with any layer of the AlexNet network (all $p_{\text{corr}} > 0.476$; Supplementary Figure 12D).

We have included these results in the new Supplementary Figure 12 and modified the methods section of our manuscript (p. 34-35), which now reads: *“In additional analyses, we specifically analyzed activity in the lateral prefrontal cortex (LPFC), a brain region that has been associated with attentional prioritization (Everling et al., 2002), the representation of rules (J. D. Wallis et al., 2001) and categories (Cromer et al., 2010) in non-human primates. We excluded all PFC electrodes with MNI x-coordinates smaller than -35 or larger than +35 and z-coordinate < -15. The new selection resulted in a group of 9 subjects with a total number of 38 electrodes in the LPFC, which were located in the*

following Freesurfer regions: 'rostral middlefrontal', 'pars triangularis', 'caudal middlefrontal', 'pars orbitalis' and 'pars opercularis' (Supplementary Figure 12)."

Specific comments:

1. In Fig. 2, the authors should label the task periods, the meaning of time zero, and the units of values in a colorbar within the figures. Readers should not have to guess or look for this information.

We have now included labels for the different task periods in Figure 2 and all other figures. We have indicated the meaning of time zero in all figures and captions. We included the label "Cue" in the plots of the maintenance period in Figures 2-5. In addition, we added information regarding the color bar in the caption of all figures.

2. In Figure 1C, the electrodes are mapped on the transparent brain, which is very difficult to see. Please illustrate them in a different format or at least add multiple coronal slices from anterior end to posterior end of the electrode locations.

We have improved the visualization of the electrodes and are now plotting them on a non-transparent brain surface (Figure 1C). Our new visualization method makes the electrodes visible even if they are occluded by the surface of the brain. We have also highlighted the Freesurfer regions corresponding to the areas included in the analyses (Figure 2B and Supplementary Figure 12).

3. Lines 170-172: the authors directly compared proportion correct rates between the two conditions which had different chance levels. Because the single-item and multi-item conditions have very different chance levels (0.167 and 0.0083, respectively), this result does not necessarily reflect the difference in performance between the two tasks.

We explored various approaches for comparing performance, considering the different chance levels in single and multi-item trials. First, we subtracted the chance levels in each condition and assessed performance relative to this difference (i.e., whether performance differed from $0.167 - 0.0083 = 0.1587$). This analysis revealed a significantly higher performance for single as compared to multi-item trials ($t(31) = 3.21$, $p = 0.003$). Second, we evaluated a normalization technique, standardizing scores as relative improvements with respect to chance, individually in each condition. We used the following formula: $\text{Performance} = (\text{Proportion correct} - \text{Chance Level}) / (1 - \text{Chance Level})$. Again, we observed a highly significant difference between the single and multi-item conditions ($t(31) = 9.86$, $p = 4.49e-11$). Finally, we evaluated performance by calculating the proportion of correct responses at each encoding position separately for single and multi-item trials. In this approach, multi-item trials provide more data due to each item being presented three times within a trial, impacting the statistical power in the two conditions. However, both conditions have equal chance levels (1/6). Again, we found significantly higher performance for items in the single-item condition ($t(31) = 3.21$, $p = 0.0031$). After careful consideration of these different approaches, we suggest quantifying performance

differences using the latter, more unbiased measure, which eliminates the need for normalization. We have accordingly updated the revised manuscript in the main text (p. 6), and in Figure 2.

4. Lines 348-349: Ecoset-trained BL net was only applied to the time-frequency clusters that showed significant results in the ImageNet-trained BL net. The same analysis should be performed for all time-frequency ranges for the eco-set BL net as in Figure 4 and the results should be presented.

We have now included a figure with the results of the BL-NET analysis conducted for all time-frequency bins (Supplementary Figure 10).

5. Lines 398 and 403: In the discussion section, the authors stated that in the encoding phase, “PFC exhibited robust category-specific representations during WM encoding” (L403), but the representation in PFC “matched only high-level visual and abstract formats from a recurrent DNN following the retro-cue.” Again, what is meant by high-level visual and abstract formats is still not clearly mentioned here. Since this is the most important argument of the paper, it should be specifically indicated with some evidence.

We apologize for the lack of clarity in the terminology we employed in the manuscript. As we have argued in response to comment #1 (and to comment #5 of reviewer 2), we agree that the term ‘abstract’ does not appropriately describe the representational formats we observed in PFC (both the categorical coding scheme during encoding and the format during maintenance that fits to BL-NET are abstract, while the latter also captures high-level visual information). We have now removed the term ‘abstract’ from the manuscript and refer to the representational format observed in PFC during encoding as ‘categorical’, and during prioritization as ‘prioritized’. Regarding the ‘high-level visual’ formats, we have now clarified the meaning of this term, which specifically refers to the deep layers of our convolutional DNNs.

We have changed the discussion section in the lines highlighted by the reviewer, which now read (p. 15): “*Matched only the deepest layer of a recurrent DNN following retro-cue, suggesting a prioritized format in which high-level visual features of images are preponderant*”.

REVIEWERS' COMMENTS

Reviewer #1 (Remarks to the Author):

The authors have substantially revised their manuscript in response to the first set of reviews. I believe they have done a sufficient job in addressing the concerns that I raised, and have provided important clarifications and additional analyses to support their overall conclusions. I agree with the other reviewers regarding the inferences that can be drawn when comparing these two different DNN models, but it appears to me that with this revision the authors have addressed many of those concerns.

Reviewer #2 (Remarks to the Author):

The authors have thoroughly addressed by previous remarks.

Reviewer #3 (Remarks to the Author):

The authors answered my concerns fully and clarified them all. In the revised manuscript, the description of the analytical methods and results has been significantly improved to better illustrate the importance of the findings of this study to a broader audience.

In particular, for my question 4), there was a very clear answer regarding the comparison between previous primate and the present (human) studies. Responses to my points 5-6 are also very clear, and I appreciate the additional analysis concerning my point 7; the removal of the orbital prefrontal electrodes. I am now fully convinced that this paper would bring important insights to the field.

Reviewer #3 (Remarks on code availability):

Each program was well commented and included instructions on which data to load. It is relatively easy for the reader to know which code to execute in sequence to reproduce the results of each figure.